# Obesity rise plateaus in developed nations and accelerates in developing nations

NCD Risk Factor Collaboration (NCD-RisC)*

Global reporting of obesity is commonly based on comparisons over multiple decades[1] and lacks a granular and systematic analysis of its dynamics. We used 4,050 population-based studies with measured height and weight data on 232 million participants to assess the worldwide dynamics of obesity from 1980 to 2024. The rise in obesity decelerated in school-aged children and adolescents throughout the 1990s in many high-income countries, and subsequently plateaued in most at age-standardized prevalences spanning 20 percentage points, from 3–4% for girls in Japan, Denmark and France to 23% for boys in the USA. There were indications of a small decline in obesity in children and adolescents in some high-income western countries (for example, Italy, Portugal and France) since the 2000s. Similar trends were seen in some countries in Central and Eastern Europe. In adults, the rise in obesity slowed down in high-income western countries about a decade after children, followed by a plateau or possibly a small reversal of the rise in some countries (for example, Spain). In most low-income and middle-income countries, the annual absolute change in prevalence has remained stable or increased over time, even though prevalence has surpassed that of high-income countries. These highly varied dynamics suggest that the social, economic and technological trends that influence the availability, affordability and use of different foods may have helped control the rise in obesity in high-income countries, but require policy interventions in low-income and middle-income countries.

Obesity increases the risk of cardiovascular, renal, liver and respiratory diseases, musculoskeletal and neuropsychiatric disorders, diabetes, some cancers, adverse reproductive and obstetric outcomes, and severe COVID-19. Obesity is currently more prevalent than in the late twentieth century[1], and since the 1990s, the term 'epidemic' has been used to describe its rise[2,3].

Change in the prevalence of obesity in a population is driven by changes in height and weight, which themselves result from the quantity and quality of nutrition, the living environment and physical activity. These determinants of obesity vary across countries and change over time owing to changes in food production, processing, storage and transportation technologies that affect the availability and cost of various foods; economic resources of countries and households; social norms and knowledge; corporate and commercial practices; and fiscal and regulatory policies that affect food price and availability[1,4–13]. Despite these dynamics, global reporting of obesity has typically compared prevalence over decades and has not systematically evaluated the trajectory of obesity in a population at more granular timescales[1]. Rather, descriptions of how obesity trends evolve over time in different countries have been largely qualitative[14].

The multi-decadal timeline limits our ability to benchmark the long-term and recent performance of countries in controlling obesity, to set priorities for nutrition and public health programmes and policies, to provide access to health care such as weight loss medications or bariatric surgery, and to assess the impacts of these policies and programmes. To systematically and consistently investigate the dynamics of obesity in different countries, we analysed its velocity, calculated as the annual absolute change in prevalence and reported in percentage points per year. Positive velocity indicates an increase in prevalence, and negative velocity a decrease. We report velocity from 1980 to 2024 for school-aged children and adolescents (5–19 years of age) and adults (20 years of age and older). We analysed children and adolescents separately from adults because cut-offs for underweight and obesity differ between them and because obesity trends and dynamics may differ between school ages and adulthood[1].

To estimate velocity and characterize the dynamics of the obesity epidemic, we used 4,050 population-based studies that measured height and weight in 232 million participants 5 years of age and older (Supplementary Tables 1 and 2 and Supplementary Figs. 1 and 2). We used these data to calculate body-mass index (BMI), which was then used in a Bayesian hierarchical meta-regression model to estimate the prevalence of obesity from 1980 to 2024 in 200 countries and territories (referred to as countries hereafter). Obesity was defined as BMI ≥ 30 kg m$^{-2}$ for adults and BMI > 2 s.d. above the median of the WHO growth reference for children and adolescents. We calculated velocity as the annual absolute change in the estimated prevalence. We also used dimensionality reduction and clustering techniques to identify phenotypes of national obesity trajectories, defined as clusters of countries with similar estimated obesity prevalence time trends over the 45 years of analysis. Details of data and methodology are provided in Methods.

*A list of authors and their affiliations appears online.

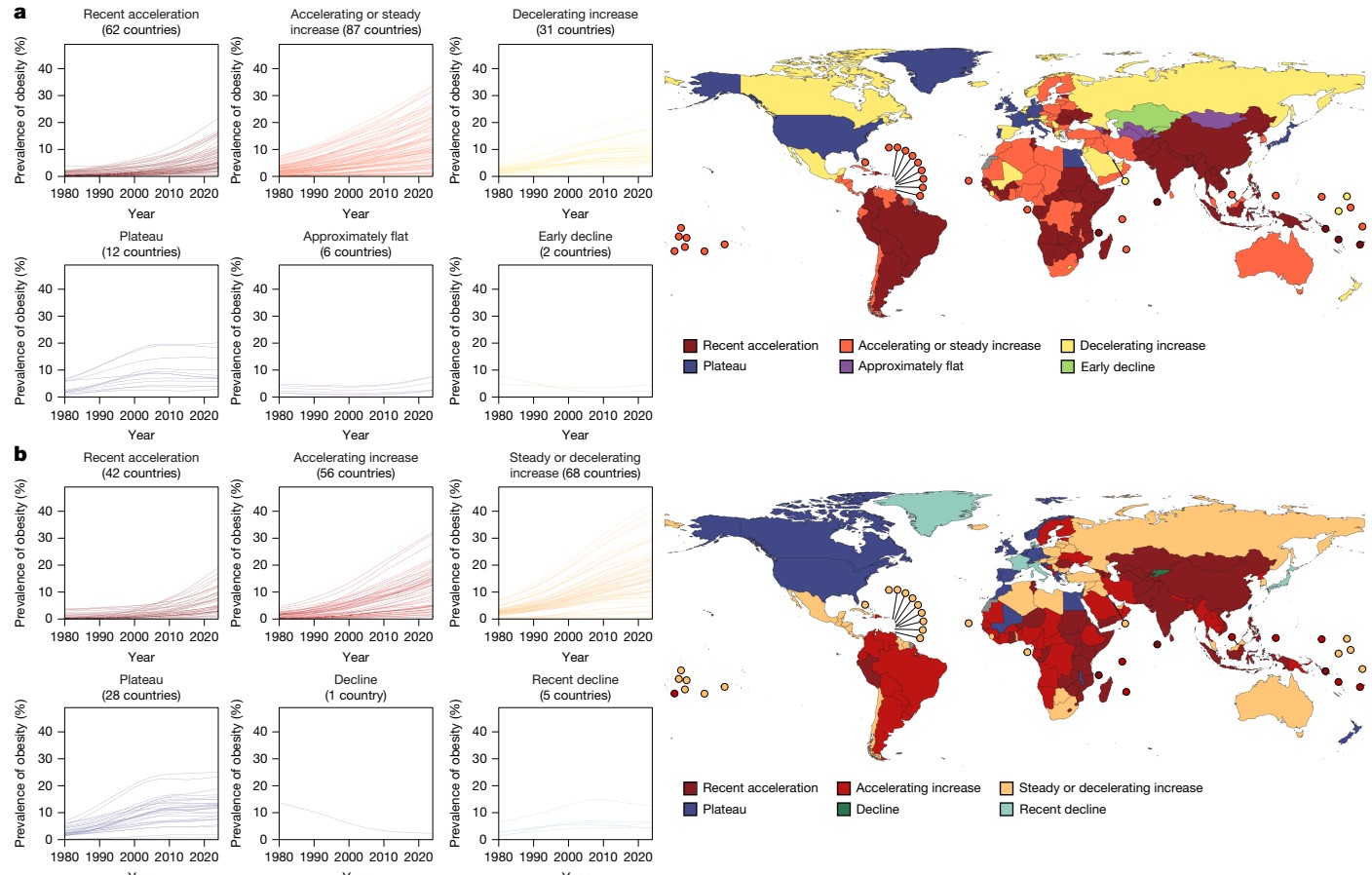

**Fig. 1 | Phenotypes of national obesity trajectories in children and adolescents. a,b,** Time-series plots of age-standardized obesity prevalence (left) and cluster allocation of these trends by country (right) for girls (**a**) and boys (**b**) from 1980 to 2024. Each panel represents a cluster of countries with similar shapes of obesity prevalence time series, and is labelled with its typology of trend. Each line on the plots represents obesity prevalence over time in one country. The maps display countries coloured according to their cluster allocation. See Extended Data Figs. 1 and 2 for allocation of countries in each super-region to clusters. See Supplementary Fig. 3 for trends in age-standardized obesity prevalence by country.

## Children and adolescents

From 1980 to 2024, age-standardized prevalence of obesity increased with a posterior probability (PP) > 0.80 in all but 5 of 200 countries for girls, and all but 2 of 200 countries for boys. The countries that did not experience an increase were in Central Asia plus France for boys. The increase in obesity over these 45 years ranged from 0.6 to 27 percentage points among girls, and from 0.4 to 35 percentage points among boys. National obesity trajectories had highly heterogeneous dynamics during this period.

In most high-income western countries (that is, high-income countries in Western Europe, North America and Australasia), as well as in Japan and Taiwan, the rise in obesity prevalence among school-aged children and adolescents predominantly occurred before the beginning of the millennium; this rising trend has slowed down, plateaued or may have even reversed slightly since then (Figs. 1 and 2 and Extended Data Figs. 1 and 2). The earliest slowdown occurred around 1990 in Denmark for both sexes, followed by some other Northwestern European countries including Iceland, Switzerland, Belgium and Germany through the 1990s (Fig. 2). By the mid-2000s, obesity prevalence among school-aged children and adolescents started to stabilize in most high-income countries, and may have even started to decline in some. In 2024, the velocity of obesity was below 0.25 percentage points per year in most of these countries and may have become negative in some (for example, Italy, Portugal and France). The negative velocities had magnitudes smaller than 0.15 percentage points per year. These small velocities, including all the negative velocities, were indistinguishable from zero at a PP of 0.80 (Figs. 2 and 3 and Extended Data Figs. 3 and 4). Beyond this plateauing, in some high-income countries, such as France, the Netherlands, Switzerland and Japan, velocity remained low (less than 0.2 percentage points per year) over the entire 45-year period. The exceptions to this plateauing and reversal in high-income western countries were among girls and boys in Australia, Finland and Sweden, where the prevalence of obesity increased steadily or even accelerated.

Slowdowns or plateaus in the rise of obesity among school-aged children and adolescents also occurred in some countries in Central and Eastern Europe (for example, Croatia and Slovenia for both sexes and Czechia and Montenegro for boys) and in some middle-income countries where obesity prevalence in school ages was relatively high (for example, Mexico and Kuwait). In most of these countries, the slowdown of the rise in prevalence started in the 2000s, about a decade after the slowdown began in high-income western countries (Fig. 2). Furthermore, in some countries in Central Asia, such as Kyrgyzstan and Kazakhstan, school-aged children and adolescents, especially girls, did not experience the rise in obesity seen elsewhere throughout these four decades, or for parts of it experienced a decline (Fig. 1).

The plateauing and any possible reversal of the rise in obesity in children and adolescents happened at vastly different prevalences across countries. In many high-income countries in Western Europe and Japan, age-standardized prevalence plateaued below 10% in school ages. For example, obesity prevalence has had near-zero or negative

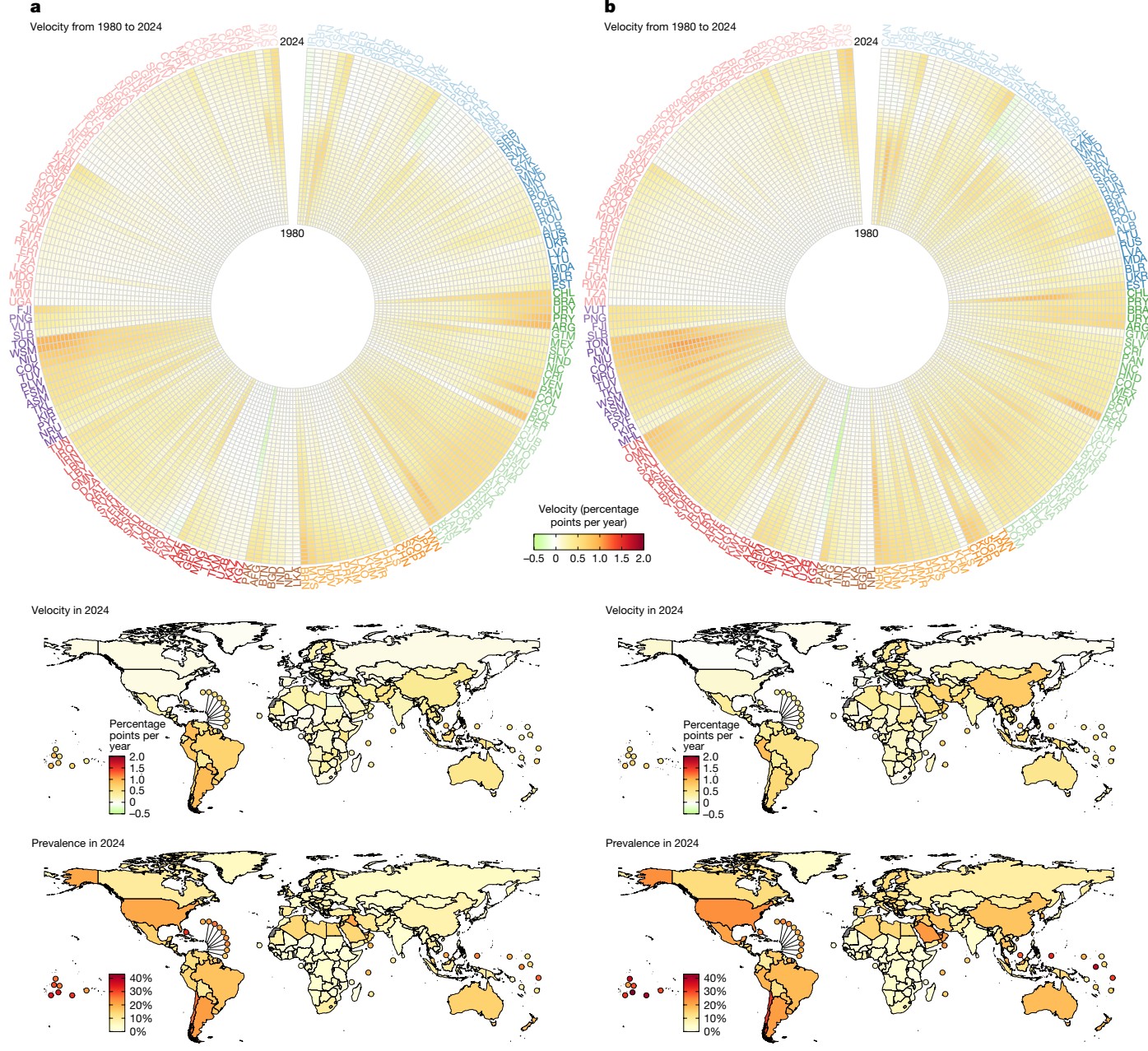

**Fig. 2 | Velocity and prevalence of obesity in children and adolescents.**
**a,b**, Velocity and prevalence of obesity in girls (**a**) and boys (**b**). The circular wheel plots show the year-on-year velocity of obesity from 1980 to 2024 by country. Each cell represents the velocity in one year. A positive velocity (red) indicates a year-on-year increase in obesity prevalence, whereas a negative (green) velocity indicates a year-on-year decrease. Years when no change in obesity was observed are coloured in white. Countries are labelled by their International Organization for Standardization (ISO) 3166-1 alpha-3 codes (Supplementary Note 1) and coloured by super-region. Countries are ordered by increasing 2024 velocity within each region. The top maps show the velocity of obesity in 2024 in each country, following the same colour scheme as the velocity wheel plots, and the bottom maps show the age-standardized prevalence of obesity in 2024. See Extended Data Fig. 3 for velocity of obesity in 2024 and its uncertainty by country, Extended Data Fig. 4 for a map of PP that velocity of obesity was positive in 2024, Supplementary Fig. 3 for trends in the prevalence of obesity by country and Supplementary Fig. 5 for trends in the velocity of obesity by country.

velocity for at least the past decade at an age-standardized prevalence of 3–6% in Japan, France, Denmark and the Netherlands for one or both sexes. Elsewhere, obesity plateaued at higher endemic prevalences. For example, the velocity of obesity was indistinguishable from zero for at least the past 10 years at a PP of 0.80 among girls and boys in Kuwait and the USA, and among boys in New Zealand, where the prevalence of obesity was 19–25%, much higher than in the aforementioned countries in Western Europe (Figs. 2 and 3).

Contrasting with these plateaus and reversals, the prevalence of obesity in children and adolescents increased steadily or accelerated in most low-income and middle-income countries in Asia, Africa, Latin America, and Caribbean and Pacific Island nations (Figs. 1 and 2 and Extended Data Figs. 1 and 2). The velocity of obesity was higher in 2024 than in any other year since 1980 for girls in 110 of 200 countries and boys in 91 of 200 countries, the majority of which were in low-income and middle-income regions (Fig. 2). This steady or accelerating increase occurred both where prevalence is still low, such as in countries in East Africa (for example, Tanzania, Rwanda and Ethiopia) and South Asia (for example, Nepal and Bangladesh), and where prevalence has already increased to higher levels, in some Caribbean and Pacific Island

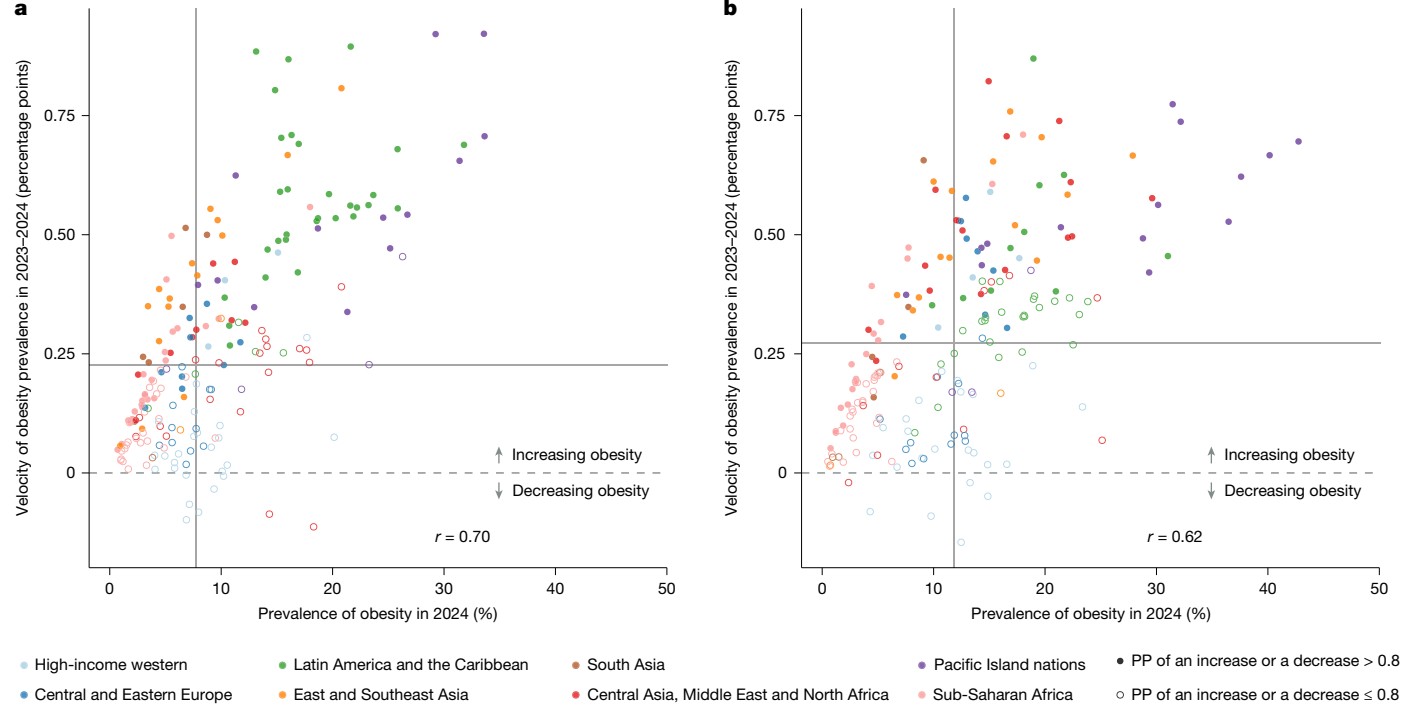

**Fig. 3 | Velocity and prevalence of obesity in 2024 for children and adolescents. a,b,** Velocity of obesity in relation to age-standardized prevalence of obesity in 2024 for girls (**a**) and boys (**b**). Each point shows one country, coloured by its super-region. The solid grey lines show the sex-specific median national prevalence and velocity of obesity in 2024.

nations (for example, Niue and the Bahamas) and some countries in the Middle East and North Africa (for example, Saudi Arabia, Qatar and Oman), Southeast Asia (for example, Brunei and Malaysia) and Latin America (for example, Chile).

There was a positive correlation between prevalence and velocity in 2024 (correlation coefficient = 0.70 for girls and 0.62 for boys), indicating that prevalence continues to grow in low-income and middle-income countries where it is already high. In 2024, obesity prevalence had a velocity of more than 0.5 percentage points per year with a PP > 0.80 in 36 countries for girls and in 35 countries for boys, with the highest velocities observed among girls in Tonga and Samoa and among boys in Peru (0.9 percentage points per year; Figs. 2 and 3 and Extended Data Figs. 3 and 4). Countries with velocity greater than 0.5 percentage points per year were in Latin America and the Caribbean, Pacific Island nations, some countries in South and Southeast Asia and sub-Saharan Africa for both sexes, and in East Asia and the Middle East and North Africa for boys. The only high-income western country with a velocity above this threshold was Finland for boys. The rise in obesity in all 36 countries in this group for girls and in 24 of the 35 countries for boys was classified as accelerating.

## Adults

From 1980 to 2024, age-standardized prevalence of obesity in women increased with a PP > 0.80 in 183 countries. The remaining 17 countries, where prevalence either did not change at a PP of at least 0.80 or decreased slightly with a PP > 0.80, were all in Europe. Among men, obesity increased in all countries with a PP > 0.80. In countries where prevalence increased with a PP > 0.80, the magnitude of the increase ranged from 2 to 43 percentage points for women, and from 1 to 36 percentage points for men. National obesity trajectories among adults had highly heterogeneous dynamics during this 45-year period, as was also the case for children and adolescents.

In most high-income western countries, the prevalence of obesity among adults was increasing with a PP > 0.80 in 1980 but the rise decelerated or plateaued around or after 2000, and may have even reversed slightly in some (Figs. 4 and 5 and Extended Data Figs. 5 and 6). The deceleration and plateauing among adults began later than in children and adolescents in most countries and typically occurred among women before men. By 2024, most of these countries exhibited small velocities indistinguishable from zero at a PP of 0.80 (Figs. 5 and 6 and Extended Data Figs. 7 and 8). In some countries (Spain and Italy for both sexes and France for women), velocity had become negative, that is, obesity was declining, with PP > 0.80; the negative velocities had magnitudes smaller than 0.5 percentage points per year. However, in some high-income countries, plateaus and reversals did not occur and the increase in obesity prevalence was either steady or accelerated, such as among both sexes in Finland, and among women in Norway and Belgium. Despite the acceleration, the velocity of obesity remained below 0.5 percentage points per year in these countries.

Slowdowns or plateaus of the rise in adult obesity for one or both sexes also happened in two other groups of countries. In the first group, obesity prevalence reached high levels, exceeding 40% in some cases, before a slowdown in the rise or plateau occurred. This group included some Caribbean (for example, the Bahamas) and Pacific (for example, American Samoa and Kiribati) Island nations and some countries in the Middle East and North Africa (for example, Kuwait, Jordan and United Arab Emirates). In the second group, obesity prevalence plateaued at lower levels than in the first group. This group included some countries in Central and Eastern Europe (for example, Poland and Estonia) and for men in sub-Saharan Africa (Cameroon, Ghana, Sierra Leone and South Africa). In addition, women in some Central and Eastern European countries (for example, Czechia and Russia) did not experience a rise in obesity throughout these 45 years at a PP of 0.80, and velocity remained below 0.25 percentage points per year. Unlike high-income countries, where the slowdown or plateau in adult obesity followed that of children and adolescents, most of these countries saw a slowdown or plateau of the rise in adult obesity before or in the absence of slowdown of the rise in obesity among children and adolescents (Figs. 2 and 5).

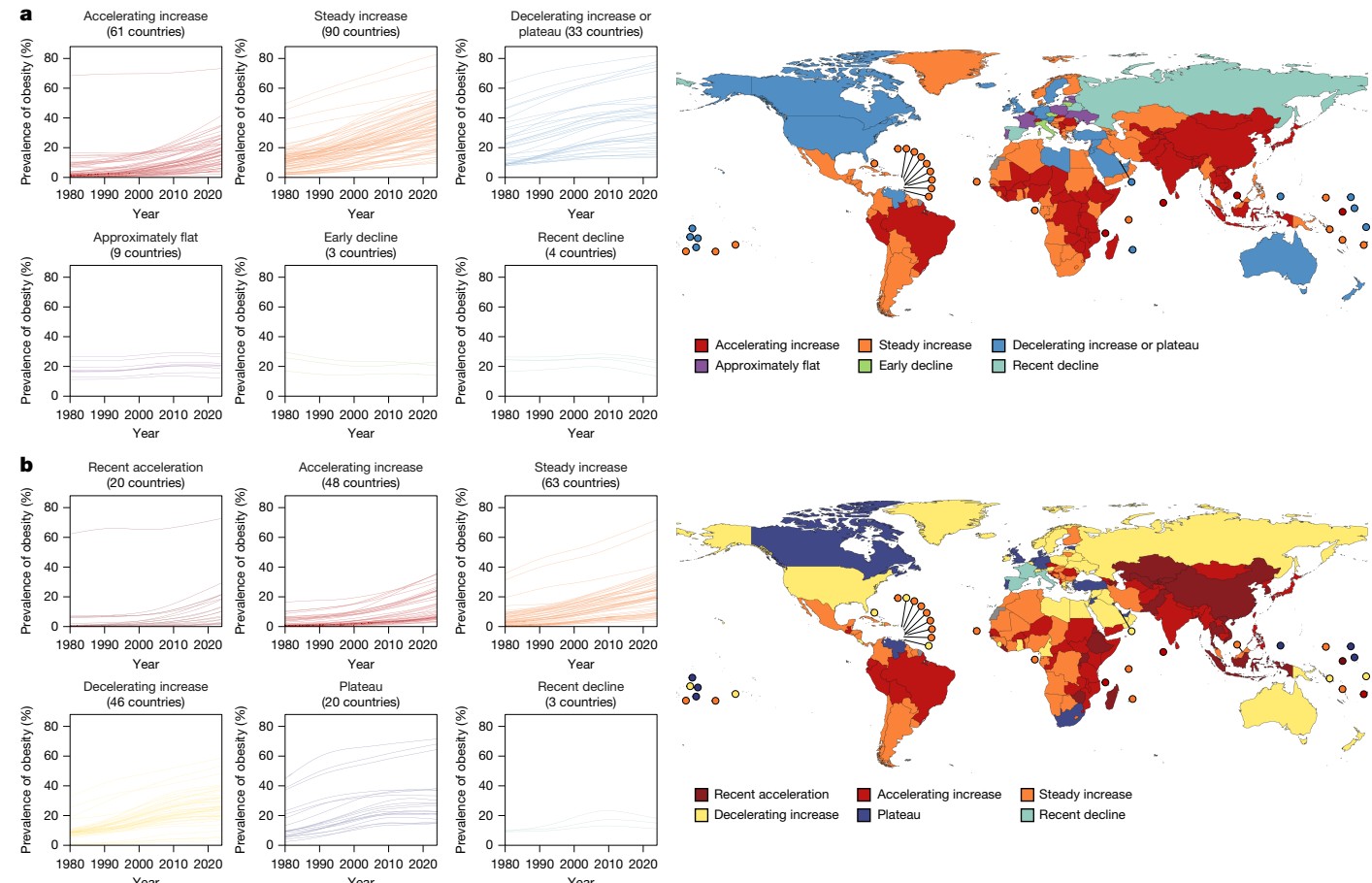

**Fig. 4 | Phenotypes of national obesity trajectories in adults. a,b,** Time-series plots of age-standardized obesity prevalence (left) and cluster allocation of these trends by country (right) for women (**a**) and men (**b**) from 1980 to 2024. Each panel represents a cluster of countries with similar shapes of obesity prevalence time series, and is labelled with its typology of trend. Each line on the plots represents obesity prevalence over time in one country. The maps display countries coloured according to their cluster allocation. See Extended Data Figs. 5 and 6 for allocation of countries in each super-region to clusters and Supplementary Fig. 4 for trends in age-standardized obesity prevalence by country.

The deceleration, plateauing and reversal of the rise in adult obesity, where it occurred, also happened at a wide range of prevalences. In high-income countries in Western Europe, the age-standardized prevalence of obesity in 2024 was typically below 25%, and as low as 11% in some countries (for example, France). By contrast, in high-income English-speaking countries such as the UK, Canada and the USA, prevalence in 2024 ranged from 25% to 43% (Fig. 6). Elsewhere, a deceleration or plateau occurred at even higher endemic levels. For example, age-standardized prevalence in 2024 ranged from 40% to 50% in some countries in the Middle East and North Africa, and from 50% to 80% in some Pacific Island nations where obesity decelerated or plateaued.

Contrasting with these decelerations, plateaus and reversals, the rise in adult obesity was steady or accelerated throughout these 45 years in the majority of low-income and middle-income countries in sub-Saharan Africa, Asia and Latin America, and in some Caribbean and Pacific Island nations. Many countries in Central Europe also experienced a steady or accelerating increase in adult obesity prevalence. The velocity of obesity was greater in 2024 than in any other year over the 45-year period for women in 84 of 200 countries and for men in 109 of 200 countries; these were predominantly low-income and middle-income countries (Fig. 5). This steady or accelerating rise occurred at a wide range of prevalences (Fig. 6). At the low end, obesity prevalence was still less than 10% but accelerating in parts of South and Southeast Asia and sub-Saharan Africa, where the burden of underweight was relatively high[1]. At the high end, prevalence surpassed 65% in women and men in some Pacific Island nations (for example, Tonga and Cook Islands) and was more than 35% in many other countries in the Middle East and North Africa and Latin America and the Caribbean, with a steady or accelerating increase in prevalence.

In 2024, obesity prevalence was rising in women in 100 countries and men in 66 countries with a velocity of more than 0.5 percentage points per year and a PP > 0.80 (Figs. 5 and 6 and Extended Data Figs. 7 and 8), a larger number of countries than that crossing the same threshold for children and adolescents. The only high-income western country in this group was Finland for women. The other countries were in various low-income and middle-income regions, with sub-Saharan Africa represented more for women than men, and the opposite for Central and Eastern Europe. In 41 of these countries for women and in 29 for men, the trend in prevalence was classified as accelerating. Nonetheless, the correlation coefficient between prevalence and velocity was only 0.24 for women and 0.27 for men across all countries, smaller than for children and adolescents (Fig. 6). These weaker correlations are a result of the aforementioned dynamics, including that in some countries with high prevalence, the rise in prevalence has slowed down (for example, men in some Pacific Island nations), whereas in some with low prevalence, it is rapidly increasing (for example, in some countries in South and Southeast Asia for both sexes and in some countries in sub-Saharan Africa for women). The velocity of obesity in 2024 was more than 1.0 percentage point per year in 11 countries for one or both sexes; these were predominantly low-income and middle-income countries. Velocity in 2024 was higher among men than among women in Central and Eastern Europe, Central Asia and East Asia, whereas the

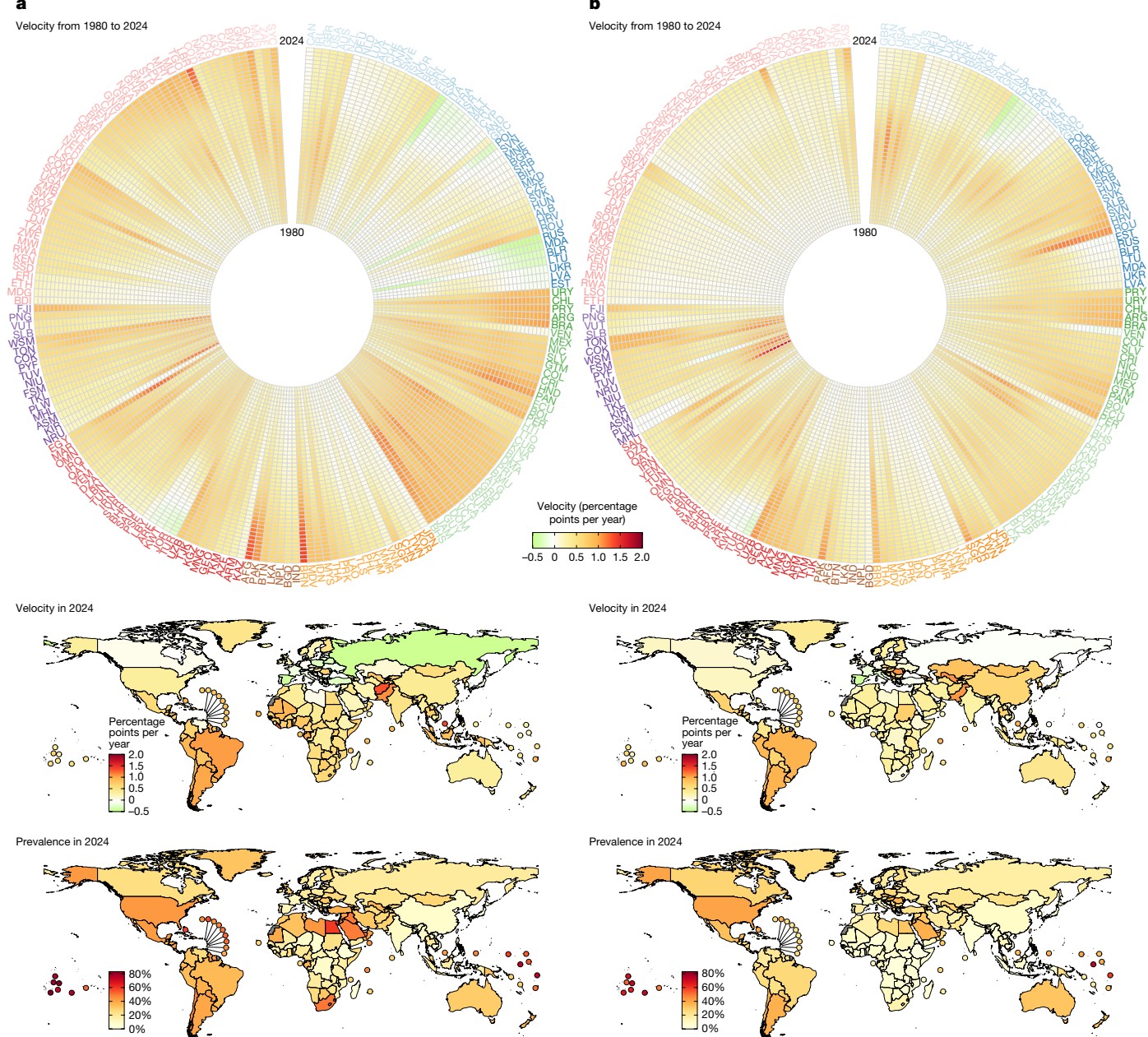

**Fig. 5 | Velocity and prevalence of obesity in adults. a,b**, Velocity and prevalence of obesity in women (**a**) and men (**b**). The circular wheel plots show the year-on-year velocity of obesity from 1980 to 2024 by country. Each cell represents the velocity in one year. A positive velocity (red) indicates a year-on-year increase in obesity prevalence, whereas a negative (green) velocity indicates a year-on-year decrease. Years when no change in obesity was observed are coloured in white. Countries are labelled by their ISO 3166-1 alpha-3 codes (Supplementary Note 1) and coloured by super-region. Countries are ordered by increasing 2024 velocity within each region. The top maps show the velocity of obesity in 2024 in each country, following the same colour scheme as the velocity wheel plots, and the bottom maps show the age-standardized prevalence of obesity in 2024. See Extended Data Fig. 7 for velocity of obesity in 2024 and its uncertainty by country, Extended Data Fig. 8 for a map of PP that velocity of obesity was positive in 2024, Supplementary Fig. 4 for trends in the prevalence of obesity by country and Supplementary Fig. 6 for trends in the velocity of obesity by country.

opposite was true in South and Southeast Asia, the Caribbean and most of sub-Saharan Africa. It was similar between women and men in high-income western countries, Latin America, the Middle East and Pacific Island nations (Extended Data Fig. 9).

## Strengths and limitations

Our study has strengths related to its scope, data and methods. We conducted an analysis and presentation of trends in obesity that went beyond the traditional narrative of long-term increase and systematically quantified highly heterogeneous temporal dynamics. We used a large amount of population-based data, from countries covering more than 99% of the population of the world. We maintained a high standard of data quality through repeated checks of the study sample and characteristics, and did not use self-reported data to avoid bias. Data were analysed according to a consistent protocol. We used a statistical model that accounted for the age patterns of BMI during childhood, adolescence and adulthood. We used all available data while giving more weight to national data than to subnational and community data.

As with all global analyses, our study has limitations. Some countries had fewer data and 3 of the 200 countries (Bermuda, Djibouti and North Korea) had none; their estimates were informed to a greater

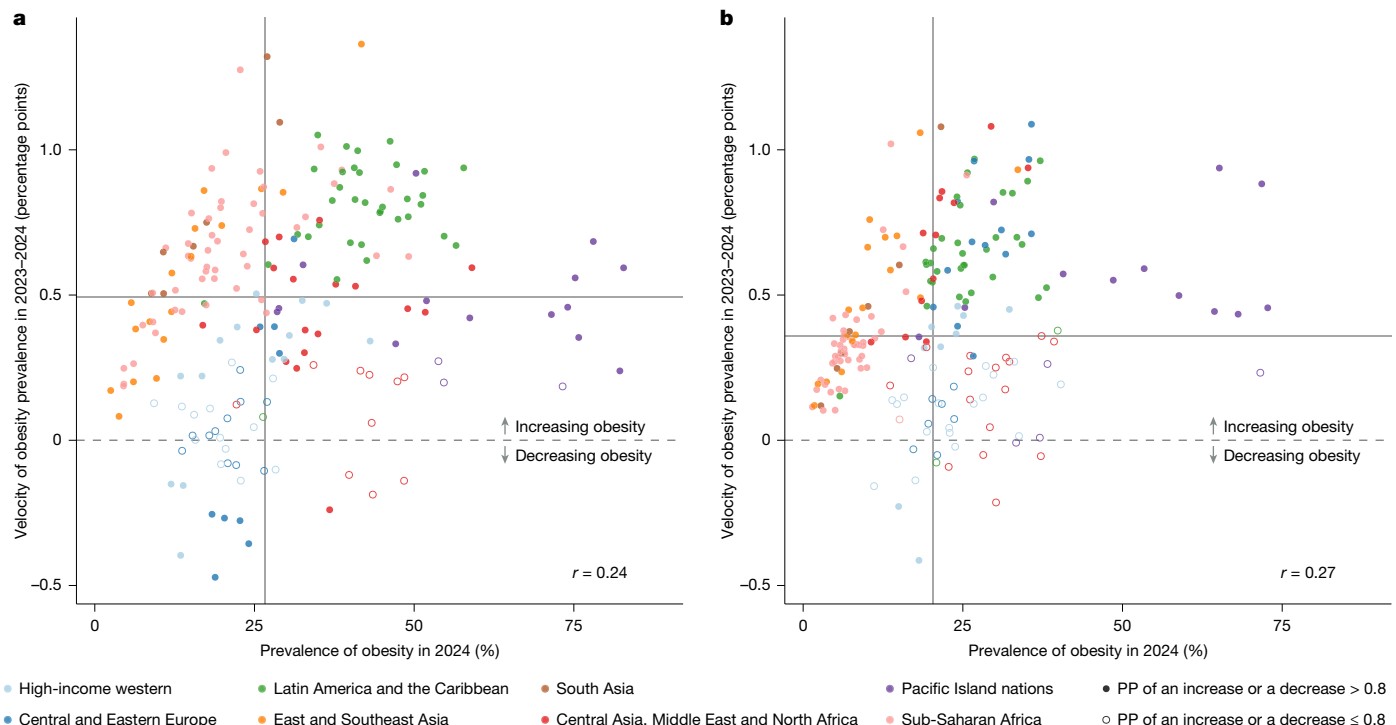

**Fig. 6 | Velocity and prevalence of obesity in 2024 for adults. a,b,** Velocity of obesity in relation to age-standardized prevalence of obesity in 2024 for women (**a**) and men (**b**). Each point shows one country, coloured by super-region. The solid grey lines show the sex-specific median national prevalence and velocity of obesity in 2024.

extent by data from other countries, especially those in their respective regions, through a data-driven hierarchy, which was based on geography and epidemiology. Although the cross-validation analysis shows that the estimates for countries without data had relatively small errors, alternative hierarchical arrangement of countries is possible. There were also differences in data availability by time period, with less data available in the first decade of the analysis in some regions. There are other approaches for clustering besides *k*-means, including some that provide probabilistic allocation to clusters. Given our purpose of general classification of national obesity trajectories (not allocation of resources or policies), we used *k*-means, which produces non-overlapping clusters.

## Implications for obesity prevention

Our results demonstrate that generalizing the trends in obesity as a global epidemic masks highly heterogeneous temporal dynamics across countries and in different age groups. In some high-income western countries, the velocity of obesity in children and adolescents began to slow down as early as around 1990 with the rise coming to a halt by the mid-2000s, and with some indications of a subsequent decline. A similar slowdown and possible reversals followed in adults in these countries and could also be observed in Central and Eastern Europe. The plateauing of the trends in these countries has created a state of endemicity at wide range of prevalences. By contrast, the rise in obesity accelerated in many developing regions among children and adolescents and among adults. In many countries in these regions, the rise continues to accelerate at prevalences that are higher than those in most high-income countries.

The results of our comparative global analysis show that the direction and pace of change in obesity vary over time in the same country, and cover the spectrum from steady or accelerating rise, to a slowdown or plateau in rise, or even decline. This feature was qualitatively presented in the obesity transition framework[14]. Here we quantified these dynamics and their similarities and variations across the globe using a vast amount of data. Of note, we found indications that a small decline in

obesity in national populations may have begun, especially for women, which was rare when the original transition framework was formulated[14]. In Italy and Portugal, obesity had a negative velocity in children, adolescents and adults of both sexes in 2024, although some of these were not statistically distinguishable from zero. If these declines persist with additional years of data, they will confirm the feasibility of the fourth stage of the obesity transition, which envisioned a decline in prevalence. Our analysis revealed and quantified three further features of worldwide obesity dynamics that were not formally included in the obesity transition framework. First, countries, both across and within regions, vary in whether and when the increase in obesity accelerated, decelerated, plateaued or reversed. This diversity in shape and timing of change led to the emergence of distinct phenotypes of national obesity trends, for example, accelerating, steady or decelerating increase or recent acceleration, beyond the general concept of increase in stages 1 and 2 of transition, as presented in the obesity transition framework[14]. For example, countries in Eastern Europe and Latin America with similar economic development had distinct trajectories, with many countries in Eastern Europe having started to plateau, whereas most countries in Latin America are still experiencing a steady or accelerating rise in obesity. As an example of within-region diversity, in the Middle East and North Africa, obesity plateaued in Kuwait but continues to rise in Iran, Oman and Saudi Arabia. Second, for any trajectory phenotype, there is substantial variation in the prevalence of obesity both across and within regions. For example, obesity plateaued at a much higher prevalence in high-income English-speaking countries than in those in continental Western Europe for both children and adolescents and adults. Within continental Western Europe, obesity prevalence has stabilized at 11–23% for adults and 4–15% for children and adolescents; in high-income English-speaking countries, it reached 25–43% for adults and 7–23% for children and adolescents. For countries with an accelerating increase in obesity, prevalence among adults in 2024 was less than 5% in East African countries (for example, Ethiopia and Rwanda), but reached 30–40% in some countries in Central Europe (for example, Romania and Czechia) and Latin America (for example, Brazil). Within a region, for example, East and Southeast Asia, countries with

an accelerating increase in obesity had a prevalence in 2024 ranging from 2–3% in Timor-Leste and Vietnam to 20–40% in Thailand and Brunei. Third, the above characteristics can vary between children and adolescents and adults, and between sexes, further distinguishing regions and countries. For example, women in both Central Asia and Latin America had a steady or accelerating increase in obesity, whereas trends among girls were largely flat in Central Asia but had an accelerating increase in Latin America. Of note, we demonstrated earlier plateauing of the rise in obesity prevalence in children and adolescents than in adults in high-income countries, whereas in most other regions, deceleration and plateauing of the rise in prevalence occurred in adults before they did in children and adolescents or while obesity continued to increase in children and adolescents. In terms of sexes, the velocity of obesity was typically higher in men than in women in Central and Eastern Europe, whereas in South and Southeast Asia and most of sub-Saharan Africa obesity had higher velocity in women than in men (Extended Data Fig. 9). Similarly, the rise in obesity plateaued or decelerated for both sexes in most high-income western countries, but this phenomenon was limited to women in Central Europe.

These heterogeneities collectively demonstrate that obesity dynamics and trajectories differ across countries that are similar in their economy (for example, national income; market-driven versus managed economy; and agricultural, manufacturing or service economy), environment (for example, extent of urbanization) and technology (for example, extent of mechanization, motorization, electrification and penetration of information technology). Explanations of the rise in obesity have typically been based on generalizations about the so-called shared triggers and global drivers, such as availability and marketing of certain foods and energy expenditure of physical activity at work, transport and leisure, and framed as consequences of macro trends such as urbanization. Although these factors may be relevant, they alone do not explain the heterogeneities that we uncovered, which suggests that their roles are modified or even countered by other social, economic and policy factors. These may include cultural factors and social norms and roles that influence what and how much children, adolescents and adults eat at home and in social settings[15–17], how much they participate in sports, play and active commuting[18,19], and social norms and perceptions related to body image and the discordance between ideal, perceived and actual body weight[20–22]. They also include levels and distributions of income and education that affect food choices and participation in sports, either through access and affordability or the ability to use information about the nutritional value and potential harms of specific foods[23–26]. Finally, some of these heterogeneities may reflect differences across countries in technological, institutional and community characteristics that influence food, sports and other forms of physical activity, such as healthy school meals and, in some countries, sports and physical education programmes at schools and community centres[27]. Identifying the role of these phenomena, and their complex interactions, can inform programmes and policies that curb or reverse the rise in obesity. Doing so requires detailed data on food and physical activity, and the related technological, economic, social and cultural factors, including their distributions within each country, and methods to estimate their contributions to change. However, such data are even more scarce than data on obesity. As a result, understanding the complex drivers of change and drawing lessons on good practice has to use a combination of multi-country analyses and in-depth case studies in specific countries, especially those where obesity did not increase, plateaued at low levels or plateaued early (for example, Denmark, France and Spain for both children and adolescents and adults; and Kazakhstan, Japan and Taiwan for children and adolescents), as has been done for growth and nutrition in children[28,29]. There may also be an as-yet-unmeasured effect from variations in a broader group of factors such as sleep duration and patterns, stress and other psychosocial factors, and environmental exposures such as endocrine-disrupting chemicals. Furthermore, these environmental and nutritional factors interact with genetics and possibly with the phenotypic characteristics that arise from fetal and early-life nutrition[30]. Although recent medications are efficacious in terms of weight loss, their effects on trends presented here are likely to be small given relatively low coverage in most countries during the period of our analysis[31]; these medicines may nonetheless have an increasingly important influence on future trends.

The distinction in trends between high-income countries and low-income and middle-income countries, in relation to the shape of the trajectories and the timing of acceleration, plateau and deceleration at different ages, may be because in high-income countries, the rise in obesity resulted from gradual economic and technological trends that affected food availability, composition and cost[5–7,32]. This gradual change coincided with expanding opportunities for healthier diets among those with higher income and education[25,26], which contributed to a subsequent deceleration in obesity in these populations. The predominant policy response to obesity has been provision of information about its harms, and encouraging prevention through nutrition and physical activity, which became more common in the 2000s[33]. A side effect of such a response, which relies on individuals' use of information and advice without enhancing access to and affordability of healthy foods, has been increasing inequality in obesity in these countries[23–26,34–39]. There have been few policies and programmes that have attempted to systematically change nutrition and physical activity[40,41], especially in those with lower education and income where obesity prevalence is higher, beyond demonstration projects[42], which have not been scaled and sustained nationally. There is insufficient evidence to ascertain whether specific policies and programmes have curbed the rise in obesity, or even improved nutrition and physical activity, in real-world settings[43–48], with the possible exception of sugar-sweetened beverage taxes, which have had a measurable, albeit small, effect on obesity in multiple places[49–53]. These taxes have only been recently implemented hence do not account for the earlier plateaus.

In low-income and middle-income countries, despite the heterogeneities in the shape and pace of increase, the rise in obesity continues unabated both where prevalence is still low and where prevalence has already surpassed those of high-income countries, especially for children and adolescents as seen by the correlation between velocity and prevalence. In these settings, the mechanization of work and transport[54], improved purchasing power and food trade and commercialization[4,55] brought many health and nutritional benefits, including gains in height[56], while also leading to a rise in obesity because the public health system and household consumption practices[11,57] were primarily focused on addressing undernutrition. Furthermore, until recently, there was no or limited fiscal or regulatory response to aggressive marketing of items such as sugar-sweetened beverages that have no nutritional benefit and worsen obesity. The food production, supply and distribution infrastructure, and in many cases income[58], remain insufficient for regular availability and purchase of healthy foods such as fresh fruits and vegetables or unprocessed whole-grain items. This is particularly the case in many island nations where rapid changes in diet followed changes from locally sourced to imported processed foods[59].

Our results, and the potential reasons for these dynamics and their heterogeneities, highlight both opportunities and challenges for curbing the rise in obesity across the world. The numerous cases of plateau and even reversal in obesity show that the rise can be contained even at low levels. At the same time, the wide variation in whether, when and at what prevalence the plateau occurred shows that stopping and reversing the rise may be harder in some places owing to the diverse factors that affect what and how much people eat. Some of these factors go beyond those commonly stated as drivers of obesity, and may include social norms, the food culture, and levels and inequalities in education, income, food infrastructure and access, the living environment and health care. In the future, weight loss medications also provide an additional route for addressing obesity, but their highly variable costs through public

and private providers are currently an obstacle to increasing their coverage and may increase inequalities. The combination of the opportunities and challenges demonstrates that what is needed is nuanced nutritional and health policies and programmes that are relevant for each country, especially those that support people with lower income and education towards eating healthy foods, having an active lifestyle and using relevant health care interventions to attain and maintain health, functional capacity and quality of life across the life course.

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

NCD Risk Factor Collaboration (NCD-RisC)

Bin Zhou[1,890], Nowell H. Phelps[2,890], Agnese Galeazzi[1,890], Olivia N. O'Driscoll[1,890], James E. Bennett[1], Lakshya Jain[1], Ysé D'Ailhaud De Brisis[1], Ana Barradas-Pires[1], Fulvio Deo[1], Gretchen A. Stevens[3], Vasilis Kontis[1], Christopher J. Paciorek[4], Rodrigo M. Carrillo-Larco[5], Anu Mishra[6], Yefeng Fan[1], Andrea Rodriguez-Martinez[1], Vishwa Nath[1], Archie W. Rayner[1], Annalise Zouein[1], Natalie R. Evans[7], Jennifer L. Baker[8], Günther Fink[9,10], Maroje Sorić[11,12], Carlos A. Aguilar-Salinas[13], Ranjit Mohan Anjana[14], Zulfiqar A. Bhutta[15,16], Pascal Bovet[17,18], Goodarz Danaei[19], Kairat Davletov[20], Shubash Ganapathy[21], Edward W. Gregg[1,22], Nayu Ikeda[23], Andre Pascal Kengne[24], Young-Ho Khang[25], Kamlesh Khunti[26], Tiina Laatikainen[27,28], Avula Laxmaiah[29], Lee-Ling Lim[30], Hsien-Ho Lin[31], Jean Claude N. Mbanya[32], J. Jaime Miranda[33], Anja Schienkiewitz[34], Sylvain Sebert[35], Namuna Shrestha[36], Yi Song[37], Luis Adrián Soto-Mota[13], Limin Wang[38], Novie O. Younger-Coleman[39], Francesco Zaccardi[26], Julie Aarestrup[8], Leandra Abarca-Gómez[40], Mohsen Abbasi-Kangevari[41], Ziad A. Abdeen[42], Shynar Abdrakhmanova[43], Suhaila Abdul Ghaffar[44], Hanan F. Abdul Rahim[45], Zulfiya Abdurahmnova[46], Niveen M. Abu-Rmeileh[47], Jamila Abubakar Gaye[48], Benjamin Acosta-Cazares[49], Ishag Adam[50], Marzena Adamczyk[51], Robert J. Adams[52], Seth Adu-Afarwuah[53], Wichai Aekplakorn[54], Tin Afifah[55], Kaosar Afsana[56], Shoaib Afzal[57,58], Imelda A. Agdeppa[59], Javad Aghazadeh-Attari[60], Åsa Ågren[61], Hassan Aguenaou[62], Charles Agyemang[63], Mohamad Hasnan Ahmad[44], Noor Ani Ahmad[44], Ali Ahmadi[64], Naser Ahmadi[41], Nastaran Ahmadi[65], Imran Ahmed[66], Soheir H. Ahmed[66], Wolfgang Ahrens[67], Gulmira Aitmurzaeva[68], Kamel Ajlouni[69], Dilorom Akhmedova[70], Nilufar Akhmedova[70], Nasser Al-Daghri[71], Sarah F. Al-Hamli[72], Hazzaa M. Al-Hazzaa[73], Halima Al-Hinai[74], Jawad A. Al-Lawati[74], Rajaa Al-Raddadi[75], Islam K. Al-Shami[76], Deena A. Al Asfoor[77], Huda M. Al Hourani[76], Nawal M. Al Qaoud[78], Monira Alarouj[79], Fadia AlBuhairan[80], Shahla AlDhukair[81], Maryam A. Aldwairji[76], Sílvia Alexius[82], Mohamed M. Ali[3], Mohammed K. Ali[5], Nazirah Alias[44], Anna V. Alieva[83], Abdullah Alkandari[79], Buthaina M. Alkhatib[76], Mar Alvarez-Pedrerol[84,85], Eman Aly[77], Deepak N. Amarapurkar[86], Parisa Amiri[87], John Amoah[88], Norbert Amougou[89], Philippe Amouyel[90,91], Atieh Amouzegar[92], Lars Bo Andersen[93], Sigmund A. Anderssen[94], Jonas S. Andersson[61], Odysseas Androutsos[95], Malick Anne[96], Alireza Ansari-Moghaddam[97], Elena Anufrieva[98], Hajer Aounallah-Skhiri[99], Joana Araújo[100], Inger Ariansen[101], Carmen Arias López[102], Tahir Aris[44], Raphael E. Arku[103], Nimmathota Arlappa[29], Enrique G. Artero[104], Krishna K. Aryal[105,106], Thor Aspelund[107], Felix K. Assah[32], Nega Assefa[108], Batyrbek Assembekov[20], Maria Cecília F. Assunção[109], Shiu Lun Au Yeung[110], May Soe Aung[111], Juha Auvinen[35,112], Mária Avdičová[113], Kishwar Azad[114], Ana Azevedo[100], Mohsen Azimi-Nezhad[115], Fereidoun Azizi[116], Bontha V. Babu[117], Flora Bacopoulou[118], Azli Baharudin[44], Suhad Bahijri[75], Borko Bajić[119], Izet Bajramovic[120], Marta Bakacs[121], Nagalla Balakrishna[29], Yulia Balanova[122], Mohamed Bamoshmoosh[123], Maciej Banach[124], José R. Banegas[84,125], Joanna Baran[126], Rafał Baran[126], Carlo M. Barbagallo[127], Valter Barbosa Filho[128], Alberto Barcelo[129], Maja Baretić[11], Amina Barkat[131], Joaquin Barnoya[132], Lena Barrera[133], Marta Barreto[134,135], Aluisio J. D. Barros[109], Mauro Virgílio Gomes Barros[136], Anna Bartosiewicz[126], Alicja Basiak-Rasała[137], Abdul Basit[138], Joao Luiz D. Bastos[139], Iqbal Bata[140], Anwar M. Batieha[141], Aline P. Batista[142], Rosangela L. Batista[143], Zhamilya Battakova[144], Susanne Bauer[145], Louise A. Baur[33], Pascal M. Bayauli[146], Robert Beaglehole[147], Silvia Bel-Serrat[148], Antonisamy Belavendra[149], María R. Beltran-Valls[150], Habiba Ben Romdhane[151], Theodora Benedek[152], Judith Benedics[153], Mikhail Benet[154], Gilda Estela Benitez Rolandi[155], Pedro J. Benito[156], Michaela Benzeval[7], Elling Bere[157], Nicolas Berger[158], Ingunn Holden Bergh[101], Yemane Berhane[159], Salim Berkinbayev[160], Antonio Bernabe-Ortiz[161], Gailute Bernotiene[162], Ximena Berrios Carrasola[163], Paula Berruezo[164,165], Heloísa Bettiol[166], Manfred E. Beutel[167], Augustin F. Beybey[32], Jorge Bezerra[136], Aroor Bhagyalaxmi[168], Sumit Bharadwaj[169], Santosh K. Bhargava[170], Hongsheng Bi[171], Yufang Bi[172], Daniel Bia[173], Katia Biasch[174], Elysée Claude Bika Lele[175], Mukharram M. Bikbov[176], Bihungum Bista[177], Dusko J. Bjelica[178], Anne A. Bjerregaard[8], Peter Bjerregaard[179], Espen Bjertness[180], Marius B. Bjertness[180], Cecilia Björkelund[181], Ieva Blauzde[162], Moran Blaychfeld Magnazi[182], Katia V. Bloch[183], Anneke Blokstra[184], Simona Bo[185], Martin Bobak[186], Lynne M. Boddy[187], Bernhard O. Boehm[188], Jose G. Boggia[173], Elena Bogova[189], Carlos P. Boissonnet[190], Stig E. Bojesen[57,58], Marialaura Bonaccio[191], Americo Bonanni[191], Vanina Bongard[192], Alice Bonilla-Vargas[40], Matthias Bopp[193], Elaine Borghi[3], Herman Borghs[194], Steve Botomba[195], Rupert Bourne[196], Khadichamo Boymatova[197], Francesca Bracone[191], Lien Braeckevelt[198], Lutgart Braeckman[199], Marjolijn C. E. Bragt[200], Tasanee Braithwaite[201], Imperia Brajkovich[202], Francesco Branca[203], Juergen Breckenkamp[204], João Breda[205], Hermann Brenner[206], Lizzy M. Brewster[207], Garry R. Brian[208], Yajaira Briceño[209], Lăcrămioara Brîndușe[210], Bettina Bringolf-Isler[9,10], Miguel Brito[211], Sinead Brophy[212], Johannes Brug[184], Anna Bugge[213], Marta Buoncristiano[214], Genc Burazeri[215], Con Burns[216], Antonio Cabrera de León[217], Joseph Cacciottolo[218], Cristina Cadenas-Sanchez[219], Hui Cai[220], Roberta B. Caixeta[221], Tilema Cama[222], Christine Cameron[223], José Camolas[224], Francesco Campa[225], Günay Can[226], Ana Paula C. Cândido[227], Felicia Cañete[155], Mario V. Capanzana[59], Naděžda Čapková[228], Eduardo Capuano[229], Rocco Capuano[229], Vincenzo Capuano[229], Marloes Cardol[230], Viviane C. Cardoso[166], Axel C. Carlsson[37], Esteban Carmuega[232], Joana Carvalho[100], Deborah Carvalho Malta[233], José A. Casajús[234], Felipe F. Casanueva[235], Jose Castro-Piñero[236], Ertugrul Celikcan[237], Laura Censi[238], Chiara Cerletti[191], Marvin Cervantes-Loaiza[40], Juraci A. Cesar[239], Parinya Chamnan[240], Snehalatha Chamukuttan[241], Angelique W. Chan[242], Queenie Chan[5], Fadi J. Charchar[243], Marie-Aline Charles[244], Himanshu K. Chaturvedi[245], Nish Chaturvedi[186], Norsyamlina Che Abdul Rahim[44], Fangfang Chen[246], Huashuai Chen[247], Long-Sheng Chen[248], Shuohua Chen[249], Zhengming Chen[250], Ching-Yu Cheng[251], Yiling J. Cheng[252], Leila Cheraghi[87], Bahman Cheraghian[253], Angela Chetrit[254], Ekaterina Chikova-Iscener[255], Mai J. M. Chinapaw[3], Arnaud Chiolero[256], Adela Chirita-Emandi[257], María-Dolores Chirlaque[84], Chean Lin Chong[258], Kaare Christensen[259], Diego G. Christofaro[260], Jerzy Chudek[261], Silvia Ciardullo[262], Renata Cifkova[263,264], Michelle Cilia[265], Eliza Cinteza[210], Massimo Cirillo[266], Frank Claessens[194], Maria Clapperton[267], Philip Clare[33], Janine Clarke[268], Svetlana Cociu[269], Emmanuel Cohen[89], Sandra Colorado-Yohar[84,270], Laura-María Compañ-Gabucio[84,270], Hans Concin[145], Susana C. Confortin[271], Cyrus Cooper[272], Tara C. Coppinger[216], Lorraine S. Cordeiro[103], Eva Corpeleijn[230], Lilia Yadira Cortés[273], Cojocaru R. Cosmin[210],

Simona Costanzo[191,274], Melanie J. Cowan[3], Chris Cowell[33], Cora L. Craig[223], Amelia C. Crampin[275], Haddy Crookes[276], Amanda J. Cross[1], Sarah Crozier[272], Ana B. Crujeiras[277], Juan J. Cruz[84,125], Tamás Csányi[278], Semánová Csilla[279], Alexandra M. Cucu[210,280], Liufu Cui[249], Felipe V. Cureau[281], Sarah Cuschieri[218], Ewelina Czenczek-Lewandowska[126], Graziella D'Arrigo[282], Eleonora d'Orsi[283], Haroldo da Silva-Ferreira[284], Alanna G. da Silva[233], Liliana Dacica[285], Tukur Dahiru[286], Christina C. Dahm[287], María Ángeles Dal Re Saavedra[102], Jean Dallongeville[288], Albertino Damasceno[289], Camilla T. Damsgaard[58], Maryam S. Daneshpour[116], Rachel Dankner[254], Parasmani Dasgupta[290], Saeed Dastgiri[291], Luc Dauchet[90,91], Afonso de Almeida[292], Francisco de Assis Guedes de Vasconcelos[283], Maria Alice Altenburg de Assis[283], Guy De Backer[199], Dirk De Bacquer[199], Jaco De Bacquer[199], Jeroen de Bont[293], Luigi G. De Filippis[294], Patrícia de Fragas Hinnig[283], Stefaan De Henauw[199], Pilar De Miguel-Etayo[277,295], Jan-Walter De Neve[296], Paula Duarte de Oliveira[109], Luz Maria De Regil[3], David De Ridder[203], Karin De Ridder[158], Susanne R. de Rooij[207], Ana Carolina M. G. N. de Sá[233], Delphine De Smedt[199], Thomas V. de Souza[142], Marco A. de Valois Correia Júnior[136], George Dedoussis[297], Mohan Deepa[14], Alexander D. Deev[122], Vincent DeGennaro Jr[298], Francis Delpeuch[299], Stefan Demarest[158], Elaine Dennison[272], Katarzyna Dereń[126], Valérie Deschamps[300], Ruslan D. Devrishov[301], Meghnath Dhimal[177], Juvenal Soares Dias-da-Costa[302], Alejandro Diaz[303], Francisco Diez-Canseco[161], Zivka Dika[11], Shirin Djalalinia[304], Visnja Djordjic[305], Ha T. P. Do[306], Annette J. Dobson[307], Liria Domínguez[308], Maria Benedetta Donati[191], Chiara Donfrancesco[262], Guanghui Dong[309], Li Dong[310], Yanhui Dong[37], Silvana P. Donoso[311], Cecilia Dorado-García[312], Angela Döring[313], Maria Dorobantu[210,314], Ahmad Reza Dorosty[65], Marcus Dörr[315], Nico Dragano[316], Wojciech Drygas[317,318], Shufa Du[319], Jia Li Duan[320], Charmaine A. Duante[59], Priscilla Duboz[321], Marina Duishenkulova[44], Vesselka L. Duleva[255], Virginija Dulskiene[162], Samuel C. Dumith[239], Anar Dushpanova[322,323], Terence Dwyer[250,324], Azhar Dyussupova[325], Vilnis Dzerve[326], Elzbieta Dziankowska-Zaborszczyk[317], Anna Dzielska[327], Narges Ebrahimi[41], Guadalupe Echeverría[163], Ricky Eddie[328], Ebrahim Eftekhar[329], Vasiliki Efthymiou[118], Eruke E. Egbagbe[330], Robert Eggertsen[181], Sareh Eghtesad[65], Gabriele Eiben[331], Ulf Ekelund[94], Mohammad El-Khateeb[69], Laila El Ammari[332], Jalila El Ati[333], Denise Eldemire-Shearer[39], Paul Elliott[1], Ofem Enang[334], Ronit Endevelt[182,335], Reina Engle-Stone[336], Jonas Englund[337], Rajiv T. Erasmus[338], Cihangir Erem[339], Gul Ergor[340], Louise Eriksen[179], Johan G. Eriksson[341], Jorge Escobedo-de la Peña[49], Ali Esmaeili[342], Vanesa España-Romero[236], Alun Evans[343], Roger G. Evans[344], David Faeh[35,112], Guy Fagherazzi[345], Noushin Fahimfar[65], Ildar Fakhradiyev[160], Albina A. Fakhretdinova[176], Caroline H. Fall[272], Elnaz Faramarzi[346], Mojtaba Farjam[347], Victoria Farrugia Sant'Angelo[265], Farshad Farzadfar[3], Yosef Farzi[41], Mohammad Reza Fattahi[348], Asher Fawwad[349], Wafaie W. Fawzi[19], Rosemarie Felder-Puig[350], Francisco J. Félix-Redondo[351], Trevor S. Ferguson[39], Romulo A. Fernandes[260], Daniel Fernández-Bergés[352], Desha R. Fernando[353], Rashida A. Ferrand[354], Daniel Ferrante[355], Thomas Ferrao[268], Gerson Ferrari[356], Marika Ferrari[238], Marco M. Ferrario[274], Catterina Ferreccio[163], Eldridge Ferrer[59], Jean Ferrieres[192], Thamara Hubler Figueiró[283], Anna Fijalkowska[327], Mauro Fisberg[357], Krista Fischer[358], Malang N. Fofana[276], Maria Forsner[61], Edward F. Fottrell[186], Heba M. Fouad[77], Damian K. Francis[359], Maria do Carmo Franco[360], Zlatko Fras[361], Brooklyn Fraser[362], Guillermo Frontera[363], Flavio D. Fuchs[364], Sandra C. Fuchs[364], Yuki Fujita[366], Matsuda Fumihiko[367], Viktoriya Furdela[368], Takuro Furusawa[367], Zbigniew Gaciong[369], Lutfi Gafarov[370], Mihai Gafencu[257], Sonya V. Galcheva[371], Henrike Galenkamp[207], Daniela Galeone[372], Myriam Galfo[238], Fabio Galvano[373], Jingli Gao[249], Chandalene Garabwan[374], Natalia García-Corada[375], Manoli Garcia-de-la-Hera[84,270], Marta García Solano[102], Dickman Gareta[376], Sarah P. Garnett[33], Jean-Michel Gaspoz[377], Magda Gasull[84,378], Victoria Gauthier[90,91], Adroaldo Cesar Araujo Gaya[365,891], Anelise Reis Gaya[109], Andrea Gazzinelli[233], Ulrike Gehring[379], Johanna M. Geleijnse[380], Ronnie George[381], Eva Gerdts[106], Ibrahim D. Gezawa[382], Ebrahim Ghaderi[383], Seyyed-Hadi Ghamari[41], Ali Ghanbari[41], Asghar Ghasemi[384], Erfan Ghasemi[41], Hala Ghattas[385,386], Oana-Florentina Gheorghe-Fronea[210], Simona Giampaoli[262], Francesco Gianfagna[274], Christian Gieger[313], Tiffany K. Gill[387], Ntombifuthi Ginindza[388], Jonathan Giovannelli[90,91], Glen Gironella[59], Aleksander Giwercman[389], Konstantinos Gkiouras[390], Natalya Glushkova[20,323], Ramesh Godara[391], Keith M. Godfrey[272], Justyna Godos[373], Sibel Gogen[237], Marcel Goldberg[244,392], David Goltzman[393], Georgina Gómez[394], Luis F. Gomez[273], Santiago F. Gómez[395,396], Aleksandra Gomula[397], Bruna Gonçalves Cordeiro da Silva[109], Helen Gonçalves[109], Mauer Gonçalves[398], Ana D. González-Alvarez[399], David A. Gonzalez-Chica[387], Esther M. González-Gil[234], Marcela Gonzalez-Gross[156], Margot González-Leon[49], Juan P. González-Rivas[400], Angel R. Gonzalez[401], Frederic Gottrand[90], Antonio Pedro Graça[402], Dušan Grafnetter[403], Aneta Grajda[404], Maria G. Grammatikopoulou[405], Andriene Grant[406], Ronald D. Gregor[140], Maria João Gregório[402], Anne Sameline Grimsgaard[407], Else Karin Grøholt[101], Anders Grøntved[259], Giuseppe Grosso[373], Dongfeng Gu[408], Viviana Guajardo[409], Emanuela Gualdi-Russo[410], Pilar Guallar-Castillón[84,125], Elias F. Gudmundsson[411], Vilmundur Gudnason[107], Maëlenn Guerchet[412], Ramiro Guerrero[413], Idris Guessous[377], Andre L. Guimaraes[414], Unjali P. Gujral[5], Martin C. Gulliford[415], Johanna Gunnlaugsdottir[411], Marc J. Gunter[1], Xiu-Hua Guo[416], Yin Guo[310], Prakash C. Gupta[417], Preeti Gupta[418], Rajeev Gupta[419], Oye Gureje[420], Mirjana A. Gurinović[421], Enrique Gutiérrez González[102], Laura Gutierrez[422], Felix Gutzwiller[193], Mònica Guxens[423], Xinyi Gwee[251], Seongjun Ha[424], Farzad Hadaegh[425], Charalambos A. Hadjigeorgiou[426], Rosa Haghshenas[41], Gahraman Hagverdiyev[370], Hamid Hakimi[342], Jytte Halkjær[427], Sameh S. Hallaq[42], Ian R. Hambleton[428], Behrooz Hamzeh[429], Dominique Hange[181], Abu A. M. Hanif[56], Sari Hantunen[27], Jie Hao[416], Carla Menêses Hardman[430], Louise Hardy[33], Tina Harmer Lassen[431], Javad Harooni[432], Seyed Mohammad Hashemi-Shahri[433], Mitra Hasheminia[425], Maria Hassapidou[434], Jun Hata[435], Teresa Haugsgjerd[106], Chika Hayashi[436], Alison J. Hayes[33], Jiang He[437], Yuan He[438], Yuna He[38], Mehdi Hedayati[116], Regina Heidinger-Felső[439], Margit Heier[313], Mirjam Heinen[148], Tatjana Hejgaard[440], Marleen Elisabeth Hendriks[441], Rafael dos Santos Henrique[430], Ana Henriques[100], Leticia Hernandez Cadena[442], Sauli Herrala[112], Marianella Herrera-Cuenca[202], Victor M. Herrera[443], Isabelle Herter-Aeberli[444], Karl-Heinz Herzig[35,112], Ramin Heshmat[445], Barbara Heude[244,392], Allan G. Hill[272], Sai Yin Ho[110], Michael Hobbs[446], Doroteia A. Höfelmann[447], Michelle Holdsworth[299], Reza Homayounfar[116], Clara Homs[395,448],

Emiel O. Hoogendijk[449], Wilma M. Hopman[450], Andrea R. V. R. Horimoto[166], Claudia M. Hormiga[451], Bernardo L. Horta[109], Farhad Hosseinpanah[452], Leila Houti[453], Christina Howitt[428], Thein Thein Htay[454], Aung Soe Htet[407], Maung Maung Than Htike[455], Yonghua Hu[37], José María Huerta[84,456], Ilpo Tapani Huhtaniemi[1], Laetitia Huiart[300], Constanta Huidumac Petrescu[280], Martijn Huisman[449], Abdullatif S. Husseini[47], Chinh Nguyen Huu[306], Inge Huybrechts[457], Nahla Hwalla[386], Jolanda Hyska[458], Licia Iacoviello[191,459], Ellina M. Iakupova[176], Jesús M. Ibarluzea[84], Norazizah Ibrahim Wong[44], Jannicke Igland[106], Chinwuba Ijoma[460], Edolem Ikerdeu[461], M. Arfan Ikram[462], Carmen Iñiguez[84,463], Violeta Iotova[371], Maybelline Joy B. Ipil[464], Vilma E. Irazola[422], Takafumi Ishida[465], Godsent C. Isiguzo[466], Muhammad Islam[15], Sheikh Mohammed Shariful Islam[467], Duygu Islek[5], Ivaila Y. Ivanova-Pandourska[468], Masanori Iwasaki[469], Tuija Jääskeläinen[28], Rod T. Jackson[147], Jeremy M. Jacobs[470], Michel Jadoul[471], Tazeen H. Jafar[242], Kenneth James[39], Konrad Jamrozik[387,891], Nataša Jan[472], Anna Jansson[473], Imre Janszky[474], Edward Janus[475], Juel Jarani[476], Gerald Jarnig[477], Marjo-Riitta Jarvelin[1,35], Grazyna Jasienska[478], Ana Jelaković[479], Bojan Jelaković[11], Garry Jennings[480], A. M. Jibo[481], Ramon O. Jimenez[482], Karl-Heinz Jöckel[483], Michel Joffres[139], Jari J. Jokelainen[112], Jost B. Jonas[484], Lars Jøran Kjerpeseth[101], Torben Jørgensen[8], Rohina Joshi[485], Josipa Josipović[486], Farahnaz Joukar[487], Pekka Jousilahti[28,341], Jacek J. Jóźwiak[488], Debra S. Judge[446], Anne Juolevi[28], Gregor Jurak[12], Iulia Jurca Simina[257], Vesna Juresa[11], Rudolf Kaaks[206], Niina E. Kaartinen[28], Felix O. Kaducu[489], Agnes L. Kadvan[490], Anthony Kafatos[491], Maria Kafyra[297], Mónika Kaj[492], Eero O. Kajantie[200], Sree Ramakrishna Kakani[29], Bernard Kakuhikire[493], Natia Kakutia[494,495], Daniela Kállayová[496], Zhanna Kalmatayeva[323], Natasa Kalpourtzi[118], Ofra Kalter-Leibovici[497], Yves Kameli[299], Kodanda R. Kanala[498], Srinivasan Kannan[499], Efthymios Kapantais[500], Anna Kapustina[122], Eva Karaglani[297], Line L. Kårhus[8], Khem B. Karki[501], Omat Karlsson[502], Adoubi Kassi Anicet[503], Philippe B. Katchunga[504], Marzieh Katibeh[287], Prasad Katulanda[353], Joanne Katz[2], Peter T. Katzmarzyk[505], Jussi Kauhanen[27], Prabhdeep Kaur[506], Maryam Kavousi[462], Gyulli M. Kazakbaeva[176], François F. Kaze[32], Benson M. Kazembe[507], Calvin Ke[508], Youzhi Kel[510], Ulrich Keil[510], Lital Keinan Boker[51], Sirkka M. Keinänen-Kiukaanniemi[112], Roya Kelishadi[512], Cecily Kelleher[148], Han C. G. Kemper[449], Maryam Keramati[513], Mathilde Kersting[514], Bobby Kgosiemang[515], Yousef Saleh Khader[141], Kazem Khalagi[65], Arsalan Khaledifar[64], Davood Khalili[516], Bahareh Kheiri[116], Motahareh Kheradmand[517], Irina V. Khorosheva[301], Alireza Khosravi Farsani[518], Ilse M. S. L. Khouw[30], Saeed Khwaja Mir Islam[519], Ursula Kiechl-Kohlendorfer[520], Stefan Kiechl[520,521], Japhet Killewo[522], Hyeon Chang Kim[523], Jenny M. Kindblom[181,524], Heidi Klakk[525], Suntara Klanarong[526,891], Jana Klanova[527], Magdalena Klimek[478], Jurate Klumbiene[162], Michael Knoflach[520], Susanne Kobel[528], Maciej Kochman[126], Bhawesh Koirala[529], Sweta Koirala[304], Elin Kolle[94], Sanda M. Kolo[530], Patrick Kolsteren[199], Jürgen König[532], Päivikki Koponen[28], Raija Korpelainen[35], Paul Korrovits[533], Magdalena Korzycka[327], Jelena Kos[130], Seppo Koskinen[341], Katsuyasu Kouda[366], Malik Koussoh Simone[534], Éva Kovács[535], Viktoria Anna Kovacs[492], Irina Kovalskys[536], Sudhir Kowlessur[537], Slawomir Koziel[397], Jana Kratenova[228], Wolfgang Kratzer[538], Vilma Kriaucioniene[162], Susi Kriemler[193], Peter Lund Kristensen[259], Helena Krizan[539], Maria F. Kroker-Lobos[540], Steinar Krokstad[474], Daan Kromhout[230], Herculina S. Kruger[541,542], Ruan Kruger[541], Łukasz Kryst[543], Ruzena Kubinova[228], Renata Kuciene[162], Urho M. Kujala[544], Enisa Kujundzic[119], Zbigniew Kulaga[404], Mukhtar Kulimbet[545], Vaitheeswaran Kulothungan[546], Richard Kumapley[3], R. Krishna Kumar[547], Meena Kumari[7], Marie Kunešová[548], Yadlapalli S. Kusuma[549], Vladimir Kutsenko[122], Kari Kuulasmaa[28], Catherine Kyobutungi[24], Quang Ngoc La[550], Fatima Zahra Laamiri[551], Demetre Labadarios[338,552], Idoia Labayen[553], Carl Lachat[199], Karl J. Lackner[167], Jouni Lahti[28], Daphne Lai[554], Wai Kent Lai[44], Youcef Laid[555], Lachmie Lall[556], Ecosse L. Lamoureux[418], Maritza Landaeta Jimenez[202], Edwige Landais[299], Anne Langsted[57,58], Tiina Lankila[557], Vera Lanska[403], Georg Lappas[558], Bagher Larijani[559], Mina P. Lateva[371], Tint Swe Latt[560], Martino Laurenzi[561], María Lazo-Porras[562], Gwenaëlle Le Coroller[345], Khanh Le Nguyen Bao[306], Agnès Le Port[299], Tuyen D. Le[306], Jeannette Lee[251,563], Paul H. Lee[27], Terho Lehtimäki[564,565], Daniel Lemogoum[566], David A. Leon[354], Elvynna Leong[554], Aitana Lertxundi[567], Nerea Lertxundi[568], Branimir Leskošek[12], Justyna Leszczak[126], Katja B. Leth-Møller[8], Gabriel M. Leung[110], Esko Levälahti[28], Sergey P. Levushkin[569], Yanping Li[19], Merike Liivak[570], Christa L. Lilly[571], Charlie Lim[251,563], Wei-Yen Lim[572], Maria Fernanda Lima-Costa[573], Yi-Jing Lin[248], Lars Lind[574], Vijaya Lingam[381], Birgit Linkohr[313], Allan Linneberg[8], Jakob Linseisen[575], Lauren Lissner[181], Mieczyslaw Litwin[404], Jing Liu[576], Lijuan Liu[310], Liping Liu[320], Xiaotian Liu[577], Yang Liu[509], Sabrina Llop[578], Wei-Cheng Lo[579], Helle-Mai Loit[570], Ayesha Lokubalasooriya[627], Khuong Quynh Long[550], Carla Lopes[100], Luis Lopes[100], Marcus V. V. Lopes[581], Oscar Lopes[582], Esther Lopez-Garcia[84,125], José Francisco López-Gil[583,584], Tania Lopez[585], Paulo A. Lotufo[166], José-Eugenio Lozano-Alonso[586], Gabriel Lozano-Berges[234], Janice L. Lukrafka[587], Dalia Luksiene[162], María Delia Luna[588], Nuno Lunet[100], Charles Lunogelo[589], Michala Lustigová[228,263], Edyta Łuszczki[126], Jean-René M'Buyamba-Kabangu[590], Guansheng Ma[37], Jun Ma[37], Xu Ma[438], George L. L. Machado-Coelho[142], Aristides M. Machado-Rodrigues[591], Enguerran Macia[592], Luisa M. Macieira[593], Ahmed A. Madar[180], Sherilynn Madraisau[461], Anja L. Madsen[8], Gladys E. Maestre[594], Stefania Maggi[195], Dianna J. Magliano[596], Emmanuella Magriplis[597], Gowri Mahasampath[149], Bernard Maire[299], Marjeta Majer[11], Marcia Makdisse[598], Päivi Mäki[28], Mohammad-Reza Malekpour[41], Fatemeh Malekzadeh[65], Reza Malekzadeh[65], Rahul Malhotra[242], Laurent Malisoux[345], Sofia K. Malyutina[599], Lynell V. Maniego[59], Yannis Manios[297], Jim I. Mann[600], Satu Männistö[8], Fariborz Mansour-Ghanaei[487], Taru Manyanga[601], Enzo Manzato[225], Mala Ali Mapatano[195], Anie Marcil[268], Francisco Mardones[163], Paula Margozzini[163], Joany Mariño[315], Mihaela Marinović Glavić[479], Anastasia Markaki[602], Oonagh Markey[603], Josko Markic[604], Eliza Markidou Ioannidou[605], Pedro Marques-Vidal[378,604], Larissa Pruner Marques[283], Jaume Marrugat[378,609], Dries Martens[610], Yves Martin-Prevel[299], Rosemarie Martin[41], Borja Martinez-Tellez[219], Vicente Martínez-Vizcaíno[612], Reynaldo Martorell[5], Eva Martos[613], Fatai A. Maruf[614], Katharina Maruszczak[615], Stefano Marventano[373], Giovanna Masala[616], Luis P. Mascarenhas[617], Mannix Masimango Imani[618], Masoud Masinaei[41], Ellisiv B. Mathiesen[407], Prashant Mathur[546], Alicia Matijasevich[166], Piotr Matłosz[126], Tandi E. Matsha[619], Victor Matsudo[620], Giletta Matteo[199], Pallab K. Maulik[621], Christina Mavrogianni[297], Artur Mazur[126], Camille M. Mba[32], Shelly R. McFarlane[39],

Stephen T. McGarvey[622], Keeley McGee[446], Martin McKee[354], Rachael M. McLean[600], Scott B. McLean[268], Breige A. McNulty[148], Sounnia Mediene Benchekor[453], Jurate Medzioniene[162], Kirsten Mehlig[181], Vrinda Mehra[436], Amir Houshang Mehrparvar[623], Jørgen Meisfjord[101], Christine Meisinger[313], Jesus D. Melgarejo[594], Marina Melkumova[624], Júlio B. Mello[625], Sofia Mendes[626], Fabián Méndez[133], Carlos O. Mendivil[627], Ana Maria B. Menezes[109], Geetha R. Menon[245], Gert B. M. Mensink[34], Maria Teresa Menzano[372], Indrapal I. Meshram[29], Diane T. Meto[628], Haakon E. Meyer[180], Jie Mi[246], Kim F. Michaelsen[58], Nathalie Michels[199], Kairit Mikkel[358], Jelena P. Milešević[421], Jody C. Miller[600], Olga Milushkina[629], Cláudia S. Minderico[630], G. K. Mini[631], Juan Francisco Miquel[163], Mohammad Reza Mirjalili[623], Daphne Mirkopoulou[632], Parvin Mirmiran[633], Masoud Mirzaei[623], Marjeta Mišigoj-Durakovic[11], Antonio Mistretta[373], Veronica Mocanu[634], Ana Mocumbi[635], Pietro A. Modesti[636], Jobe Modou[354], Sahar Saeedi Moghaddam[41], Shukri F. Mohamed[24], Kazem Mohammad[65], Mohammad Reza Mohammadi[637], Zahra Mohammadi[65], Noushin Mohammadifard[638], Viswanathan Mohan[14], Sherina Mohd Sidik[639], Muhammad Fadhli Mohd Yusoff[44], Iraj Mohebbi[60], Diego Moliner Urdiales[150], Line T. Møllehave[8], Niels C. Møller[259], Dénes Molnár[640], Amirabbas Momenan[425], Charles K. Mondo[641], Rafael Monge-Rojas[642], Michele M. Monroy-Valle[643], Roger A. Montenegro Mendoza[644], Eric Monterrubio-Flores[442], Kotsedi Daniel K. Monyeki[645], Jin Soo Moon[646], Mahmood Moosazadeh[517], Hermine T. Mopa[32], Farhad Moradpour[383], Leila B. Moreira[365], Alain Morejon[647], Luis A. Moreno[277,295], Francis Morey[648], Karen Morgan[22], Suzanne N. Morin[268], Erik Lykke Mortensen[58], George Moschonis[649], Alireza Moslem[650], Mildrey Mosquera[133], Malgorzata Mossakowska[651], Aya Mostafa[652], Seyed-Ali Mostafavi[65], Anabela Mota-Pinto[591], Eugen Mota[653], Jorge Mota[100], Maria Mota[653], Mohammad Esmaeel Motlagh[253], Jorge Moukawa[654], Robert Moumakwa[15], Marcos André Moura-dos-Santos[136], Yeva Movsesyan[624], Malay K. Mridha[56], Kelias P. Msyamboza[655], Alicia Mtijasevich[656], Thet Thet Mu[657], Magdalena Muc[591], Florian Muca[658], Boban Mugoša[119], Maria L. Muiesan[659], Martina Müller-Nurasyid[167], Patricia B. Munroe[660], Adrià Muntaner-Mas[661], Thomas Münzel[167], Molly M. Murphy[642], Celine Murrin[148], Jaakko Mursu[27], Elaine M. Murtagh[662], Kamarul Imran Musa[663], Sanja Music Milanovic[11,539], Vera Musil[11], Geofrey Musinguzi[664], Muel Telo M. C. Muyer[195], Iraj Nabipour[665], Gabriele Nagel[666], Farid Najafi[429], Harunobu Nakamura[366], Hanna Nalecz[667], Jana Námešná[113], Ei Ei K. Nang[251,563], Vinay B. Nangia[668], Martin Nankap[669], Sameer Narake[417], K. M. Venkat Narayan[5], Paola Nardone[262], Take Naseri[670], Tim Nawrot[610,671], William A. Neal[571], Nareemarn Neelapaichit[54], Mayssam Nehme[377], Azim Nejatizadeh[329], Ilona Nenko[478], Martin Neovius[293], Flavio Nervi[163], Olena Nesterova[672], Dinesh Neupane[2], Tze Pin Ng[251], Chung T. Nguyen[673], Nguyen D. Nguyen[674], Quang Ngoc Nguyen[675], Michael Y. Ni[110], Rodica Nicolescu[280], Peng Nie[676], Ramfis E. Nieto-Martínez[677], Yury P. Nikitin[599], Guang Ning[172], Toshiharu Ninomiya[435], Nobuo Nishi[23], Sania Nishtar[678], Marianna Noale[595], Oscar A. Noboa[173], Helena Nogueira[591], Maria Nordendahl[61], Børge G. Nordestgaard[57,58], Kevin I. Norton[485], Davide Noto[127], Natalia Nowak-Szczepanska[397], Mohannad Al Nsour[679], Irfan Nuhoğlu[339], Kim R. Nuño[570], Moffat Nyirenda[354], Terence W. O'Neill[680], Dermot O'Reilly[343], Galina Obreja[269], Caleb Ochimana[19], Angélica M. Ochoa-Avilés[311], Eiji Oda[681], Augustine N. Odili[682], Kyungwon Oh[683], Kumiko Ohara[684], Claes Ohlsson[181,524], Ryutaro Ohtsuka[685,891], Örn Olafsson[411], Brian Oldenburg[596], Maria Teresa A. Olinto[365], Isabel O. Oliveira[109], Mohd Azahadi Omar[44], Sok Seang M. Omar[686], Altan Onat[687,891], Sok King Ong[688], N. Charlotte Onland-Moret[379], Lariane M. Ono[447], Obinna Onodugo[460], Pedro Ordunez[221], Rui Ornelas[689], Francisco B. Ortega[219], Ana P. Ortiz[690], Pedro J. Ortiz[220], Merete Osler[8], Clive Osmond[272], Sergej M. Ostojic[305], Afshin Ostovar[65], Johanna A. Otero[691], Charlotte B. Ottendahl[179], Akaninyene Otu[334], Kim Overvad[287], Ellis Owusu-Dabo[692], Adetoyeje Y. Oyeyemi[693], Adewale L. Oyeyemi[694], Fred Michel Paccaud[18], Cristina P. Padez[591], Ioannis Pagkalos[434], Marat Pahimov[20], Elena Pahomova[326], Karina Mary de Paiva[283], Andrzej Pająk[478], Natalja Pajula[570], Alberto Palloni[695], Luigi Palmieri[262], Demosthenes Panagiotakos[297], Songhomitra Panda-Jonas[696], Arvind Pandey[245], Zengchang Pang[697], Francesco Panza[698], Antonio Paoli[225], Mariela Paoli[209], Sousana K. Papadopoulou[434], Dimitrios Papandreou[699], Rossina G. Pareja[308], Suvi Parikka[28], Soon-Woo Park[700], Suyeon Park[683], Winsome R. Parnell[600], Mahboubeh Parsaeian[65], Ionela Pascanu[152], Patrick Pasquet[89], Chona F. Patalen[59], Roengrudee Patanavanich[701], Nikhil D. Patel[702], Marcos Pattussi[302], Halyna Pavlyshyn[398], Raimund Pechlaner[520], Ivan Pećin[130], Dorthe C. Pedersen[8], Mangesh S. Pednekar[417], João M. Pedro[703], Ana B. Peinado[156], Sergio Viana Peixoto[573], Markku Peltonen[28], Guillem Pera[704], Alexandre C. Pereira[166], Marco A. Peres[705], Napoleon Perez-Farinos[102], Agustín Perez-Londoño[627], Cynthia M. Pérez[690], Markus Perola[28], Valentina Peterkova[189], Annette Peters[313], Janina Petkeviciene[162], Ausra Petrauskiene[162], Olga Petrovna Kovtun[98], Emanuela Pettenuzzo[706], Niloofar Peykari[304], Norbert Pfeiffer[167], Son Thai Pham[707], Felix P. Phiri[708], Rafael N. Pichardo[709], Preux Pierre-Marie[710], Iris Pigeot[67], Hynek Pikhart[186], Aida Pilav[120], Pavel Piler[527], Lorenza Pilotto[711], Francesco Pistelli[712], Freda Pitakaka[713], Aleksandra Piwonska[714], Andreia N. Pizarro[100], Pedro Plans-Rubió[715], Alina G. Platonova[716], Bee Koon Poh[717], Hermann Pohlabeln[67], Raluca M. Pop[152], Barry M. Popkin[319], Stevo R. Popovic[178], Miquel Porta[84,378], Georg Posch[145], Anil Poudyal[177], Dimitrios Poulimeneas[434], Hamed Pouraram[65], Farhad Pourfarzi[718], Akram Pourshams[65], Hossein Poustchi[65], Dorairaj Prabhakaran[14], Rajendra Pradeepa[14], Andrew Prentice[354], Alison J. Price[354], Jacqueline F. Price[720], Antonio Prista[721], Rui Providencia[186], Jardena J. Puder[606], Iveta Pudule[722], Soile Puhakka[557], Maria Puiu[257], Margus Punab[533], Muhammed S. Qadir[723], Radwan F. Qasrawi[42], Qing Qiao[724], Mostafa Qorbani[725], Anna Quialheiro[726], Hedley K. Quintana[644], Pedro J. Quiroga-Padilla[627], Tran Quoc Bao[727], Stefan Rach[67], Maria-Victoria Racu[728], Ivana Radic[729], Ricardas Radisauskas[162], Salar Rahimikazerooni[348], Mahfuzar Rahman[730], Mahmudur Rahman[731], Olli Raitakari[732], Manu Raj[547], Tamerlan Rajabov[733], Sherali Rakhmatulloev[46], Ivo Rakovac[214], Sudha Ramachandra Rao[506], Ambady Ramachandran[241], Otim P. C. Ramadan[241], Virgílio V. Ramires[735], Manuel Ramirez-Zea[540], Jacqueline Ramke[147], Elisabete Ramos[100], Rafel Ramos[736,737], Lekhraj Rampal[639], Sanjay Rampal[30], Sheena E. Ramsay[738], João F. L. B. Rangel Junior[136], Lalka S. Rangelova[255], Harish Ranjani[14], Ravindra P. Rannan-Eliya[739], João F. Raposo[740], Patricia Rarau[3], Vayia Rarra[741], Ramon A. Rascon-Pacheco[49], Mohammad-Mahdi Rashidi[41], Cassiano Ricardo Rech[283], Cristina Recuero Carretero[102], Josep Redon[742], Valéria Regecová[743], Jane D. P. Renner[744], Judit A. Repasy[439], Cézane P. Reuter[744], Luís Revilla[585], Andrew Reynolds[600], Negar Rezaei[65],

Abbas Rezaianzadeh[348], Yeunsook Rho[424], Lourdes Ribas-Barba[745], Robespierre Ribeiro[746,891], Rogério T. Ribeiro[740], Elio Riboli[1], Fernando Rigo[747], Attilio Rigotti[163], Leanne M. Riley[3], Natascia Rinaldo[410], Tobias F. Rinke de Wit[748], Ulf Risérus[574], Ana I. Rito[134], Raphael M. Ritti-Dias[749], Juan A. Rivera[442], Reina G. Roa[750], Romana Roccaldo[238], Daniela Rodrigues[591], Fernando Rodríguez-Artalejo[84,125], Manuel A. Rodríguez-Pérez[104], María del Cristo Rodriguez-Perez[751], Laura A. Rodríguez-Villamizar[752], Andrea Y. Rodríguez[753], Ulla Roggenbuck[483], Peter Rohloff[754], Fabian Rohner[755], Rosalba Rojas-Martinez[442], Gemma Rojo-Martínez[756,757], Nipa Rojroongwasinkul[54], Almudena Rollán Gordo[102], Dora Romaguera[277,758], Elisabetta L. Romeo[294], Gil B. Rosa[630], Rafaela V. Rosario[759], Annika Rosengren[181,524], Ian Rouse[760], Vanessa Rouzier[761], Joel G. R. Roy[268], Maira Ruano Estrada[762], Maira H. Ruano[763], Adolfo Rubinstein[422], Frank J. Rühli[193], Jean-Bernard Ruidavets[192], Blanca Sandra Ruiz-Betancourt[49], Maria Ruiz-Castell[345], Emma Ruiz Moreno[84,764], Iuliia A. Rusakova[176], Wojciech Rusek[765], Kenisha Russell Jonsson[473], Paola Russo[766], Petra Rust[532], Marcin Rutkowski[767], Katri Sääksjärvi[28], Marge Saamel[570], Crizian G. Saar[233], Charumathi Sabanayagam[768], Kalpana Sabapathy[354], Hamideh Sabbaghi[116], Shaun Sabico[71], Harshpal S. Sachdev[769], Alireza Sadjadi[65], Ali Reza Safarpour[348], Sare Safi[116], Mohammad Hossien Saghi[650], Olfa Saidi[151], Calogero Saieva[616], Satoko Sakata[435], Nader Saki[253], Sanja Šalaj[11], Benoit Salanave[300], Eduardo Salazar Martinez[442], Akkumis Salkhanova[770], Diego Salmerón[84,456], Veikko Salomaa[732], Jukka T. Salonen[341], Massimo Salvetti[659], Margarita Samoutian[771], Jose Sánchez-Abanto[772], Guillermo Sanchez-Delgado[219], Mairena Sánchez-López[612], Joaquin Sanchis-Moysi[312], Sandjaja[773], Susana Sans[774], Loreto Santa-Marina[775], Ethel Santacruz Lezcano[155], Diana A. Santos[630], Ina S. Santos[109], Lèlita C. Santos[593], Maria Paula Santos[100], Osvaldo Santos[776], Palmira Santos[635], Rute Santos[759], Tamara R. Santos[777], Vonthanak Saphonn[778], Jouko L. Saramies[779], Luis B. Sardinha[630], Nizal Sarrafzadegan[638], Yoko Sato[780], Kai-Uwe Saum[206], Stefan Savin[3], Savvas Savva[426], Mathilde Savy[299], Norie Sawada[781], Mariana Sbaraini[365], Marcia Scazufca[782], Beatriz D. Schaan[365], Angelika Schaffrath Rosario[34], Herman Schargrodsky[783,891], Karin Schindler[784], Amand Floriaan Schmidt[186,207], Börge Schmidt[483], Carsten O. Schmidt[315], Andrea Schneider[313], Peter Schnohr[8], Catherine Mary Schooling[110], Ben Schöttker[206], Sara Schramm[483], Stine Schramm[259], Helmut Schröder[84,378], Constance Schultsz[207], Gry Schultz[785], Matthias B. Schulze[786], Aletta E. Schutte[485,787], Moslem Sedaghattalab[432], Rusidah Selamat[44], Amand Sen[788], Idowu O. Senbanjo[789], Sadaf G. Sepanlou[65], Guillermo Sequera[155], Luis Serra-Majem[312], Jennifer Servais[784], Ľudmila Ševčíková[790], Ronel Sewpaul[791], Svetlana A. Shalnova[122], Teresa Shamah-Levy[442], Seyed Morteza Shamshirgaran[115], Shubash Shander[44], Coimbatore Subramaniam Shanthirani[14], Maryam Sharafkhah[65], Sanjib K. Sharma[529], Almaz Sharman[770], Jonathan E. Shaw[596], Amaneh Shayanrad[65], Ali Akbar Shayesteh[253], Nurzhamal Sheisheeva[68], Ching-Fen Shen[248], Lela Shengelia[494], Kenji Shibuya[792], Hana Shimizu-Furusawa[793], Tal Shimony[511], Avi Shina[794], Igor D. Shkrobanets[716], Marat Shoranov[160], Khairil Si-Ramlee[688], Alfonso Siani[766], Abla M. Sibai[386], Labros S. Sidossis[795], Mark J. Siedner[796,797], Antonio M. Silva[143], Caroline Ramos de Moura Silva[136], Diego Augusto Santos Silva[283], Kelly Samara Silva[283], Xueling Sim[251,563], Mary Simon[241], Judith Simons[798], Leon A. Simons[485], Ivan Simunovic[604], Agneta Sjöberg[181], Michael Sjöström[293,891], Elena V. Skoblina[799], Natalia A. Skoblina[629], Tatyana Slazhnyova[144], Jolanta Slowikowska-Hilczer[651], Przemyslaw Slusarczyk[651], Liam Smeeth[354], Lee Smith[196], Hung-Kwan So[110], Fernanda Cunha Soares[136], Grzegorz Sobek[126], Eugène Sobngwi[32], Morten Sodemann[259], Stefan Söderberg[61], Moesijanti Y. E. Soekatri[800], Agustinus Soemantri[801,891], Raquel Soler-Blasco[578], Vincenzo Solfrizzi[698], Yuliya V. Solovieva[802], Mohammad Hossein Somi[346], Emily Sonestedt[389], Sajid Soofi[16], Thorkild I. A. Sørensen[58], Elin P. Sørgjerd[474], Victoria E. Soto-Rojas[413], Aïcha Soumaré[803], Alfonso Sousa-Poza[804], Mafalda Sousa-Uva[134], Mam Sovatha[778], Agnieszka Sozańska[126], Bente Sparboe-Nilsen[805], Karen Sparrenberger[365], Vita Speckauskiene[162], Phoebe R. Spencer[446], Angela Spinelli[262], Igor Spiroski[806,807], Jan A. Staessen[194], Aleksandra Stamenova[807], Hanspeter Stamm[808], Laura Stanciulescu[809], Andreas Stang[483], Kaspar Staub[193], Bill Stavreski[480], Jostein Steene-Johannessen[94], Peter Stehle[810], Aryeh D. Stein[5], Silje Steinsbekk[474], George S. Stergiou[118], Jochanan Stessman[470], Jutta Stieber[313,891], Doris Stöckl[313], Jakub Stokwiszewski[811], Katarzyna Stoś[811], Ekaterina Stoyanova[812], Gareth Stratton[212], Karien Stronks[207], Maria Wany Strufaldi[360], Lela Sturua[494], Milton F. Suarez-Ortegón[813], Phalakorn Suebsamran[814], Mindy S. Sugiyama[461], Machi Suka[815], Gerhard Sulo[106], Malin Sund[61], Johan Sundström[574], Yn-Tz Sung[816], Jordi Sunyer[85], Suparmi[55], Unursaikhan Surenjav[67], Paibul Suriyawongpaisal[54], Kitti Susovits[121], Nabil William G. Sweis[818], Boyd A. Swinburn[147], René Charles Sylva[819], Unni Syversen[820], Lucjan Szponar[811], Yasuharu Tabara[780], Lorraine Tabone[265], E. Shyong Tai[251,563], Konstantinos D. Tambalis[118], Mari-Liis Tammesoo[358], Abdonas Tamosiunas[162], Eng Joo Tan[344], Baimakhan Tanabayev[821], Nikhil Tandon[822], Xun Tang[37], Maya Tanrygulyyeva[823], Frank Tanser[338], Yong Tao[37], Mohammed Rasoul Tarawneh[824], Jakob Tarp[37], Carolina B. Tarqui-Mamani[772], Radka Taxová Braunerová[548], Anne Taylor[387], Félicité Tchibindat[825], Saskia Te Velde[826], William R. Tebar[260], Fahimeh R. Tehrani[827], Grethe S. Tell[106], Tania Tello[562], Masresha Tessema[828], Lukas Teufl[784], Yih Chung Tham[251,768], K. R. Thankappan[829], Holger Theobald[231], Xenophon Theodoridis[390], Sathish Thirunavukkarasu[14], Nihal Thomas[830], Barbara Thorand[313], Amanda G. Thrift[344], Ľubica Tichá[790], Erik J. Timmermans[831], Dwi Hapsari Tjandrarini[55], Anne Tjonneland[427], Ervin Toçi[458], Maryam Tohidi[425], Hanna K. Tolonen[28], Janne S. Tolstrup[179], Maciej Tomaszewski[680], Murat Topbas[339], Roman Topór-Mądry[478], Pere Torán-Monserrat[704,737], Liv Elin Torheim[805], Michael J. Tornaritis[426], Maties Torrent[832], Laura Torres-Collado[84,270], Duarte Torres[100], Silvia Torres[395,833], Stefania Toselli[834], Giota Touloumi[118], Luciana Tovo-Rodrigues[109], Pierre Traissac[299], Thi Tuyet-Hanh Tran[550], Mark S. Tremblay[581], Areti Triantafyllou[390], Antonia Trichopoulou[835], Oanh T. H. Trinh[674], Justina Trišauskė[162], Atul Trivedi[836], Alexander C. Tsai[837], Lechaba Tshepo[838], Thomas Tsiampalis[297], Maria Tsigga[839], Panagiotis Tsintavis[434], Shoichiro Tsugane[23], John Tuitele[840,841], Azaliia M. Tuliakova[176], Marshall K. Tulloch-Reid[39], Fikru Tullu[842], Tomi-Pekka Tuomainen[27], Jaakko Tuomilehto[28], Maria L. Turley[843], Gilad Twig[844], Per Tynelius[293], Evangelia Tzala[1], Themistoklis Tzotzas[500], Christophe Tzourio[803], Nwanneamaka Udoji[845], Peter Ueda[293], Eunice Ugel[846], Flora A. M. Ukoli[847], Hanno Ulmer[520], Belgin Unal[340], Zhamyila Usupova[68], Hannu M. T. Uusitalo[348], Nalan Uysal[849], Sergio Valdes[756,757], Gonzalo Valdivia[163],

Susana Vale[850], Popov I. Valery[851], Majid Valizadeh[452], Damaskini Valvi[852], Rob M. van Dam[853], Bert-Jan van den Born[207], Johan Van der Heyden[158], Yvonne T. van der Schouw[379], Koen Van Herck[199], Wendy Van Lippevelde[199], Hoang Van Minh[550], Natasja M. Van Schoor[449], Irene G. M. van Valkengoed[207], Dirk Vanderschueren[194], Diego Vanuzzo[854], Anette Varbo[57,58], Gregorio Varela-Moreiras[855], Luz Nayibe Vargas[273], Senthil K. Vasan[272], Daniel G. Vasques[365], Radu Vatasescu[210], Tomas Vega[856], Toomas Veidebaum[570], Gustavo Velasquez-Melendez[233], Biruta Velika[722], Michel Velten[174,857], Charlotte Verdot[300], Maïté Verloigne[199], Giovanni Veronesi[274], W. M. Monique Verschuren[184], Germán Vicente-Rodríguez[234,277], Cesar G. Victora[109], Josep Vidal-Conti[661], Giovanni Viegi[858], Lucie Viet[184], Frøydis N. Vik[157], Monica Vilar[859], Salvador Villalpando[442], Luis Villarroel[163], Jesus Vioque[84,270], Napaphan Viriyautsahakul[860], Jyrki K. Virtanen[27], Marjolein Visser[449], Bharathi Viswanathan[17], Chiranthika Vithana[580], Mihaela Vladulescu[861], Tiina Vlasoff[862], Peter Vollenweider[606,607], Henry Völzke[315], Georgia Vourli[863], Ari Voutilainen[27], Martine Vrijheid[84,85], Tanja G. M. Vrijkotte[207], Silvije Vuletić[11], Alisha N. Wade[864], Wakenge Wakilongo[244], Thomas Waldhör[784], Janette Walton[216], Elvis O. A. Wambiya[24], Wan Mohamad Wan Bebakar[663], Wan Nazaimoon Wan Mohamud[865], Rildo de Souza Wanderley Júnior[136], Chongjian Wang[577], Huijun Wang[38], Ningli Wang[310], Qian Wang[866], Xiangjun Wang[867], Ya Xing Wang[416], Yi-Ren Wang[248], S. Goya Wannamethee[186], Nicholas Wareham[868], Olivia Wartha[528], Adelheid Weber[153], Karen Webster-Kerr[406], Niels Wedderkopp[259], Daniel Weghuber[615], Li Wei[869], Wenbin Wei[310], Aneta Weres[126], Bo Werner[870], Leo D. Westbury[272], Peter H. Whincup[186], Lars Wichstrøm[474], Kremlin Wickramasinghe[214], Kurt Widhalm[784], Indah S. Widyahening[872], Andrzej Więcek[261], Nilmini Wijemunige[739], Philipp S. Wild[167], Rainford J. Wilks[39], Johann Willeit[520], Karin Willeit[520], Peter Willeit[520], Julianne Williams[214], Tom Wilsgaard[407], James P. Wirth[873], Agnieszka Wiśniowska-Szurlej[811], Bogdan Wojtyniak[811], Meseret Woldeyohannes[828], Kathrin Wolf[313], Roy A. Wong-McClure[40], Andrew Wong[186], Emily B. Wong[796], Jyh Eiin Wong[717], Mark Woodward[1,485], Agnieszka E. Woźniak[811], Frederick C. Wu[680], Hon-Yen Wu[874], Jianfeng Wu[171], Li Juan Wu[416], Shouling Wu[249], Justyna Wyszyńska[126], Haiquan Xu[875], Liang Xu[876], Can Can Xue[768], Nor Azwany Yaacob[663], Uruwan Yamborisut[54], Li Yan[1], Lily D. Yan[877], Weili Yan[878], Ling Yang[250], Xiaoguang Yang[38], Yang Yang[877], Nazan Yardim[237], Chao-Yu Yeh[248], Martha Yépez García[859], Panayiotis K. Yiallouros[879], Agneta Yngve[574], Chandra Mandil Yogal[474], Moein Yoosefi[41], Akihiro Yoshihara[880], Yoto Yotov[371], Qi Sheng Yow[416], Yu-Ling Yu[293], Yunjiang Yu[881], Safiah Md Yusof[882], Ahmad Faudzi Yusoff[44], Luciana Zaccagni[410], Vassilis Zafiropulos[883], Ahmad A. Zainuddin[44], Farhad Zamani[884], Sabina Zambon[225], Antonis Zampelas[597], Hana Zamrazilová[548], Maria Elisa Zapata[232], Tomasz Zatoński[137], Ko Ko Zaw[560], Ayman A. Zayed[818], Tomasz Zdrojewski[767], Magdalena Żegleń[543], Kristyna Zejglicova[228], Girum Zeleke[828], Tajana Zeljkovic Vrkic[130], Yi Zeng[885], Andrea Zentai[37,502], Bing Zhang[38], Luxia Zhang[885], Zhen-Yu Zhang[194], Dong Zhao[576], Ming-Hui Zhao[885], Wenhua Zhao[38], Yanitsa V. Zhecheva[468], Wei Zheng[220], Bekbolat Zholdin[886], Maigeng Zhou[38], Dan Zhu[887], Oleg F. Zhukov[888], Sophia Zollner-Kiechl[520], Paul Zimmet[344], Marie Zins[244,392], Emanuel Zitt[145], Yanina Zocalo[173], Julio Zuñiga Cisneros[654], Monika Zuziak[889] & Majid Ezzati[1,53✉]

[1]Imperial College London, London, UK. [2]Johns Hopkins Bloomberg School of Public Health, Baltimore, MD, USA. [3]World Health Organization, Geneva, Switzerland. [4]University of California Berkeley, Berkeley, CA, USA. [5]Emory University, Atlanta, GA, USA. [6]Bill & Melinda Gates Foundation, Seattle, WA, USA. [7]University of Essex, Colchester, UK. [8]Copenhagen University Hospital — Bispebjerg and Frederiksberg, Copenhagen, Denmark. [9]Swiss Tropical and Public Health Institute, Basel, Switzerland. [10]University of Basel, Basel, Switzerland. [11]University of Zagreb, Zagreb, Croatia. [12]University of Ljubljana, Ljubljana, Slovenia. [13]Instituto Nacional de Ciencias Médicas y Nutrición, Mexico City, Mexico. [14]Madras Diabetes Research Foundation, Chennai, India. [15]The Hospital for Sick Children, Toronto, Ontario, Canada. [16]The Aga Khan University, Karachi, Pakistan. [17]Ministry of Health, Victoria, Seychelles. [18]Unisanté, Lausanne, Switzerland. [19]Harvard TH Chan School of Public Health, Boston, MA, USA. [20]Asfendiyarov Kazakh National Medical University, Almaty, Kazakhstan. [21]National Institute of Health, Kuala Lumpur, Malaysia. [22]RCSI University of Medicine and Health Sciences, Dublin, Ireland. [23]National Institutes of Biomedical Innovation, Health and Nutrition, Osaka, Japan. [24]African Population and Health Research Center, Nairobi, Kenya. [25]Seoul National University College of Medicine, Seoul, Republic of Korea. [26]University of Leicester, Leicester, UK. [27]University of Eastern Finland, Kuopio, Finland. [28]Finnish Institute for Health and Welfare, Helsinki, Finland. [29]ICMR — National Institute of Nutrition, Hyderabad, India. [30]University of Malaya, Kuala Lumpur, Malaysia. [31]National Taiwan University, Taipei, Taiwan. [32]University of Yaoundé 1, Yaoundé, Cameroon. [33]University of Sydney, Sydney, New South Wales, Australia. [34]Robert Koch Institute, Berlin, Germany. [35]University of Oulu, Oulu, Finland. [36]Public Health Promotion and Development Organization, Kathmandu, Nepal. [37]Peking University, Beijing, China. [38]Chinese Center for Disease Control and Prevention, Beijing, China. [39]The University of the West Indies, Kingston, Jamaica. [40]Caja Costarricense de Seguro Social, San José, Costa Rica. [41]Non-Communicable Diseases Research Center, Tehran, Iran. [42]Al-Quds University, East Jerusalem, Palestine. [43]S Kairbekova National Research Center for Health Development, Astana, Kazakhstan. [44]Ministry of Health, Kuala Lumpur, Malaysia. [45]Qatar University, Doha, Qatar. [46]Ministry of Health and Social Protection, Dushanbe, Tajikistan. [47]Birzeit University, Birzeit, Palestine. [48]Usmanu Danfodiyo University Teaching Hospital, Sokoto, Nigeria. [49]Instituto Mexicano del Seguro Social, Mexico City, Mexico. [50]Qassim University, Unaizah, Saudi Arabia. [51]RehaKlinika, Rzeszów, Poland. [52]Flinders University, Adelaide, South Australia, Australia. [53]University of Ghana, Accra, Ghana. [54]Mahidol University, Nakhon Pathom, Thailand. [55]National Research and Innovation Agency, Jakarta, Indonesia. [56]BRAC James P Grant School of Public Health, Dhaka, Bangladesh. [57]Copenhagen University Hospital — Herlev and Gentofte, Copenhagen, Denmark. [58]University of Copenhagen, Copenhagen, Denmark. [59]Food and Nutrition Research Institute, Taguig, Philippines. [60]Urmia University of Medical Sciences, Urmia, Iran. [61]Umeå University, Umeå, Sweden. [62]Ibn Tofail University, Kénitra, Morocco. [63]Amsterdam University Medical Center, Amsterdam, The Netherlands. [64]Shahrekord University of Medical Sciences, Shahrekord, Iran. [65]Tehran University of Medical Sciences, Tehran, Iran. [66]University of Hargeisa, Hargeisa, Somalia. [67]Leibniz Institute for Prevention Research and Epidemiology — BIPS, Bremen, Germany. [68]Republican Center for Health Promotion and Mass Communication, Bishkek, Kyrgyzstan. [69]National Center for Diabetes, Endocrinology and Genetics, Amman, Jordan. [70]Tashkent Pediatric Medical Institute, Tashkent, Uzbekistan. [71]King Saud University, Riyadh, Saudi Arabia. [72]Kuwait Institute

for Scientific Research, Kuwait City, Kuwait. [73]Princess Nourah bint Abdulrahman University, Riyadh, Saudi Arabia. [74]Ministry of Health, Muscat, Oman. [75]King Abdulaziz University, Jeddah, Saudi Arabia. [76]The Hashemite University, Zarqa, Jordan. [77]World Health Organization Regional Office for the Eastern Mediterranean, Cairo, Egypt. [78]Ministry of Health, Kuwait City, Kuwait. [79]Dasman Diabetes Institute, Kuwait City, Kuwait. [80]Aldara Hospital and Medical Center, Riyadh, Saudi Arabia. [81]King Abdullah International Medical Research Center, Riyadh, Saudi Arabia. [82]Universidade Federal da Integração Latino-Americana, Foz do Iguaçu, Brazil. [83]Republican Specialized Scientific-and-Practical Medical Centre of Endocrinology named after academician Y.Kh. Turakulov, Tashkent, Uzbekistan. [84]CIBERESP, Madrid, Spain. [85]Barcelona Institute for Global Health, Barcelona, Spain. [86]Bombay Hospital and Medical Research Centre, Mumbai, India. [87]Research Center for Social Determinants of Health, Tehran, Iran. [88]Ghana Health Service, Kintampo, Ghana. [89]UMR CNRS-MNHN 7206, Paris, France. [90]University of Lille, Lille, France. [91]Lille University Hospital, Lille, France. [92]Endocrine Research Center, Tehran, Iran. [93]Western Norway University of Applied Sciences, Sogndal, Norway. [94]Norwegian School of Sport Sciences, Oslo, Norway. [95]University of Thessaly, Trikala, Greece. [96]Ministry of Health and Social Action, Dakar, Senegal. [97]Health Promotion Research Center, Zahedan, Iran. [98]Yekaterinburg State Medical Academy, Yekaterinburg, Russia. [99]National Institute of Public Health, Tunis, Tunisia. [100]University of Porto, Porto, Portugal. [101]Norwegian Institute of Public Health, Oslo, Norway. [102]Spanish Agency for Food Safety and Nutrition, Madrid, Spain. [103]University of Massachusetts Amherst, Amherst, MA, USA. [104]University of Almería, Almería, Spain. [105]Nepal Health Economics Association, Kathmandu, Nepal. [106]University of Bergen, Bergen, Norway. [107]University of Iceland, Reykjavik, Iceland. [108]Haramaya University, Dire Dawa, Ethiopia. [109]Universidade Federal de Pelotas, Pelotas, Brazil. [110]The University of Hong Kong, Hong Kong, China. [111]University of Medicine 1, Yangon, Myanmar. [112]Oulu University Hospital, Oulu, Finland. [113]Regional Authority of Public Health, Banska Bystrica, Slovakia. [114]Diabetic Association of Bangladesh, Dhaka, Bangladesh. [115]Neyshabur University of Medical Sciences, Neyshabur, Iran. [116]Shahid Beheshti University of Medical Sciences, Tehran, Iran. [117]Indian Council of Medical Research, New Delhi, India. [118]National and Kapodistrian University of Athens, Athens, Greece. [119]Institute of Public Health, Podgorica, Montenegro. [120]University of Sarajevo, Sarajevo, Bosnia and Herzegovina. [121]National Center for Public Health and Pharmacy, Budapest, Hungary. [122]National Medical Research Center for Therapy and Preventive Medicine, Moscow, Russia. [123]University of Science and Technology, Sana'a, Yemen. [124]The John Paul II Catholic University of Lublin, Lublin, Poland. [125]Universidad Autónoma de Madrid, Madrid, Spain. [126]University of Rzeszów, Rzeszów, Poland. [127]University of Palermo, Palermo, Italy. [128]Ceara State University, Ceara, Brazil. [129]University of Miami, Miami, FL, USA. [130]University Hospital Centre Zagreb, Zagreb, Croatia. [131]Mohammed V University, Rabat, Morocco. [132]Unidad de Cirugia Cardiovascular, Guatemala City, Guatemala. [133]Universidad del Valle, Cali, Colombia. [134]National Institute of Health Doutor Ricardo Jorge, Lisbon, Portugal. [135]NOVA University Lisbon, Lisbon, Portugal. [136]University of Pernambuco, Recife, Brazil. [137]Wroclaw Medical University, Wroclaw, Poland. [138]Indus Hospital & Health Network, Karachi, Pakistan. [139]Simon Fraser University, Burnaby, British Columbia, Canada. [140]Dalhousie University, Halifax, Nova Scotia, Canada. [141]Jordan University of Science and Technology, Irbid, Jordan. [142]Universidade Federal de Ouro Preto, Ouro Preto, Brazil. [143]Universidade Federal do Maranhão, São Luís, Brazil. [144]National Center of Public Health, Astana, Kazakhstan. [145]Agency for Preventive and Social Medicine, Bregenz, Austria. [146]Cliniques Universitaires de Kinshasa, Kinshasa, Democratic Republic of Congo. [147]University of Auckland, Auckland, New Zealand. [148]University College Dublin, Dublin, Ireland. [149]Christian Medical College Vellore, Vellore, India. [150]Universitat Jaume I, Castellon de la Plana, Spain. [151]University Tunis El Manar, Tunis, Tunisia. [152]George Emil Palade University of Medicine, Pharmacy, Science and Technology of Târgu Mureş, Târgu Mureş, Romania. [153]Federal Ministry of Social Affairs, Health, Care and Consumer Protection, Vienna, Austria. [154]Cafam University Foundation, Bogotá, Colombia. [155]Ministerio de Salud Pública y Bienestar Social, Asunción, Paraguay. [156]Universidad Politécnica de Madrid, Madrid, Spain. [157]University of Agder, Kristiansand, Norway. [158]Sciensano, Brussels, Belgium. [159]Addis Continental Institute of Public Health, Addis Ababa, Ethiopia. [160]Kazakh National Medical University, Almaty, Kazakhstan. [161]Universidad Científica del Sur, Lima, Peru. [162]Lithuanian University of Health Sciences, Kaunas, Lithuania. [163]Pontificia Universidad Católica de Chile, Santiago, Chile. [164]Gasol Foundation, Madrid, Spain. [165]University of Alcalá, Alcalá de Henares, Spain. [166]University of São Paulo, São Paulo, Brazil. [167]Johannes Gutenberg University, Mainz, Germany. [168]B J Medical College, Ahmedabad, India. [169]Chirayu Medical College, New Delhi, India. [170]Sunder Lal Jain Hospital, Delhi, India. [171]Shandong University of Traditional Chinese Medicine, Jinan, China. [172]Shanghai Jiao-Tong University School of Medicine, Shanghai, China. [173]Universidad de la República, Montevideo, Uruguay. [174]University of Strasbourg, Strasbourg, France. [175]Institute of Medical Research and Medicinal Plant Studies, Yaoundé, Cameroon. [176]Ufa Eye Research Institute, Ufa, Russia. [177]Nepal Health Research Council, Kathmandu, Nepal. [178]University of Montenegro, Niksic, Montenegro. [179]University of Southern Denmark, Copenhagen, Denmark. [180]University of Oslo, Oslo, Norway. [181]University of Gothenburg, Gothenburg, Sweden. [182]Ministry of Health, Jerusalem, Israel. [183]Universidade Federal do Rio de Janeiro, Rio de Janeiro, Brazil. [184]National Institute for Public Health and the Environment, Bilthoven, The Netherlands. [185]University of Turin, Turin, Italy. [186]University College London, London, UK. [187]Liverpool John Moores University, Liverpool, UK. [188]Nanyang Technological University, Singapore, Singapore. [189]National Medical Research Center for Endocrinology, Moscow, Russia. [190]Centro de Educación Médica e Investigaciones Clínicas, Buenos Aires, Argentina. [191]IRCCS Neuromed, Pozzilli, Italy. [192]University of Toulouse, Toulouse, France. [193]University of Zurich, Zurich, Switzerland. [194]KU Leuven, Leuven, Belgium. [195]University of Kinshasa, Kinshasa, Democratic Republic of Congo. [196]Anglia Ruskin University, Cambridge, UK. [197]World Health Organization Country Office, Dushanbe, Tajikistan. [198]Flemish Agency for Care and Health, Brussels, Belgium. [199]Ghent University, Ghent, Belgium. [200]FrieslandCampina, Amersfoort, The Netherlands. [201]KHP Centre for Translational Medicine, London, UK. [202]Universidad Central de Venezuela, Caracas, Venezuela. [203]University of Geneva, Geneva, Switzerland. [204]Bielefeld University, Bielefeld, Germany. [205]World Health Organization Athens Quality of Care Office, Athens, Greece. [206]German Cancer Research Center, Heidelberg, Germany. [207]University of Amsterdam, Amsterdam, The Netherlands. [208]The Fred Hollows Foundation, Auckland, New Zealand. [209]University of the Andes, Mérida, Venezuela. [210]Carol Davila University of Medicine and Pharmacy, Bucharest, Romania. [211]Instituto Politécnico de Lisboa, Lisbon, Portugal. [212]Swansea University, Swansea, UK. [213]University College Copenhagen, Copenhagen, Denmark. [214]World Health Organization Regional Office for Europe, Copenhagen, Denmark. [215]University of Medicine, Tirana, Albania. [216]Munster Technological University, Cork, Ireland. [217]Universidad de La Laguna, Tenerife, Spain. [218]University of Malta, Msida, Malta. [219]University of Granada, Granada, Spain. [220]Vanderbilt University, Nashville, TN, USA. [221]Pan American Health Organization, Washington, DC, USA. [222]Ministry of Health, Tongatapu, Tonga. [223]Canadian Fitness and Lifestyle Research Institute, Ottawa, Ontario, Canada. [224]Hospital Santa Maria, Lisbon, Portugal. [225]University of Padua, Padua, Italy. [226]Istanbul University-Cerrahpasa, Istanbul, Türkiye. [227]Universidade Federal de Juiz de Fora, Juiz de Fora, Brazil. [228]National Institute of Public Health, Prague, Czechia. [229]Canopo Study Center - Salerno, Salerno, Italy. [230]University of Groningen, Groningen, The Netherlands. [231]Karolinska Institutet, Huddinge, Sweden. [232]Centro de Estudios Sobre Nutrición Infantil, Buenos Aires, Argentina. [233]Universidade Federal de Minas Gerais, Belo Horizonte, Brazil. [234]University of Zaragoza, Zaragoza, Spain. [235]Santiago de Compostela University, Santiago de Compostela, Spain. [236]University of Cádiz, Cádiz, Spain. [237]Ministry of Health, Ankara, Türkiye. [238]Council for Agricultural Research and Economics, Rome, Italy. [239]Universidade Federal do Rio Grande, Rio Grande, Brazil. [240]Chulalongkorn University, Bangkok, Thailand. [241]India Diabetes Research Foundation, Chennai, India. [242]Duke-NUS Medical School, Singapore, Singapore. [243]Federation University Australia, Ballarat, Victoria, Australia. [244]Paris Cité University, Paris, France. [245]ICMR — National Institute of Medical Statistics, New Delhi, India. [246]Capital Institute of Pediatrics, Beijing, China. [247]Xiangtan University, Xiangtan, China. [248]Ministry of Health and Welfare, Taipei, Taiwan. [249]Kailuan General Hospital, Tangshan, China. [250]University of Oxford, Oxford, UK. [251]National University of Singapore, Singapore, Singapore. [252]US Centers for Disease Control and Prevention, Atlanta, GA, USA. [253]Ahvaz Jundishapur University of Medical Sciences, Ahvaz, Iran. [254]The Gertner Institute for Epidemiology and Health Policy Research, Ramat Gan, Israel. [255]National Center of Public Health and Analyses, Sofia, Bulgaria. [256]University of Fribourg, Fribourg, Switzerland. [257]Victor Babes University of Medicine and Pharmacy, Timisoara, Romania. [258]RIPAS Hospital, Bandar Seri Begawan, Brunei. [259]University of Southern Denmark, Odense, Denmark. [260]Universidade Estadual Paulista, Presidente Prudente, Brazil. [261]Medical University of Silesia, Katowice, Poland. [262]Istituto Superiore di Sanità, Rome, Italy. [263]Charles University, Prague, Czechia. [264]Thomayer University Hospital, Prague, Czechia. [265]Ministry for Health and Active Ageing, Floriana, Malta. [266]University of Salerno, Fisciano, Italy. [267]Ministry of Health, Port of Spain, Trinidad and Tobago. [268]Statistics Canada, Ottawa, Ontario, Canada. [269]Nicolae Testemiţanu State University of Medicine and Pharmacy, Chisinau, Moldova. [270]Universidad Miguel Hernández de Elche, Alicante, Spain. [271]University of the Extreme South of Santa Catarina, São Luís, Brazil. [272]University of Southampton, Southampton, UK. [273]Pontificia Universidad Javeriana, Bogotá, Colombia. [274]University of Insubria, Varese, Italy. [275]Malawi Epidemiology and Intervention Research Unit, Lilongwe, Malawi. [276]National Nutrition Agency, Banjul, The Gambia. [277]CIBEROBN, Madrid, Spain. [278]Hungarian University of Sports Science, Budapest, Hungary. [279]University of Debrecen, Debrecen, Hungary. [280]National Institute of Public Health, Bucharest, Romania. [281]Universidade Federal da Paraíba, João Pessoa, Brazil. [282]National Research Council, Reggio Calabria, Italy. [283]Universidade Federal de Santa Catarina, Florianópolis, Brazil. [284]Universidade Federal de Alagoas, Alagoas, Brazil. [285]Eftimie Murgu University Resita, Resita, Romania. [286]Ahmadu Bello University, Zaria, Nigeria. [287]Aarhus University, Aarhus, Denmark. [288]Institut Pasteur de Lille, Lille, France. [289]Eduardo Mondlane University, Maputo, Mozambique. [290]Indian Statistical Institute, Kolkata, India. [291]Tabriz Health Services Management Research Center, Tabriz, Iran. [292]Universidade Nasionál Timór Lorosa'e, Dili, Timor-Leste. [293]Karolinska Institutet, Stockholm, Sweden. [294]Associazione Calabrese di Epatologia, Reggio Calabria, Italy. [295]Aragón Health Research Institute Foundation, Zaragoza, Spain. [296]University of Heidelberg, Heidelberg, Germany. [297]Harokopio University of Athens, Athens, Greece. [298]Innovating Health International, Port-au-Prince, Haiti. [299]French National Research Institute for Sustainable Development, Montpellier, France. [300]The National Public Health Agency, St Maurice, France. [301]Astrakhan State Medical University, Astrakhan, Russia. [302]Universidade do Vale do Rio dos Sinos, São Leopoldo, Brazil. [303]National Council of Scientific and Technical Research, Buenos Aires, Argentina. [304]Ministry of Health and Medical Education, Tehran, Iran. [305]University of Novi Sad, Novi Sad, Serbia. [306]National Institute of Nutrition, Hanoi, Viet Nam. [307]University of Queensland, Brisbane, Queensland, Australia. [308]Instituto de Investigación Nutricional, Lima, Peru. [309]Sun Yat-sen University, Guangzhou, China. [310]Capital Medical University Beijing Tongren Hospital, Beijing, China. [311]Universidad de Cuenca, Cuenca, Ecuador. [312]University of Las Palmas de Gran Canaria, Las Palmas de Gran Canaria, Spain. [313]Helmholtz Zentrum München, Munich, Germany. [314]Romanian Academy, Bucharest, Romania. [315]University Medicine Greifswald, Greifswald, Germany. [316]University Hospital Düsseldorf, Düsseldorf, Germany. [317]Medical University of Lodz, Lodz, Poland. [318]Calisia University, Kalisz, Poland. [319]University of North Carolina at Chapel Hill, Chapel Hill, NC, USA. [320]Beijing Center for Disease Prevention and Control, Beijing, China. [321]IRL 3189 ESS, Marseille, France. [322]Scuola Superiore Sant'Anna, Pisa, Italy. [323]Al-Farabi Kazakh National University, Almaty, Kazakhstan. [324]Murdoch Children's Research Institute, Parkville, Victoria, Australia. [325]Semey Medical University, Semey, Kazakhstan. [326]University of Latvia, Riga, Latvia. [327]Institute of Mother and Child, Warsaw, Poland. [328]Ministry of Health and Medical Services, Gizo, Solomon Islands. [329]Hormozgan University of Medical Sciences, Bandar Abbas, Iran. [330]University of Benin, Benin City, Nigeria. [331]University of Skövde, Skövde, Sweden. [332]Ministry of Health, Rabat, Morocco. [333]National Institute of Nutrition and Food Technology, Tunis, Tunisia. [334]University of Calabar, Calabar, Nigeria. [335]University of Haifa, Haifa, Israel. [336]University of California Davis, Davis, CA, USA. [337]Region Västernorrland, Härnösand, Sweden. [338]Stellenbosch University, Stellenbosch, South Africa. [339]Karadeniz Technical University, Trabzon, Türkiye. [340]Dokuz Eylul University, Izmir, Türkiye. [341]University of Helsinki, Helsinki, Finland. [342]Rafsanjan University of Medical Sciences, Rafsanjan, Iran. [343]Queen's University Belfast, Belfast, UK. [344]Monash University, Melbourne, Victoria, Australia. [345]Luxembourg Institute of Health, Strassen, Luxembourg. [346]Tabriz University of Medical Sciences, Tabriz, Iran. [347]Fasa University of Medical Sciences, Fasa, Iran. [348]Shiraz University of Medical Sciences, Shiraz, Iran. [349]Baqai Medical University, Karachi, Pakistan. [350]Gesundheit Österreich, Wien, Austria. [351]Centro de Salud Villanueva Norte, Badajoz, Spain. [352]Hospital Don Benito-Villanueva de la Serena, Badajoz, Spain. [353]University of Colombo, Colombo, Sri Lanka. [354]London School of Hygiene & Tropical Medicine, London, UK. [355]Ministry of Health, Buenos Aires, Argentina. [356]Universidad de Santiago de Chile, Santiago, Chile. [357]Instituto PENSI Sabara Hospital, São Paulo, Brazil. [358]University of Tartu, Tartu, Estonia. [359]Georgia College and State University, Milledgeville, GA, USA. [360]Universidade Federal de São Paulo, São Paulo, Brazil.

[361]University Medical Centre Ljubljana, Ljubljana, Slovenia. [362]University of Tasmania, Hobart, Tasmania, Australia. [363]Hospital Universitario Son Espases, Palma, Spain. [364]Hospital de Clinicas de Porto Alegre, Porto Alegre, Brazil. [365]Universidade Federal do Rio Grande do Sul, Porto Alegre, Brazil. [366]Kansai Medical University, Hirakata, Japan. [367]Kyoto University, Kyoto, Japan. [368]Ternopil National Medical University, Ternopil, Ukraine. [369]Medical University of Warsaw, Warsaw, Poland. [370]Public Health and Reforms Center, Baku, Azerbaijan. [371]Medical University of Varna, Varna, Bulgaria. [372]Ministero della Salute DG Prevenzione Sanitaria, Rome, Italy. [373]University of Catania, Catania, Italy. [374]Ministry of Public Health, Yaren, Nauru. [375]University of Deusto, Bilbao, Spain. [376]Africa Health Research Institute, Mtubatuba, South Africa. [377]Geneva University Hospitals, Geneva, Switzerland. [378]Hospital del Mar Research Institute, Barcelona, Spain. [379]Utrecht University, Utrecht, The Netherlands. [380]Wageningen University, Wageningen, The Netherlands. [381]Medical Research Foundation, Chennai, India. [382]Bayero University, Kano, Nigeria. [383]Kurdistan University of Medical Sciences, Sanandaj, Iran. [384]Endocrine Physiology Research Center, Tehran, Iran. [385]University of South Carolina, Columbia, SC, USA. [386]American University of Beirut, Beirut, Lebanon. [387]University of Adelaide, Adelaide, South Australia, Australia. [388]Ministry of Health, Mbabane, Eswatini. [389]Lund University, Lund, Sweden. [390]Aristotle University of Thessaloniki, Thessaloniki, Greece. [391]Central University of Kerala, Kasaragod, India. [392]Institut National de la Santé et de la Recherche Médicale, Paris, France. [393]McGill University, Montreal, Québec, Canada. [394]Universidad de Costa Rica, San José, Costa Rica. [395]Gasol Foundation, Barcelona, Spain. [396]University of Lleida, Sant Boi de Llobregat, Spain. [397]PASs Hirszfeld Institute of Immunology and Experimental Therapy, Wroclaw, Poland. [398]University Agostinho Neto, Luanda, Angola. [399]Kansas State University, Manhattan, KS, USA. [400]St. Ann's University Hospital, Brno, Czechia. [401]Universidad Autónoma de Santo Domingo, Santo Domingo, Dominican Republic. [402]Ministry of Health, Lisbon, Portugal. [403]Institute for Clinical and Experimental Medicine, Prague, Czechia. [404]Children's Memorial Health Institute, Warsaw, Poland. [405]University of Thessaly, Larissa, Greece. [406]Ministry of Health and Wellness, Kingston, Jamaica. [407]UiT The Arctic University of Norway, Tromsø, Norway. [408]National Center of Cardiovascular Diseases, Beijing, China. [409]International Life Science Institute, Buenos Aires, Argentina. [410]University of Ferrara, Ferrara, Italy. [411]Icelandic Heart Association, Kopavogur, Iceland. [412]French National Research Institute for Sustainable Development, Limoges, France. [413]Universidad Icesi, Cali, Colombia. [414]State University of Montes Claros, Montes Claros, Brazil. [415]King's College London, London, UK. [416]Capital Medical University, Beijing, China. [417]Healis-Sekhsaria Institute for Public Health, Navi Mumbai, India. [418]Singapore National Eye Centre, Singapore, Singapore. [419]Eternal Heart Care Centre and Research Institute, Jaipur, India. [420]University of Ibadan, Ibadan, Nigeria. [421]University of Belgrade Institute for Medical Research, Belgrade, Serbia. [422]Institute for Clinical Effectiveness and Health Policy, Buenos Aires, Argentina. [423]ISGlobal, Barcelona, Spain. [424]National Health Insurance Service, Wonju, Republic of Korea. [425]Prevention of Metabolic Disorders Research Center, Tehran, Iran. [426]Research and Education Institute of Child Health, Nicosia, Cyprus. [427]Danish Cancer Institute, Copenhagen, Denmark. [428]The University of the West Indies, Cave Hill, Barbados. [429]Kermanshah University of Medical Sciences, Kermanshah, Iran. [430]Universidade Federal de Pernambuco, Recife, Brazil. [431]National Institute of Public Health, Copenhagen, Denmark. [432]Yasuj University of Medical Sciences, Yasuj, Iran. [433]Zahedan University of Medical Sciences, Zahedan, Iran. [434]International Hellenic University, Thessaloniki, Greece. [435]Kyushu University, Fukuoka, Japan. [436]New York City, NY, USA. [437]University of Texas Southwestern Medical Center, Dallas, TX, USA. [438]National Research Institute for Health and Family Planning, Beijing, China. [439]University of Pécs, Pécs, Hungary. [440]Danish Health Authority, Copenhagen, Denmark. [441]Joep Lange Institute, Amsterdam, The Netherlands. [442]National Institute of Public Health, Cuernavaca, Mexico. [443]Universidad Autónoma de Bucaramanga, Bucaramanga, Colombia. [444]ETH Zurich, Zurich, Switzerland. [445]Chronic Diseases Research Center, Tehran, Iran. [446]University of Western Australia, Perth, Western Australia, Australia. [447]Universidade Federal do Paraná, Curitiba, Brazil. [448]University Ramon Llull, Barcelona, Spain. [449]Vrije Universiteit Amsterdam, Amsterdam, The Netherlands. [450]Kingston Health Sciences Centre, Kingston, Ontario, Canada. [451]Fundación Oftalmológica de Santander, Bucaramanga, Colombia. [452]Obesity Research Center, Tehran, Iran. [453]University Oran 1, Oran, Algeria. [454]Independent Public Health Specialist, Nay Pyi Taw, Myanmar. [455]Ministry of Health and Sports, Nay Pyi Taw, Myanmar. [456]Instituto Murciano de Investigación Biosanitaria, Murcia, Spain. [457]International Agency for Research on Cancer, Lyon, France. [458]Institute of Public Health, Tirana, Albania. [459]Giuseppe Degennaro LUM University, Casamassima, Italy. [460]University of Nigeria, Enugu, Nigeria. [461]Ministry of Health and Human Services, Koror, Palau. [462]Erasmus Medical Center Rotterdam, Rotterdam, The Netherlands. [463]Fundación para la Investigación Sanitaria y Biomédica de la Comunidad Valenciana, Valencia, Spain. [464]Marshall Islands Epidemiology & Prevention Initiatives, Majuro, Marshall Islands. [465]The University of Tokyo, Tokyo, Japan. [466]Alex Ekwueme Federal University Teaching Hospital, Abakaliki, Nigeria. [467]Texas Tech University, Dallas, TX, USA. [468]Bulgarian Academy of Sciences, Sofia, Bulgaria. [469]Hokkaido University, Tokyo, Japan. [470]Hadassah University Medical Center, Jerusalem, Israel. [471]Université Catholique de Louvain, Brussels, Belgium. [472]Slovenian Heart Foundation, Ljubljana, Slovenia. [473]Public Health Agency of Sweden, Stockholm, Sweden. [474]Norwegian University of Science and Technology, Trondheim, Norway. [475]University of Melbourne, Melbourne, Victoria, Australia. [476]Sports University of Tirana, Tirana, Albania. [477]University of Graz, Graz, Austria. [478]Jagiellonian University Medical College, Kraków, Poland. [479]University of Rijeka, Rijeka, Croatia. [480]Heart Foundation, Melbourne, Victoria, Australia. [481]University of Bisha, Bisha, Saudi Arabia. [482]Universidad Eugenio Maria de Hostos, Santo Domingo, Dominican Republic. [483]University of Duisburg-Essen, Essen, Germany. [484]Rothschild Foundation Hospital, Paris, France. [485]University of New South Wales, Sydney, New South Wales, Australia. [486]University Hospital Centre Sestre Milosrdnice, Zagreb, Croatia. [487]Guilan University of Medical Sciences, Rasht, Iran. [488]University of Opole, Opole, Poland. [489]Gulu University, Gulu, Uganda. [490]Capacity Development in Nutrition CAPNUTRA, Belgrade, Serbia. [491]University of Crete, Heraklion, Greece. [492]Hungarian School Sport Federation, Budapest, Hungary. [493]Mbarara University of Science and Technology, Mbarara, Uganda. [494]National Center for Disease Control and Public Health, Tbilisi, Georgia. [495]Tbilisi State Medical University, Tbilisi, Georgia. [496]Ministry of Health, Bratislava, Slovakia. [497]Reichman University, Herzliya, Israel. [498]Sri Venkateswara University, Tirupati, India. [499]Sree Chitra Tirunal Institute for Medical Sciences and Technology, Trivandrum, India. [500]Hellenic Medical Association for Obesity, Athens, Greece. [501]Maharajgunj Medical Campus, Kathmandu, Nepal. [502]Duke University, Durham, NC, USA. [503]University of Bouaké, Bouaké, Côte d'Ivoire. [504]Université Officielle de Bukavu, Bukavu, Democratic Republic of Congo. [505]Pennington Biomedical Research Center, Baton Rouge, LA, USA. [506]National Institute of Epidemiology, Chennai, India. [507]UNICEF, Lilongwe, Malawi. [508]University of Toronto, Toronto, Ontario, Canada. [509]Shanghai University of Sport, Shanghai, China. [510]University of Münster, Münster, Germany. [511]Israel Center for Disease Control, Ramat Gan, Israel. [512]Child Growth and Development Research Center, Isfahan, Iran. [513]Mashhad University of Medical Sciences, Mashhad, Iran. [514]Research Institute of Child Nutrition, Dortmund, Germany. [515]Ministry of Health, Gaborone, Botswana. [516]Research Institute for Endocrine Sciences, Tehran, Iran. [517]Mazandaran University of Medical Sciences, Sari, Iran. [518]Hypertension Research Center, Isfahan, Iran. [519]Ministry of Public Health, Kabul, Afghanistan. [520]Medical University of Innsbruck, Innsbruck, Austria. [521]VASCage Research Centre on Vascular Ageing and Stroke, Innsbruck, Austria. [522]Muhimbili University of Health and Allied Sciences, Dar es Salaam, Tanzania. [523]Yonsei University College of Medicine, Seoul, Republic of Korea. [524]Sahlgrenska University Hospital, Gothenburg, Sweden. [525]University College South Denmark, Haderslev, Denmark. [526]Thaksin University, Songkhla, Thailand. [527]Masaryk University, Brno, Czechia. [528]Ulm University Hospital, Ulm, Germany. [529]B P Koirala Institute of Health Sciences, Dharan, Nepal. [530]Nepal Development Society, Kathmandu, Nepal. [531]International Committee of Red Cross, Maiduguri, Nigeria. [532]University of Vienna, Vienna, Austria. [533]Tartu University Clinics, Tartu, Estonia. [534]National Institute of Public Health, Abidjan, Côte d'Ivoire. [535]Hildburghausen District Department of State Public Health Service, Hildburghausen, Germany. [536]Pontificia Universidad Católica Argentina, Buenos Aires, Argentina. [537]Ministry of Health and Wellness, Port Louis, Mauritius. [538]University Hospital Ulm, Ulm, Germany. [539]Croatian Institute of Public Health, Zagreb, Croatia. [540]Institute of Nutrition of Central America and Panama, Guatemala City, Guatemala. [541]North-West University, Potchefstroom, South Africa. [542]South African Medical Research Council, Cape Town, South Africa. [543]University of Physical Culture in Kraków, Kraków, Poland. [544]University of Jyväskylä, Jyväskylä, Finland. [545]Research Institute of Cardiology and Internal Diseases, Almaty, Kazakhstan. [546]ICMR — National Centre for Disease Informatics and Research, Bengaluru, India. [547]Amrita Institute of Medical Sciences, Cochin, India. [548]Institute of Endocrinology, Prague, Czechia. [549]All India Institute of Medical Sciences, New Delhi, India. [550]Hanoi University of Public Health, Hanoi, Viet Nam. [551]Hassan I University, Settat, Morocco. [552]University of Limpopo, Polokwane, South Africa. [553]Public University of Navarra, Pamplona, Spain. [554]Universiti Brunei Darussalam, Bandar Seri Begawan, Brunei. [555]Ministry of Health, Algiers, Algeria. [556]Ministry of Health, Georgetown, Guyana. [557]Oulu Deaconess Institute Foundation, Oulu, Finland. [558]Sahlgrenska Academy, Gothenburg, Sweden. [559]Endocrinology and Metabolism Research Center, Tehran, Iran. [560]University of Public Health, Yangon, Myanmar. [561]Centro Studi Epidemiologici di Gubbio, Gubbio, Italy. [562]Universidad Peruana Cayetano Heredia, Lima, Peru. [563]National University Health System, Singapore, Singapore. [564]Tampere University Hospital, Tampere, Finland. [565]Tampere University, Tampere, Finland. [566]University of Douala, Douala, Cameroon. [567]University of the Basque Country, Leioa, Spain. [568]University of the Basque Country, Donostia-San Sebastian, Spain. [569]The Russian University of Sport, Moscow, Russia. [570]National Institute for Health Development, Tallinn, Estonia. [571]West Virginia University, Morgantown, WV, USA. [572]Tan Tock Seng Hospital, Singapore, Singapore. [573]Fundação Oswaldo Cruz, Belo Horizonte, Brazil. [574]Uppsala University, Uppsala, Sweden. [575]University of Augsburg, Augsburg, Germany. [576]Capital Medical University Beijing An Zhen Hospital, Beijing, China. [577]Zhengzhou University, Zhengzhou, China. [578]FISABIO-Universitat Jaume I-Universitat de València, Valencia, Spain. [579]Taipei Medical University, Taipei, Taiwan. [580]Ministry of Health, Colombo, Sri Lanka. [581]Children's Hospital of Eastern Ontario Research Institute, Ottawa, Ontario, Canada. [582]Sports Medical Center of Minho, Braga, Portugal. [583]Universidad Espíritu Santo, Samborondón, Ecuador. [584]Universidad de Los Lagos, Osorno, Chile. [585]Universidad San Martín de Porres, Lima, Peru. [586]Regional Ministry of Health, Valladolid, Spain. [587]Universidade Federal de Ciências da Saúde de Porto Alegre, Porto Alegre, Brazil. [588]Ministerio de Salud Pública, Quito, Ecuador. [589]Ilembula Lutheran Hospital, Ilembula, Tanzania. [590]University of Kinshasa Hospital, Kinshasa, Democratic Republic of Congo. [591]University of Coimbra, Coimbra, Portugal. [592]UMR 7268 ADES, Marseille, France. [593]Coimbra University Hospital Center, Coimbra, Portugal. [594]University of Texas Rio Grande Valley, Harlingen, TX, USA. [595]Institute of Neuroscience of the National Research Council, Padua, Italy. [596]Baker Heart and Diabetes Institute, Melbourne, Victoria, Australia. [597]Agricultural University of Athens, Athens, Greece. [598]Academia VBHC, São Paulo, Brazil. [599]SB RAS Federal Research Center Institute of Cytology and Genetics, Novosibirsk, Russia. [600]University of Otago, Dunedin, New Zealand. [601]University of Northern British Columbia, Prince George, British Columbia, Canada. [602]Hellenic Mediterranean University, Siteia, Greece. [603]Loughborough University, Loughborough, UK. [604]University of Split, Split, Croatia. [605]Ministry of Health, Nicosia, Cyprus. [606]Lausanne University Hospital, Lausanne, Switzerland. [607]University of Lausanne, Lausanne, Switzerland. [608]Fundação Oswaldo Cruz, Rio de Janeiro, Brazil. [609]CIBERCV, Madrid, Spain. [610]Hasselt University, Hasselt, Belgium. [611]Mary Immaculate College, Limerick, Ireland. [612]Universidad de Castilla-La Mancha, Cuenca, Spain. [613]Hungarian Society of Sports Medicine, Budapest, Hungary. [614]Nnamdi Azikiwe University, Awka, Nigeria. [615]Paracelsus Medical University, Salzburg, Austria. [616]Institute for Cancer Research, Prevention and Clinical Network, Florence, Italy. [617]Universidade Estadual do Centro-Oeste, Guarapuava, Brazil. [618]Université Catholique de Bukavu, Bukavu, Democratic Republic of Congo. [619]Sefako Makgatho Health Sciences University, Pretoria, South Africa. [620]Centro de Estudos do Laboratório de Aptidão Física de São Caetano do Sul, São Paulo, Brazil. [621]George Institute for Global Health, New Delhi, India. [622]Brown University, Providence, RI, USA. [623]Shahid Sadoughi University of Medical Sciences, Yazd, Iran. [624]Arabkir Medical Centre — Institute of Child and Adolescent Health, Yerevan, Armenia. [625]Pontificia Universidad Católica de Valparaíso, Valparaíso, Chile. [626]Center for Studies and Research on Social Dynamics and Health, Lisbon, Portugal. [627]Universidad de los Andes, Bogotá, Colombia. [628]University of Abidjan, Abidjan, Côte d'Ivoire. [629]Pirogov Russian National Research Medical University, Moscow, Russia. [630]Universidade de Lisboa, Lisbon, Portugal. [631]Saveetha Dental College and Hospitals, Chennai, India. [632]Democritus University, Alexandroupolis, Greece. [633]Nutrition and Endocrine Research Center, Tehran, Iran. [634]Grigore T Popa University of Medicine and Pharmacy, Iasi, Romania. [635]Instituto Nacional de Saúde, Marracuene, Mozambique. [636]Università degli Studi di Firenze, Florence, Italy. [637]Psychiatry and Psychology Research Center, Tehran, Iran. [638]Isfahan Cardiovascular Research Center, Isfahan, Iran. [639]Universiti Putra Malaysia, Serdang, Malaysia. [640]University Medical School of Pécs, Pécs, Hungary. [641]Mulago Hospital, Kampala, Uganda. [642]Instituto Costarricense de Investigación y Enseñanza en Nutrición y Salud, San José, Costa Rica. [643]Universidad de San

Carlos, Guatemala City, Guatemala. [644]Gorgas Memorial Institute for Studies of Health, Panama City, Panama. [645]University of Limpopo, Sovenga, South Africa. [646]Seoul National University, Seoul, Republic of Korea. [647]University of Medical Sciences of Cienfuegos, Cienfuegos, Cuba. [648]Ministry of Health and Wellness, Belmopan, Belize. [649]La Trobe University, Melbourne, Victoria, Australia. [650]Sabzevar University of Medical Sciences, Sabzevar, Iran. [651]International Institute of Molecular and Cell Biology, Warsaw, Poland. [652]Ain Shams University, Cairo, Egypt. [653]University of Medicine and Pharmacy Craiova, Craiova, Romania. [654]Instituto Conmemorativo Gorgas de Estudios de la Salud, Panama City, Panama. [655]World Health Organization Country Office, Lilongwe, Malawi. [656]Universidade de São Paulo, São Paulo, Brazil. [657]Department of Public Health, Nay Pyi Taw, Myanmar. [658]Albanian Sport Science Association, Tirana, Albania. [659]University of Brescia, Brescia, Italy. [660]Queen Mary University, London, UK. [661]University of the Balearic Islands, Palma de Mallorca, Spain. [662]University of Limerick, Limerick, Ireland. [663]Universiti Sains Malaysia, Kelantan, Malaysia. [664]Makerere University School of Public Health, Kampala, Uganda. [665]Bushehr University of Medical Sciences, Bushehr, Iran. [666]Ulm University, Ulm, Germany. [667]Jozef Pilsudski University of Physical Education in Warsaw, Warsaw, Poland. [668]Suraj Eye Institute, Nagpur, India. [669]UNICEF, Yaoundé, Cameroon. [670]Ministry of Health, Apia, Samoa. [671]Leuven University, Leuven, Belgium. [672]Public Health Center of the Ministry of Health, Kyiv, Ukraine. [673]National Institute of Hygiene and Epidemiology, Hanoi, Viet Nam. [674]University of Medicine and Pharmacy at Ho Chi Minh City, Ho Chi Minh City, Viet Nam. [675]Hanoi Medical University, Hanoi, Viet Nam. [676]Xi'an Jiaotong University, Xi'an, China. [677]Precision Care Clinic Corp, St. Cloud, FL, USA. [678]Heartfile, Islamabad, Pakistan. [679]Eastern Mediterranean Public Health Network, Amman, Jordan. [680]University of Manchester, Manchester, UK. [681]Tachikawa General Hospital, Nagaoka, Japan. [682]University of Abuja College of Health Sciences, Abuja, Nigeria. [683]Korea Disease Control and Prevention Agency, Cheongju-si, Republic of Korea. [684]Kyoto Prefectural University of Medicine, Kyoto, Japan. [685]Japan Wildlife Research Center, Tokyo, Japan. [686]Gadarif University, Gadarif, Sudan. [687]Istanbul University, Istanbul, Türkiye. [688]Ministry of Health, Bandar Seri Begawan, Brunei. [689]University of Madeira, Funchal, Portugal. [690]University of Puerto Rico, San Juan, Puerto Rico. [691]Universidad de Santander, Bucaramanga, Colombia. [692]Kwame Nkrumah University of Science and Technology, Kumasi, Ghana. [693]University of Maiduguri, Maiduguri, Nigeria. [694]Arizona State University, Tempe, AZ, USA. [695]University of Wisconsin-Madison, Madison, WI, USA. [696]University Hospital Heidelberg, Heidelberg, Germany. [697]Qingdao Centers for Disease Control and Prevention, Qingdao, China. [698]University of Bari Aldo Moro, Bari, Italy. [699]University of Sharjah, Abu Dhabi, United Arab Emirates. [700]Catholic University of Daegu, Daegu, Republic of Korea. [701]Faculty of Medicine Ramathibodi Hospital, Maidol, Thailand. [702]Jivandeep Hospital, Anand, India. [703]Centro de Investigação em Saúde de Angola, Caxito, Angola. [704]Fundació Institut Universitari per a la recerca a l'Atenció Primària de Salut Jordi Gol i Gurina, Mataró, Spain. [705]National Dental Centre Singapore, Singapore, Singapore. [706]University Hospital of Varese, Varese, Italy. [707]Vietnam National Heart Institute, Hanoi, Viet Nam. [708]Ministry of Health Malawi, Lilongwe, Malawi. [709]Clínica de Medicina Avanzada Dr. Abel González, Santo Domingo, Dominican Republic. [710]Université de Limoges, Limoges, France. [711]Cardiovascular Prevention Centre Udine, Udine, Italy. [712]University of Pisa, Pisa, Italy. [713]Ministry of Health and Medical Services, Honiara, Solomon Islands. [714]National Institute of Cardiology, Warsaw, Poland. [715]College of Physicians of Barcelona, Barcelona, Spain. [716]O.M. Marzieiev Institute for Public Health of the National Academy of the Medical Sciences of Ukraine, Kyiv, Ukraine. [717]Universiti Kebangsaan Malaysia, Kuala Lumpur, Malaysia. [718]Ardabil University of Medical Sciences, Ardabil, Iran. [719]Public Health Foundation of India, New Delhi, India. [720]University of Edinburgh, Edinburgh, UK. [721]Universidade Pedagógica, Maputo, Mozambique. [722]Centre for Disease Prevention and Control, Riga, Latvia. [723]Sulaimani Polytechnic University, Sulaymaniyah, Iraq. [724]Boehringer Ingelheim, Ingelheim, Germany. [725]Alborz University of Medical Sciences, Karaj, Iran. [726]Cooperativa de Ensino Superior Politécnico e Universitário, Famalicão, Portugal. [727]Ministry of Health, Hanoi, Viet Nam. [728]National Agency for Public Health, Chisinau, Moldova. [729]University of Novi Sad Faculty of Medicine, Novi Sad, Serbia. [730]Pure Earth, Dhaka, Bangladesh. [731]Institute of Epidemiology Disease Control and Research, Dhaka, Bangladesh. [732]University of Turku, Turku, Finland. [733]UNICEF, Baku, Azerbaijan. [734]World Health Organization Country Office, Juba, South Sudan. [735]Instituto Federal Riograndense, Rio Grande, Brazil. [736]Institut Universitari d'Investigació en Atenció Primària Jordi Gol, Girona, Spain. [737]University of Girona, Girona, Spain. [738]Newcastle University, Newcastle, UK. [739]Institute for Health Policy, Colombo, Sri Lanka. [740]APDP Diabetes Portugal, Lisboa, Portugal. [741]Sotiria Hospital, Athens, Greece. [742]INCLIVA, Valencia, Spain. [743]Slovak Academy of Sciences, Bratislava, Slovakia. [744]University of Santa Cruz do Sul, Santa Cruz do Sul, Brazil. [745]Nutrition Research Foundation, Barcelona, Spain. [746]Minas Gerais State Secretariat for Health, Belo Horizonte, Brazil. [747]UB Bunyola Ibsalut, Palma, Spain. [748]Amsterdam Institute for Global Health and Development, Amsterdam, The Netherlands. [749]Universidade Nove de Julho, São Paulo, Brazil. [750]Ministerio de Salud, Panama City, Panama. [751]Canarian Health Service, Tenerife, Spain. [752]Universidad Industrial de Santander, Bucaramanga, Colombia. [753]Ministry of Health and Social Protection, Bogotá, Colombia. [754]Wuqu' Kawoq, Tecpan, Guatemala. [755]GroundWork, Fläsch, Switzerland. [756]CIBERDEM, Málaga, Spain. [757]Instituto de Investigación Biomédica de Málaga-Plataforma Bionand, Málaga, Spain. [758]Fundación Instituto de Investigación Sanitaria Illes Baleares, Madrid, Spain. [759]University of Minho, Braga, Portugal. [760]Fiji National University, Suva, Fiji. [761]GHESKIO Clinics, Port-au-Prince, Haiti. [762]Universidad Galileo, Guatemala City, Guatemala. [763]Universidad de San Carlos, Quetzaltenango, Guatemala. [764]National Center of Epidemiology ISCIII, Madrid, Spain. [765]Rehamed-Center, Tajęcina, Poland. [766]Institute of Food Sciences of the National Research Council, Avellino, Italy. [767]Medical University of Gdańsk, Gdańsk, Poland. [768]Singapore Eye Research Institute, Singapore, Singapore. [769]Sitaram Bhartia Institute of Science and Research, New Delhi, India. [770]Academy of Preventive Medicine, Almaty, Kazakhstan. [771]Kindergarten of Avlonari, Evia, Greece. [772]National Institute of Health, Lima, Peru. [773]Ministry of Health, Jakarta, Indonesia. [774]Catalan Department of Health, Barcelona, Spain. [775]Biogipuzkoa Health Research Institute, San Sebastián, Spain. [776]Instituto de Saúde Ambiental, Lisbon, Portugal. [777]Universidade Federal de Alagoas, Maceió, Brazil. [778]The University of Health Sciences, Phnom Pen, Cambodia. [779]Wellbeing Services County of South Karelia, Lappeenranta, Finland. [780]Shizuoka Graduate University of Public Health, Shizuoka, Japan. [781]National Cancer Center, Tokyo, Japan. [782]University of São Paulo Clinics Hospital, São Paulo, Brazil. [783]Hospital Italiano de Buenos Aires, Buenos Aires, Argentina. [784]Medical University of Vienna, Vienna, Austria. [785]Nes Municipality, Nes, Norway. [786]German Institute of Human Nutrition Potsdam-Rehbruecke, Nuthetal, Germany. [787]The George Institute for Global Health, Sydney, New South Wales, Australia. [788]Center for Oral Health Services and Research Mid-Norway, Trondheim, Norway. [789]Lagos State University College of Medicine, Lagos, Nigeria. [790]Comenius University, Bratislava, Slovakia. [791]Human Sciences Research Council, Cape Town, South Africa. [792]Medical Excellence JAPAN, Tokyo, Japan. [793]Teikyo University, Tokyo, Japan. [794]Israel Defense Forces Medical Corps, Jerusalem, Israel. [795]Rutgers University, New Brunswick, NJ, USA. [796]Africa Health Research Institute, Durban, South Africa. [797]University of KwaZulu-Natal, Durban, South Africa. [798]St Vincent's Hospital, Sydney, New South Wales, Australia. [799]RAS FCTAS Institute of Social Demography, Moscow, Russia. [800]Health Polytechnic Jakarta II Institute, Jakarta, Indonesia. [801]Diponegoro University, Semarang, Indonesia. [802]National Medical Research Center for Children's Health, Moscow, Russia. [803]University of Bordeaux, Bordeaux, France. [804]University of Hohenheim, Stuttgart, Germany. [805]Oslo Metropolitan University, Oslo, Norway. [806]Institute of Public Health, Skopje, North Macedonia. [807]Ss. Cyril and Methodius University, Skopje, North Macedonia. [808]Lamprecht and Stamm Sozialforschung und Beratung AG, Zurich, Switzerland. [809]Clinical Emergency Hospital, Bucharest, Romania. [810]Bonn University, Bonn, Germany. [811]National Institute of Public Health NIH — National Research Institute, Warsaw, Poland. [812]Kalina Malina Kindergarten, Pazardjik, Bulgaria. [813]Pontificia Universidad Javeriana Seccional Cali, Cali, Colombia. [814]Ubon Ratchathani University, Ubon Ratchathani, Thailand. [815]The Jikei University School of Medicine, Tokyo, Japan. [816]The Chinese University of Hong Kong, Hong Kong, China. [817]Mongolian Ecotoxicology Center, Ulaanbaatar, Mongolia. [818]University of Jordan, Amman, Jordan. [819]National Statistical Office, Praia, Cabo Verde. [820]St. Olavs University Hospital, Trondheim, Norway. [821]South Kazakhstan Medical Academy, Shymkent, Kazakhstan. [822]All Institute of Medical Sciences, New Delhi, India. [823]Scientific Research Institute of Maternal and Child Health, Ashgabat, Turkmenistan. [824]Ministry of Health, Amman, Jordan. [825]UNICEF, Niamey, Niger. [826]University of Applied Sciences Utrecht, Utrecht, The Netherlands. [827]Reproductive Endocrinology Research Center, Tehran, Iran. [828]Ethiopian Public Health Institute, Addis Ababa, Ethiopia. [829]Amrita Vishwa Vidyapeetham, Kochi, India. [830]CMC Vellore, Vellore, India. [831]University Medical Center Utrecht, Utrecht, The Netherlands. [832]Institut d'Investigacio Sanitaria Illes Balears, Menorca, Spain. [833]University of Vic-Central University of Catalonia, Vic, Spain. [834]University of Bologna, Bologna, Italy. [835]Hellenic Health Foundation, Athens, Greece. [836]Government Medical College, Bhavnagar, India. [837]Massachusetts General Hospital, Boston, MA, USA. [838]Sefako Makgatho Health Sciences University, Ga-Rankuwa, South Africa. [839]Alexander Technological Educational Institute, Thessaloniki, Greece. [840]Tafuna Family Health Center, Tafuna, American Samoa. [841]LBJ Hospital, Faga'alu, American Samoa. [842]Addis Ababa University, Addis Ababa, Ethiopia. [843]Ministry of Health, Wellington, New Zealand. [844]Sheba Medical Center, Tel HaShomer, Israel. [845]Everyage Carolina Senior Care, Lexington, NC, USA. [846]Universidad Centro-Occidental Lisandro Alvarado, Barquisimeto, Venezuela. [847]Meharry Medical College, Nashville, TN, USA. [848]University of Tampere Tays Eye Center, Tampere, Finland. [849]Sabiha Gokcen Ilkokulu, Ankara, Türkiye. [850]Polytechnic Institute of Porto, Porto, Portugal. [851]Voronezh N.N. Burdenko State Medical University, Voronezh, Russia. [852]Icahn School of Medicine at Mount Sinai, New York City, NY, USA. [853]George Washington University, Washington, DC, USA. [854]MONICA FRIULI Study Group, Udine, Italy. [855]Universidad CEU San Pablo, Madrid, Spain. [856]Consejería de Sanidad Junta de Castilla y León, Valladolid, Spain. [857]Strasbourg University Hospital, Strasbourg, France. [858]National Research Council, Pisa, Italy. [859]Universidad San Francisco de Quito, Quito, Ecuador. [860]Ministry of Public Health, Nonthaburi, Thailand. [861]Sunflower Nursery School, Craiova, Romania. [862]North Karelia Center for Public Health, Joensuu, Finland. [863]Academy of Athens, Athens, Greece. [864]University of the Witwatersrand, Johannesburg, South Africa. [865]Institute for Medical Research, Kuala Lumpur, Malaysia. [866]Xinjiang Medical University, Urumqi, China. [867]Shanghai Educational Development Co. Ltd, Shanghai, China. [868]University of Cambridge, Cambridge, UK. [869]Mingsii Co. Ltd, Beijing, China. [870]Örebro University, Örebro, Sweden. [871]St George's, University of London, London, UK. [872]Universitas Indonesia, Jakarta, Indonesia. [873]GroundWork, Geneva, Switzerland. [874]National Yang Ming Chiao Tung University, Taipei, Taiwan. [875]Institute of Food and Nutrition Development of Ministry of Agriculture and Rural Affairs, Beijing, China. [876]Beijing Institute of Ophthalmology, Beijing, China. [877]Weill Cornell Medicine, New York City, NY, USA. [878]Children's Hospital of Fudan University, Shanghai, China. [879]University of Cyprus, Nicosia, Cyprus. [880]Niigata University, Niigata, Japan. [881]South China Institute of Environmental Sciences, Guangzhou, China. [882]International Medical University, Shah Alam, Malaysia. [883]Hellenic Mediterranean University, Heraklion, Greece. [884]Iran University of Medical Sciences, Tehran, Iran. [885]Peking University First Hospital, Beijing, China. [886]West Kazakhstan Medical University, Aktobe, Kazakhstan. [887]Inner Mongolia Medical University, Hohhot, China. [888]Institute of Correctional Pedagogy, Moscow, Russia. [889]Przedszkole No. 81, Warsaw, Poland. [890]These authors contributed equally: Bin Zhou, Nowell H. Phelps, Agnese Galeazzi, Olivia N. O'Driscoll. [891]Deceased: Adroaldo Cesar Araujo Gaya, Konrad Jamrozik, Suntara Klanarong, Ryutaro Ohtsuka, Altan Onat, Robespierre Ribeiro, Herman Schargrodsky, Michael Sjöström, Agustinus Soemantri, Jutta Stieber. [✉]e-mail: majid.ezzati@imperial.ac.uk

## Methods

Our analytical aim was to quantify and characterize the dynamics of how obesity has changed over time, building on studies that reported the extent of change over long multi-decade periods[1]. As a quantitative measure of the dynamics of obesity, we calculated the velocity of obesity as the rate of absolute change in prevalence between consecutive years. This metric allows understanding whether the rise in the prevalence of obesity has been uniform over time, or if its pace has changed, including acceleration, deceleration, plateauing and reversal. In addition, we used clustering to categorize the national trajectories of obesity prevalence based on their shape. The input to both analyses was prevalence of obesity in 200 countries from 1980 to 2024, a period during which obesity was recognized as an epidemic[2,3,60]. To estimate prevalence, we pooled population-based studies with measurements of height and weight. Pooled data were analysed using a Bayesian hierarchical meta-regression model. The posterior estimates were then used for calculating velocity and clustering.

Our analyses addressed the dynamics of obesity in school-aged children and adolescents 5–19 years of age and in adults 20 years of age and older. Our primary outcome was the prevalence of obesity, defined as BMI $\geq 30$ kg m$^{-2}$ for adults 20 years of age and older and as BMI $> 2$ s.d. above the median of the WHO growth reference for children and adolescents 5–19 years of age[61,62]. Following previous work[1,10,63], we conducted separate analyses for children and adolescents and for adults, because different cut-offs are used to measure obesity in the two groups[61,62,64].

### Data access and data inclusion

We pooled population-based studies with measurements of height and weight in samples of the general population from a database collated by the NCD Risk Factor Collaboration (NCD-RisC). Data were obtained from publicly available multi-country and national measurement surveys (for example, Demographic and Health Surveys, WHO STEPwise approach to Surveillance (STEPS) surveys, and those identified via the Inter-University Consortium for Political and Social Research, European Health Interview & Health Examination Surveys Database and the UK Data Service). With the help of the WHO and its regional and country offices, we identified and accessed population-based survey data from national health and statistical agencies. We searched and reviewed published studies as previously detailed[63,65–67] and invited eligible studies to join NCD-RisC, as we did with data holders from earlier pooled analyses of cardiometabolic risk factors[68–71]. The NCD-RisC database is continuously updated through all the above routes as well as through periodic requests to NCD-RisC members to suggest additional sources in their countries.

We carefully checked that each study met our inclusion criteria, which are listed below. All NCD-RisC members were also periodically asked to review the list of sources from their country, to verify that they met the inclusion criteria and were not duplicates. Potential duplicate data sources were first identified by comparing studies from the same country and year, followed by checking with NCD-RisC members who had provided data whether sources from the same country and year, and with similar sample sizes and age ranges, were the same or distinct. If two sources were confirmed as duplicates, one was discarded.

For each study, we recorded the study population, the sampling approach, the years of measurement and measurement methods. Only data that were from samples of the general population were included. All data were assessed and classified by whether they covered the whole country, one or more subnational regions (that is, one or more provinces or states, more than three cities, or more than five rural communities), or one or a small number of communities (limited geographical scope not meeting above national or subnational criteria). As stated in statistical methods, these study-level attributes were included in the Bayesian hierarchical meta-regression model so the modelling was informed by all available data, but accounted for the aforementioned differences in the populations from which different studies had sampled. All submitted data were checked by at least two people independently. Questions and clarifications were discussed with NCD-RisC members and resolved before data were incorporated into the database.

Data were included if the following criteria were met: measured data on height and weight were available; study participants were 5 years of age and older; data were collected using a probabilistic sampling method with a defined sampling frame; data were from population samples at the national, subnational or community level as defined above; and data were from the countries listed in Supplementary Table 1.

We excluded all studies that were solely based on self-reported height and weight, without any measurement, because these data are subject to biases that vary by geography, time, age, sex and socioeconomic characteristics[72–74]. Owing to these variations, approaches to correcting self-reported data may leave residual bias. We excluded data sources on population subgroups whose anthropometric status may differ systematically from the general population, including studies that had included or excluded people based on their health status; and female individuals 15–19 years of age in surveys that sampled only ever-married women or measured height and weight only among mothers. We excluded studies whose participants were only from specific educational, occupational, socioeconomic or ethnic subgroups of the general population, with the exceptions of school-based studies in countries and age–sex groups with school enrolment of 80% or higher. We also excluded studies that recruited participants through contact with health facilities; the exceptions to this exclusion criterion were studies whose sampling frame was health insurance schemes whose membership is not based on occupation or socioeconomic status in countries where at least 80% of the population were insured, and studies based on the primary-care system in high-income and Central European countries with universal insurance, as contact with the primary-care systems in these countries tends to be as good as or better than response rates for population-based surveys.

### Data cleaning and management

We excluded participants whose age was younger than 18 years if their data were not reported by single year of age (less than 0.01% of all participants), because the age associations of height and weight may be non-linear in these ages, especially during growth spurts. We excluded BMI data for female individuals who were pregnant at the time of measurement (0.33% of participants), because weight changes during pregnancy. We excluded 0.23% of participants with recorded values outside of the following predefined ranges: recorded height below 60 cm or above 180 cm for those younger than 10 years of age; below 80 cm or above 200 cm for those 10–14 years of age; and below 100 cm or above 250 cm for those 15 years of age or older; recorded weight below 5 kg or above 90 kg for those younger than 10 years of age; below 8 kg or above 150 kg for those 10–14 years of age; and below 12 kg or above 300 kg for those 15 years of age or older; or recorded BMI below 6 kg m$^{-2}$ or above 40 kg m$^{-2}$ for those younger than 10 years of age; below 8 kg m$^{-2}$ or above 60 kg m$^{-2}$ for those 10–14 years of age; and below 10 kg m$^{-2}$ or above 80 kg m$^{-2}$ for those 15 years of age or older. As in previous uses of these data[1,9,10,56,75], we excluded these participants because values outside these ranges were likely to reflect measurement or data recording errors.

Anonymized individual data from the studies from 1980 to 2024 in the NCD-RisC database were reanalysed according to a common protocol. We calculated prevalence in the following BMI ranges: for children and adolescents, the prevalence of BMI less than −2 s.d., −2 s.d. to less than −1 s.d., −1 s.d. to 1 s.d., more than 1 s.d. to 2 s.d., and more than 2 s.d. from the median of the WHO growth reference[61]; for adults, the prevalence of BMI less than 18.5 kg m$^{-2}$, 18.5 kg m$^{-2}$ to less than 20 kg m$^{-2}$, 20 kg m$^{-2}$ to less than 25 kg m$^{-2}$, 25 kg m$^{-2}$ to less than 30 kg m$^{-2}$, 30 kg m$^{-2}$ to less than 35 kg m$^{-2}$, 35 kg m$^{-2}$ to less than 40 kg m$^{-2}$, and 40 kg m$^{-2}$ or higher.

Of the studies with BMI data, 79% were included in the NCD-RisC database as individual participant data; another 14% were provided as summary statistics, that is, age–sex-specific prevalence of relevant BMI categories. When summary statistics were prepared by study investigators, detailed instructions were provided, as was computer code when requested, to ensure analysis was conducted according to the study protocol. The cut-offs for calculating prevalence in the BMI categories for school-aged children and adolescents were all age-specific and sex-specific and were applied to data in single years of age. All analyses incorporated sample weights and complex survey design, when applicable, in calculating summary statistics. Information on survey design and sample weights were provided by participating studies. For studies that used multistage (stratified) sampling, we accounted for survey design features when calculating standard errors, including clusters, strata and sample weights, using Taylor series linearization as implemented in the R package 'survey' (v4.4.2)[76].

We used two additional types of studies, accounting for 7% of all studies. First, we included some data from a previous pooling analysis[68]. We invited these studies to join NCD-RisC, as stated above. However, data from some studies were no longer available, for example, because the authors had retired or moved, data had been permanently archived or data were stored using older storage technologies that could not be easily retrieved. Second, summary statistics for nationally representative data from sources that were identified but not accessed via the above routes were extracted from published reports. Data were also extracted for two STEPS surveys that were not publicly available[77,78]. The two additional types of studies made up 0.7% of our data points for children and adolescents and 8.4% for adults (a data point is an age–sex–study-specific prevalence in a BMI category, which is used in the Bayesian meta-regression model as described below to make estimates for all age groups, countries and years). These studies had information on mean BMI and/or on a subset of BMI categories that were analysed in this work. To enable us to use these data, we used previously validated conversion regressions to estimate the missing primary outcome from the available BMI metric (or metrics). Additional details on conversion regression model specifications and the model coefficients are reported on GitHub (https://github.com/NCD-RisC/ncdrisc-methods/blob/main/NCD-RisC-conversion-model-for-prev-bmi.pdf).

After the data access and cleaning procedure described above, we used 4,050 population-based studies that measured height and weight in 232 million participants 5 years of age and older from 197 countries in this study. The data included 2,582 studies for children and adolescents from 189 countries, and 2,980 studies for adults from 196 countries. We had at least one study for 197 (99%) of the 200 countries for which estimates were made (Supplementary Fig. 1), at least two studies for 188 countries (94%) and at least three studies for 177 countries (89%). 189 countries had at least one national study, of which 181 had at least two national studies and 166 had at least three national studies. Countries in the high-income western super-region (with an average of 53.3 studies per country) and the East and Southeast Asia super-region (with an average of 33.5 studies per country) had the most data, and those in Pacific Island nations (6.0 studies per country) and sub-Saharan Africa (8.6 studies per country) had the least data (Supplementary Figs. 1 and 2). Other super-regions on average had 12.1–26.7 studies per country. Details of the studies are provided in Supplementary Table 2.

## Statistical model

**Overview.** We used a Bayesian hierarchical meta-regression model to estimate trends in the prevalence of different BMI categories by sex, age, country and year from 1980 to 2024. The statistical methods for analysis of pooled data, including its implementation and computation, are described in detail in a statistical paper[79] and related substantive papers[1,9,10,56,71,80,81]. Model specification is summarized here and described using statistical notation in the sections below. In summary, the model had a hierarchical structure, in which countries were nested in regions, which were nested in super-regions, which were nested in the globe (Supplementary Table 1). Estimates for each country and year were informed by its own data, if available, and by data from other years in the same country and from other countries, especially those in the same region with data for similar time periods. The extent to which estimates for each country-year were influenced by data from other years and other countries depended on whether the country had data, the sample size of data, whether the sources were at national, subnational or community level, and the within-country and within-region variability of the available data. The model incorporated non-linear time trends through the combination of linear and second-order random walk terms, all modelled hierarchically.

The age association of BMI was modelled using a cubic spline to allow for non-linear age patterns, which might vary across countries. The coefficients of the splines were modelled hierarchically[1,10,81]. For adults, we allowed the coefficients to vary over time to reflect changing age associations[1,81]. For children and adolescents, model testing showed that a simpler model without age–time interaction had better performance[1,10]. For adults, two knots were placed at 45 and 60 years, and for children and adolescents, at 10 and 15 years, on the basis of exploratory analyses[1,10,56].

The model accounted for the possibility that BMI in subnational and community samples might systematically differ from, and have larger variation than, nationally representative surveys through the inclusion of fixed-effect and random-effect terms. The fixed effects adjusted for systematic differences between subnational or community studies and national studies and allowed these differences to vary over time. The random effects allowed national data to have a larger influence on the estimates than subnational or community data with similar sample sizes. The model also accounted for urban–rural differences in the prevalence of a BMI category, through data-driven fixed effects for urban-only and rural-only studies. These urban and rural effects were weighted by the difference between study-level and country-level urbanization in the year when the study was conducted and were also permitted to vary across time.

We fitted the statistical model using Markov chain Monte Carlo (MCMC). For model fitting, data on participants 5–19 years of age were included in the analysis of trends in children and adolescents, and on participants 18 years of age and older in the analysis of trends in adults. Data on participants 18 and 19 years of age were included in both sets of models because these groups form a transitional age from adolescence to adulthood, hence these data are informative for estimates in both groups. All analyses were done separately by sex because age, geographical and temporal patterns of BMI differ between sexes[1,56,82]. Computational details, including on initialization of MCMC chains and model convergence, are provided in the section on model implementation.

**Model specification.** Each study contributed up to 15 data points for each BMI category and sex, with the exact number depending on the age groups represented in the study. In the model specification, an observation $y_{h,i}$, that is, the number of people in the prevalence category from age group $h$ of study $i$, carried out in country $j$ at time $t$, was specified to have a binomial distribution conditional on the sample size $n_{h,i}$ and prevalence $p_{h,i}$:

$$y_{h,i}|n_{h,i}, p_{h,i} \sim \text{Bin}(n_{h,i}, p_{h,i}).$$

We modelled the prevalence $p_{h,i}$ from age group $h$ of study $i$ via a latent variable $\alpha_{h,i} = \Phi^{-1}(p_{h,i})$, representing probit-transformed prevalence, through the following Gaussian distribution:

$$\alpha_{h,i} \sim N(a_{j[i]} + b_{j[i]}t[i] + u_{j[i],t[i]} + \gamma_i(z_h) + \boldsymbol{X_i}\boldsymbol{\beta} + e_i, \tau^2),$$

where $j$, the country in which a study was carried out, and $t$, the study year, are uniquely determined by the study index $i$; we denote this

determination of $j$ and $t$ on $i$ by $j[i]$ and $t[i]$, respectively. The country-specific intercept and linear time slope from country $j$ are denoted $a_j$ and $b_j$, respectively, with $j \in \{1, \ldots J\}$, where $J = 200$ is the total number of countries and territories in our analysis. We describe the hierarchical model used for the $a$'s and the $b$'s in the section 'Linear components of country time trends'. Letting $T = 45$ be the total number of years from 1980 to 2024, the $T$-length vector $u_j$ captures smooth non-linear change over time in country $j$, as described in the section 'Nonlinear change'. The contribution of the age term for age group $h$ (with mid-age $z_h$) in study $i$ is denoted by $\gamma_i(z_h)$; these are described in detail in the section 'Age model'. The matrix $\boldsymbol{X}$ contains terms describing whether studies were representative at the national, subnational or community level, and whether they were urban only, rural only or covered both areas, and $\boldsymbol{\beta}$ contains the associated fixed effects. In addition, a random effect $e_i$ was estimated for each study. These study-specific terms are described in the section 'Study-level terms and study-specific random effects'. The variance term $\tau^2$ captures variability not accounted for by the study-specific random effects, described in the section 'Residual age-by-study variability'. Priors assigned to model hyperparameters are summarized in Supplementary Table 3. Details on model fitting and convergence are given in the section 'Model implementation'. Finally, details on how country-level inference was performed are given in the section 'Inference and post-processing'.

**Linear components of country time trends.** The model had a hierarchical structure, in which studies were nested in countries, which were nested in regions (indexed by $l$), which were nested in super-regions (indexed by $m$), which were all nested in the globe (see Supplementary Table 1 for a list of countries in each region and regions in each super-region). This structure allowed the model to share information across units to a greater degree when data were non-existent or weakly informative (for example, had a small sample size or were not nationally representative) and, to a lesser extent, in data-rich countries and regions[83].

The $a$ and $b$ terms are country-specific linear intercepts and time slopes with terms at each level of the hierarchy, denoted by the superscripts $c, r, s$ and $g$, respectively:

$$a_j = a_j^c + a_{l[j]}^r + a_{m[j]}^s + a^g,$$

$$b_j = b_j^c + b_{l[j]}^r + b_{m[j]}^s + b^g,$$

$$a^x \sim N(0, \kappa_a^x),$$

$$b^x \sim N(0, \kappa_b^x),$$

where $x \in \{c, r, s\}$. The $\kappa$ terms were each assigned a flat prior on the standard deviation scale[84]. We also assigned flat priors to $a^g$ and $b^g$.

**Nonlinear change.** The prevalence of a BMI category may change nonlinearly over time[1,82]. We captured smooth nonlinear change in time in country $j$ using the vector $u_j$. Just as $a_j$ and $b_j$ are each defined as the sum of country, region, super-region and global components, we defined

$$u_j = u_j^c + u_{l[j]}^r + u_{m[j]}^s + u^g.$$

To allow the model to differentiate between the degrees of nonlinearity that exist at the country, region, super-region and global levels, we assigned the four components of each $u$ a discrete second-order Gaussian autoregressive prior[85,86]. In particular, the vectors $u_j^c, j \in \{1, \ldots, J\}$, $u_l^r, l \in \{1, \ldots, L\}$, $u_m^s, m \in \{1, \ldots, M\}$, and $u^g$, all of length $T$, are each given a Gaussian prior with mean zero and precision $\lambda_c P$, $\lambda_r P$, $\lambda_s P$ and $\lambda_g P$, respectively, where the scaled precision matrix $P$ in the Gaussian autoregressive prior penalizes first and second differences as follows:

$$P = \begin{bmatrix} 1 & 0 & 0 & \cdots & 0 \\ -2 & 1 & 0 & \cdots & 0 \\ 1 & -2 & 1 & \cdots & 0 \\ 0 & 1 & -2 & \cdots & 0 \\ 0 & 0 & 1 & \cdots & 0 \\ \vdots & \vdots & \vdots & \ddots & \vdots \\ 0 & 0 & 0 & \cdots & 1 \end{bmatrix} \begin{bmatrix} 1 & -2 & 1 & 0 & 0 & \cdots & 0 \\ 0 & 1 & -2 & 1 & 0 & \cdots & 0 \\ 0 & 0 & 1 & -2 & 1 & \cdots & 0 \\ \vdots & \vdots & \vdots & \vdots & \vdots & \ddots & \vdots \\ 0 & 0 & 0 & 0 & 0 & \cdots & 1 \end{bmatrix}$$

$$= \begin{bmatrix} 1 & -2 & 1 & 0 & 0 & \cdots & 0 \\ -2 & 5 & -4 & 1 & 0 & \cdots & 0 \\ 1 & -4 & 6 & -4 & 1 & \cdots & 0 \\ 0 & 1 & -4 & 6 & -4 & \cdots & 0 \\ 0 & 0 & 1 & -4 & 6 & \cdots & 0 \\ \vdots & \vdots & \vdots & \vdots & \vdots & \ddots & \vdots \\ 0 & 0 & 0 & 0 & 0 & \cdots & 1 \end{bmatrix}.$$

$P$ is multiplied by the estimated precision parameters $\lambda_c$, $\lambda_r$, $\lambda_s$ and $\lambda_g$, thus upweighting or downweighting the strength of its penalties and ultimately determining the degree of smoothing at each level. For each of the four precision parameters, we used a truncated flat prior on the standard deviation scale $(1/\sqrt{\lambda})$[84]. We truncated these priors such that $\log \lambda \leq 20$ for each of the four $\lambda$'s. This upper bound is enforced as a computational convenience, so that models with $\log \lambda > 20$ are treated as equivalent to a model with $\log \lambda = 20$, as they essentially have no extra-linear variability in time. In practice, this upper bound had little effect on the parameter estimates. Furthermore, we ordered the $\lambda$'s a priori as follows: $\lambda_c < \lambda_r < \lambda_s < \lambda_g$. This prior constraint conveys the expectation that the global trend in the prevalence of a BMI category has less extra-linear variability than the trend of any given super-region, which has less than those of constituent regions, which in turn has less variability than the trends of constituent countries.

The matrix $P$ has rank $T - 2$, corresponding to a flat, improper prior on the mean and the slope of the $u_j^c$'s, the $u_l^r$'s, the $u_m^s$'s and $u^g$, and is not invertible[87]. Thus, we had a proper prior in a reduced-dimension space[85], with the prior expressed as follows:

$$P(u_j^c | \lambda_c) \propto \lambda_c^{\frac{T-2}{2}} \exp\left\{ -\frac{\lambda_c}{2} u_j^{c\prime} P u_j^c \right\}.$$

If $u_j^c$ had a non-zero mean, this would introduce nonidentifiability with respect to $a_j^c$. By the same token, $b_j^c$ would not be identifiable if $u_j^c$ had a non-zero time slope, and similarly for the other means and slopes. Thus, to achieve identifiability of the $a$'s, $b$'s and $u$'s, we constrained the mean and slope of each of $u^g$, $u^s$, $u^r$ and $u^c$ to be zero. Enforcing orthogonality between the linear and nonlinear portions of the time trends meant that each can be interpreted independently.

For the cases in which we have observations for at least two different time points, this improper prior will not lead to an improper posterior because the data will provide information about the mean and slope. To enforce the desired orthogonality between the linear and nonlinear portions of the model, we used the Rue and Held correction[85]. For the countries without data (for adults, 4 for women and 8 for men; for children and adolescents, 11 for girls and 19 for boys), we took the Moore–Penrose pseudoinverse of $P$[88], setting to infinity those eigenvalues that correspond to the non-identifiability. This effectively constrained the non-identified portions of the model to zero, as the corresponding variances are set to zero[86]; in this case the Rue and Held correction[85] is not needed. An intermediate case occurs when data are observed for only one time point in a country. In this case, the full conditional precision has rank $T - 1$ because the mean but not the linear trend of $u_j^c$ is identified by the data. We therefore constrained the linear trend of $u_j^c$ to zero in this case, by taking the generalized inverse of the full conditional precision. We then constrained the mean of $u_j^c$ to zero using the one-dimensional version of

the Rue and Held correction[85]. Computational details have been given in previous papers[71].

**Age model.** We sought a smooth function that could characterize gradual changes in the prevalence of BMI categories over age, as seen in the data. To achieve this, we modelled age using cubic splines, with the number and position of the knots of the spines selected based on epidemiological and physiological knowledge about changes in body shape[61,89] and statistical considerations, as previously described[1,56,82]. Statistically, we used age-stratified residuals to confirm the number and position of knots.

For age group $h$ with mid-age $z_h$, in study $i$, the age term is given by

$$\gamma_i(z_h) = \gamma_{1,i}z_h + \gamma_{2,i}z_h^2 + \gamma_{3,i}z_h^3 + \gamma_{4,i}(z_h - k_1)_+^3 + \gamma_{5,i}(z_h - k_2)_+^3,$$

where for children and adolescents, the two knots were placed at ages $(k_1, k_2) = (10, 15)$ and for adults at $(k_1, k_2) = (45, 60)$ years. To reduce dependence among model parameters, we centred the age variable.

We used different age models for children and adolescents and for adults, as explained below, following previous analyses[10,56,81], and visual inspection of results as well as formal model testing carried out using the Watanabe–Akaike information criterion[90,91].

For adults, each of the spline coefficients was allowed to vary across countries and was modelled hierarchically, and was further allowed to vary across time, to reflect different trends in prevalence across age groups. We modelled spline coefficients as follows, consistent with previous analyses[1,81], with the $k$-th age term coefficients for study $i$ given as follows:

$$\gamma_{k,i} = \psi_k^g + \psi_{k,j[i]}^c + \psi_{k,l[i]}^r + \psi_{k,m[i]}^s + (\phi_k^g + \phi_{k,j[i]}^c + \phi_{k,l[i]}^r + \phi_{k,m[i]}^s)t[i],$$

$$\psi_{k,j[i]}^c \sim N(0, \sigma_{\psi,k,c}^2),$$

$$\psi_{k,l[i]}^r \sim N(0, \sigma_{\psi,k,r}^2),$$

$$\psi_{k,m[i]}^s \sim N(0, \sigma_{\psi,k,s}^2),$$

$$\phi_{k,j[i]}^c \sim N(0, \sigma_{\phi,k,c}^2),$$

$$\phi_{k,l[i]}^r \sim N(0, \sigma_{\phi,k,r}^2),$$

$$\phi_{k,m[i]}^s \sim N(0, \sigma_{\phi,k,s}^2).$$

Here $\psi^g$, $\psi^c$, $\psi^r$ and $\psi^s$ are global, country, region and super-region intercepts, and $\phi^g$, $\phi^c$, $\phi^r$ and $\phi^s$ are global, country, region and super-region time slope parameters. A flat improper prior was placed on each of the $\sigma_\psi$'s and $\sigma_\phi$'s.

For children and adolescents, use of the model comparison criteria Watanabe–Akaike information criterion showed that the age–time interaction terms, $\phi$, did not improve model fit. Therefore, each of the spline coefficients was still allowed to vary across countries and was modelled hierarchically but was held constant over time, consistent with previous analyses[1,10]. The $k$-th age term coefficients for study $i$ were given as follows:

$$\gamma_{k,i} = \psi_k^g + \psi_{k,j[i]}^c + \psi_{k,l[i]}^r + \psi_{k,m[i]}^s,$$

$$\psi_{k,j[i]}^c \sim N(0, \sigma_{\psi,k,c}^2),$$

$$\psi_{k,l[i]}^r \sim N(0, \sigma_{\psi,k,r}^2),$$

$$\psi_{k,m[i]}^s \sim N(0, \sigma_{\psi,k,s}^2).$$

with flat improper prior placed on each of the $\sigma_\psi$'s.

**Study-level term and study-specific random effects.** The prevalence of a BMI category as measured in individual studies may vary from the true unobserved country–year prevalence owing to study implementation factors such as those associated with sampling, participation and response, and measurement. We included time-varying offsets (referred to above as fixed effects) to help account for potential systematic differences associated with data sources that are representative of subnational or community populations, and data sources that are representative of urban-only or rural-only populations, through the term $X_i\beta$:

$$\begin{aligned} X_i\beta = &\beta_1 I\{X_i^{\text{cvrg}} = \text{subnational}\} + \beta_2 I\{X_i^{\text{cvrg}} = \text{subnational}\}t[i] \\ &+ \beta_3 I\{X_i^{\text{cvrg}} = \text{community}\} + \beta_4 I\{X_i^{\text{cvrg}} = \text{community}\}t[i] \\ &+ \beta_5 X_{j[i],t[i]}^{\text{c.urb}}I\{X_i^{\text{s.urb}} = \text{rural}\} + \beta_6 X_{j[i],t[i]}^{\text{c.urb}}I\{X_i^{\text{s.urb}} = \text{rural}\}t[i] \\ &+ \beta_7 (1 - X_{j[i],t[i]}^{\text{c.urb}})I\{X_i^{\text{s.urb}} = \text{urban}\} + \beta_8 (1 - X_{j[i],t[i]}^{\text{c.urb}})I\{X_i^{\text{s.urb}} = \text{urban}\}t[i] \end{aligned}$$

where $X_i^{\text{cvrg}}$ is the indicator for whether the coverage of study $i$, in country $j$ and year $t$, is subnational or community, $X_i^{\text{s.urb}}$ is the indicator for whether the study $i$ covered rural-only or urban-only populations, and $X_{j[i],t[i]}^{\text{c.urb}}$ is the percentage of the national population of country $j$ in year $t$ living in urban areas, as obtained from the 2018 revision to the United Nation's World Urbanization Prospects[92]. We note that $\beta_5$ through $\beta_8$ are all multiplied by zero for studies that are urban only in countries where all residents lived in urban areas (for example, Singapore) and for studies that are rural only in countries where all residents lived in rural areas (for example, Tokelau), that is, in such cases, the model does not consider studies classified as urban (respectively rural) to have potential systematic differences from the true underlying prevalence in the country.

Even after accounting for sampling variability, national studies may still not reflect the true prevalence of a BMI category in a country with perfect accuracy, and subnational and community studies have even larger variability. We include the study-specific random effect $e_i$ to allow all age groups from the same study to have an unusually high or an unusually low prevalence, after conditioning on the other terms in the model. Each $e_i$ is assigned a Gaussian prior with variance dependent on whether study $i$ is representative at the national, subnational or community level. Random effects from national studies were constrained to have smaller variance ($v_n$) than random effects of subnational studies ($v_s$), which were in turn constrained to have smaller variance than community studies ($v_c$).

**Residual age-by-study variability.** The age patterns across communities within a given country may differ from the overall age pattern of that country. This within-study variability cannot be captured by the $e_i$ terms, which are equal across age-specific observations in each study, so we included an additional variance component for each study, $\tau^2$.

**Model implementation.** The model was fitted through a bespoke MCMC sampler coded in R, which uses a combination of Metropolis–Hastings and Gibbs updates[93]. To generate starting values for the model runs, we ran an initial set of eight MCMC chains. We generated the starting values of each initial chain by first randomly generating log variance parameter values from diffuse Gaussian distributions centred on estimates from previous analyses, and then generating all other starting values conditional on these variance parameters. We ran each of the initial chains for 50,000 iterations after burn-in, thinned and combined across chains to obtain 5,000 posterior draws. To estimate a distribution from which to sample initial values for the final model runs, we fitted a multivariate Gaussian distribution to the

posterior distribution of all non-study-specific parameters obtained from the initial chains, scaling the variance–covariance matrix by a factor of 1.5; this equates to an increase in the variance of the multivariate Gaussian distribution of approximately 50% relative to the target posterior distribution. This is a larger overdispersion than that of 10%, which is considered sufficient for the Rhat convergence diagnostic[94], and allows a larger spread of initial values to be included. To obtain initial values for study-specific parameters, we first sampled a study-specific random effect $e_i$, for each study $i$, from a Gaussian distribution with mean zero and variance given by the sampled initial values of $v_n$, $v_s$ or $v_c$, dependent on whether study $i$ was representative at the national, subnational or community level. We then sampled initial values of the latent variable $\alpha_{h,i}$ for each age group $h$ and study $i$ from its Gaussian distribution, conditional on all other sampled parameter values, including the study-specific random effect $e_i$.

We had a target of eight converged MCMC chains for generating our estimates, which is twice the recommended minimum number to assess convergence using the Rhat diagnostic[91,95]. The exact numbers of chains used for the model runs are not critical so long as at least four chains are run to enable us to estimate between chain variation, which is needed for the Rhat convergence diagnostic to be meaningful[95], and so long as there are sufficient computational resources to run chains to convergence and subsequently to collect samples. We ran ten chains for each BMI category sex combination, with chains ordered by their seeds. The additional two chains were run to allow for a small number of the first eight chains to be discarded if mixing was slow. In practice, no chain was replaced. We did not run more chains because the computational and time cost outweighed the gains, if any, in results. We identified, through visual inspection of hyperparameter trace plots, a burn-in period of 20,000 iterations for adult prevalence categories, and 30,000 for child and adolescent categories. We took 50,000 post-burn-in iterations from each of the eight target chains, and combined and thinned to obtain a final sample of 5,000 posterior draws for each outcome.

Convergence was confirmed through visual inspection as well as through calculated split-Rhat diagnostic for country–year–age outcomes as implemented in the R package 'rstan' (v2.26.15)[95,96]. The 97.5th quantile of split-Rhat ranged across BMI categories and sexes from 1.003 to 1.013 for adults, and from 1.004 to 1.014 for children and adolescents. Over 99% of country–year–age outcomes across all categories and sexes for adults and for children and adolescents had split-Rhat < 1.05.

**Inference and post-processing.** All inference was done for country–year–age combinations, through combining the $a$, $b$, $u$ and $\gamma$ terms, and setting $\beta = 0$ and $e_i = 0$. We set $\beta = 0$ as fixed effects associated with study design are not relevant for country-level inference. We set $e_i = 0$ as random effects arising from imperfections and variations in study design and implementation, and from within-country variability of the prevalence of a BMI category, are also not relevant for country-level inference.

Posterior estimates were made in 1-year age groups for 5–19 years of age, because BMI changes rapidly in relation to age in these ages, and in 5-year age groups for 20 years of age and older. As in previous work[1,82], we rescaled the estimated prevalence of different BMI ranges so that their sum was 1.0 in each sex, age, country and year. The average scaling factors across samples ranged from 1.00 to 1.02 for children and adolescents and 0.99 to 1.03 for adults, that is, the sum of the separately estimated prevalence categories was close to 1. We calculated the prevalence of obesity at the draw level as the sum of the prevalence of BMI of 30 kg m$^{-2}$ to less than 35 kg m$^{-2}$, 35 kg m$^{-2}$ to less than 40 kg m$^{-2}$, and 40 kg m$^{-2}$ or higher.

For presentation, we summarized results for 5–19 years of age for children and adolescents, and for 20 years of age and older for adults, as age-standardized results. Age standardization puts the population

for each country–year on the same (standard) age distribution, hence enables comparisons to be made over time and across countries. Age standardization was performed by taking the weighted means of age–sex-specific estimates, separately for children and adolescents and for adults, using age weights from the WHO standard population[97]. We calculated the velocity of obesity for a given year at the draw level as the absolute difference in age-standardized obesity prevalence between consecutive years. To simplify reporting notation, we refer to velocity for each set of consecutive years by the terminal year: for example, the 2023–2024 velocity is referred to as velocity in 2024.

The uncertainties of our estimates, represented by their posterior distributions, capture the following sources of uncertainty in true obesity prevalence: uncertainty due to sampling in each data source; uncertainty associated with the variability of national data beyond what is accounted for by sampling; uncertainty associated with subnational and community data, which are more variable than national data; and uncertainty due to making estimates by country, year and age when data were missing, scarce or weakly informative. The reported credible intervals represent the 2.5–97.5th percentiles of the posterior distributions, which contain the true estimates with 95% probability. We obtained the PP that an estimated change in obesity represented a true increase as the proportion of draws from the posterior distribution that indicated an increase, that is, a positive change. We obtained the PP that an estimated velocity of obesity was positive (that is, prevalence is increasing) as the proportion of posterior draws for which the velocity was positive.

**Validation of statistical model.** To evaluate how well our statistical model fitted the data, we calculated the difference between the posterior estimates of obesity prevalence from the model and data from national studies. Median errors were very close to zero (0.11 percentage points for children and adolescents and 0.06 percentage points for adults) and median absolute errors were 1.09 percentage points for children and adolescents and 1.11 percentage points for adults, indicating that the estimates were unbiased and had small deviations relative to national studies.

Although we had data for 189 out of 200 (95%) countries for children and adolescents, and 196 of 200 (98%) for adults, we also conducted the more challenging test of how well our statistical model predicts missing data, known as external predictive validity. We evaluated external predictive validity in two different tests. In test 1, we held out all data from 10% of countries with data (that is, created the appearance of countries with no data where we actually had data), a higher percentage than the actual missingness in the dataset that we used. The countries whose data were withheld were randomly selected from the following three groups: data-rich (12 or more data sources for girls, 10 or more for boys, 13 or more for women and 10 or more for men), data-poor (1–3 data sources for girls, 1–2 for boys, 1–4 for women and 1–2 for men) and average data (4–11 data sources for girls, 3–9 for boys, 5–12 for women and 3–9 for men) availability. All data-rich countries had at least one data source after 2010. We fitted the model to the data from the remaining 90% of countries and made estimates of the held-out observations. In test 2, we assessed other patterns of missing data by holding out 10% of our data sources, again from a mix of data-rich, data-poor and average-data countries, as defined above. For a given country, we either held out a random one-third of the data of that country or all of the 2010–2024 data of that country to determine, respectively, how well we filled in the gaps for countries with intermittent data and how well we estimated in countries without recent data. Given that test 1 held out 10% of countries with data and we had data for 95% of countries for children and adolescents and 98% for adults, test 2 is a better reflection of external predictive validity of our analysis.

We fitted the model to the remaining 90% of the dataset and made estimates of the held-out observations. We repeated each test five times, holding out a different subset of data in each repetition. In

both tests, we calculated the differences between the held-out data and the estimates. We also calculated the 95% credible intervals of the estimates; in a model with good external predictive validity, 95% of held-out values would be included in the 95% credible intervals, a metric referred to as coverage.

Our statistical model also performed well in the external validation tests, that is, in estimating obesity prevalence when data were missing. The estimates of obesity prevalence had median errors that were close to zero or small globally (for test 1 and test 2, respectively, 0.61 and 0.17 percentage points for girls, 1.02 and 0.63 percentage points for boys, 1.82 and 0.21 percentage points for women, and 1.46 and 0.15 percentage points for men), and ±4 percentage points or less in every subset of withheld data except for Pacific Island nations in test 1 for adults where median error was −4.51 and 4.94 percentage points for women and men, respectively (Supplementary Table 4). The 95% credible intervals of estimated prevalence of obesity covered 92–97% of true data globally for children and adolescents and 87–91% for adults; coverage was above 85% in most subsets of withheld data. Median absolute errors globally ranged from 1.72 to 3.53 percentage points for children and adolescents and from 2.21 to 4.57 percentage points for adults. Median absolute errors were smaller in test 2, where a subset of data sources from some countries are withheld, than in test 1, where all data from some countries are withheld. For comparison, median absolute differences for prevalence of obesity between pairs of nationally representative surveys done in the same country and in the same year was 2.06 percentage points for children and adolescents and 1.76 percentage points for adults, indicating that our estimates perform almost as well as conducting two distinct surveys in the same country and year.

## Clustering obesity prevalence time series

We used clustering analysis to identify national obesity prevalence time series that have similar shapes in a data-driven (unsupervised) approach. The input to the analysis was the posterior mean age-standardized obesity prevalence for each country from 1980 to 2024 estimated via the Bayesian meta-regression model as detailed above. Data preprocessing and clustering were performed separately by sex and for children and adolescents and for adults.

We first normalized the posterior mean obesity prevalence in each country by subtracting the mean and dividing by the standard deviation so that the time series for each country had zero mean and unit variance. This step allowed the subsequent clustering to be determined by the shape of trajectory as opposed to its level and magnitude of change, which are captured by prevalence and velocity metrics. We then, for each year, subtracted the annual mean normalized prevalence across all countries from the country-specific (normalized) prevalence to remove the overall temporal trend in normalized prevalence. We used principal component analysis to identify the features that explained most variance, hence removed low-variance features[98]. The first three principal components explained more than 97.9% of the variance and characterized the shape of the time series of each country.

We applied $k$-means clustering to the country scores of the first three principal components. The $k$-means algorithm partitions the data into $k$ mutually exclusive clusters that are relatively homogeneous while maximizing the heterogeneity among clusters, by minimizing the sum of Euclidean distances of all data points from the centre of the cluster they belong to. It is a widely used and computationally efficient clustering algorithm that produces non-overlapping clusters.

In $k$-means analysis, the number of clusters $k$ must be pre-specified. Various heuristics have been suggested for selecting a suitable number of clusters such as the elbow method and the silhouette method. These metrics compare measures of cluster cohesion and cluster separation for different choices of $k$. We ran the $k$-means algorithm across a range of $k$ values from 3 to 20, and selected $k = 6$ based on these heuristics as well as epidemiological interpretability of the results and consistency between sexes and between children and adolescents and adults.

The $k$-means algorithm uses random starting values and iteratively minimizes the sum of distances until reaching an optimum. To minimize the risk of converging to local optima, we initiated with 50 different random starting values and used the best cluster allocation, that is, the one with the minimum sum of distances across the 50 runs[99]. We labelled each cluster based on the typology of obesity prevalence trends of the countries in the cluster.

We evaluated the stability of the resultant clusters by calculating the mean Jaccard index[100] between the clustering results over all countries and that of 1,000 subsamples of countries drawn without replacement (Supplementary Table 5). The Jaccard index is a measure of similarity between two groups and ranges from 0 to 1, with 0 indicating no overlap and 1 indicating identical results. We calculated the Jaccard index across a range of subsample proportions from 50%, which is an extreme situation in terms of the share of countries removed, to 90%. We computed a cluster-wise Jaccard index for each cluster in a comparison with the most similar cluster obtained in each subsample[100]. The mean Jaccard index across 1,000 subsamples reflects how consistently the same cluster appears across repeated subsampling and reclustering. As a practical guideline, a mean Jaccard index larger than 0.75 is considered as showing good recovery and overall stability, whereas values smaller than 0.60 suggest that the cluster is not consistently recovered across repeated subsampling[100–102]. The results indicate highly stable clusters with average Jaccard indices across the six clusters for the four age group and sex combinations ranging 0.90–0.96 when 90% of countries were used, 0.84–0.92 when 80% were used and 0.77–0.83 in the difficult task of clustering with only 50% of countries. For children and adolescents, the Jaccard index remained above 0.75 for all clusters when up to 20% of countries were dropped. When 30% or more of countries were dropped, the Jaccard index was above 0.75 for all but one cluster each for girls (decelerating increase) and for boys (recent decline). For adults, the Jaccard index remained above 0.75 when up to 20% of countries were dropped for all but one cluster (recent decline for men) possibly because some of its constituent countries were absorbed into the related clusters such as plateau as countries were dropped. The average Jaccard index for the cluster of decelerating increase or plateau for women also became lower than 0.75 when 30% or more of countries were dropped, as it exchanged some constituent countries with other clusters. Nonetheless, even in the extreme situation of using only 50% of countries, Jaccard index was over 0.60 for all but one cluster for boys (recent decline) where it was just below the threshold (0.59).

All analyses were performed in statistical software R (v4.3.0).

## Comparison with prior studies

One study[14] presented an obesity transition framework with four stages and qualitatively applied the staging in 30 example countries. The qualitative features of the four obesity transition stages are consistent with our findings. However, the framework paper did not use data after 2016 and found few countries in the proposed fourth stage where obesity prevalence declines. The study only qualitatively considered the differences in trends between sexes and age groups in its application and did not analyse the range of prevalences at which trends plateaued. Prior global studies[1,82,103,104] have reported changes in obesity prevalence over multi-decadal periods but did not attempt to categorize or quantify dynamic trajectories, hence our results cannot be directly compared with theirs. Two of these studies graphically showed a continuing increase in obesity in most world regions from 1990 to 2021[103,104], different from our quantitative findings of plateauing of the rise in obesity in many high-income countries. The plateauing of obesity prevalence that we reported in many high-income countries for one or both sexes is consistent with reports of prevalence time series in specific countries[105–121] and reviews of published studies[122–125]. Only one study[126] has quantitatively classified obesity prevalence trends for different US states, reporting mixed typologies of subnational trends similar to what we found globally.

## Ethics and inclusion statement

This research followed the recommendations set out in the Global Code of Conduct for Research in Resource-Poor Settings.

## Reporting summary

Further information on research design is available in the Nature Portfolio Reporting Summary linked to this article.

## Data availability

Age-standardized and age-specific results of this study are available on NCD-RisC (www.ncdrisc.org) in machine-readable numerical format and as visualizations. Input data from publicly available sources and contact information for data providers can be downloaded from NCD-RisC (www.ncdrisc.org) and Zenodo[127] (https://doi.org/10.5281/zenodo.18368826).

## Code availability

The computer codes for the Bayesian hierarchical model and clustering analysis used in this work are available on NCD-RisC (www.ncdrisc.org) and Zenodo[127] (https://doi.org/10.5281/zenodo.18368826).

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

**Acknowledgements** This study was funded by the UK Medical Research Council (grant number MR/V034057/1) and UK Research and Innovation (Innovate UK grant number 10103595, for participation in the OBCT consortium funded by the European Union grant agreement 101080250). The authors alone are responsible for the views expressed in this Article and they do not necessarily represent the views, decisions or policies of the institutions with which they are affiliated.

**Author contributions** B.Z., A.G., L.J., Y.D.D.B., A.B.P., N.H.P., F.D., O.N.O., G.A.S., R.M.C.L., A.W.R., A.Z. and N.R.E. collated data from different studies and checked data sources. B.Z., A.G., N.H.P. and V.N. managed the database. B.Z., N.H.P., A.M., J.E.B., O.N.O., V.K., A.R.M. and Y.F. developed and coded the statistical method with inputs from C.J.P. and M.E. B.Z., A.G., O.N.O. and N.H.P. analysed the data and prepared the results. Other authors collected and reanalysed the data and checked the pooled data. M.E., B.Z., A.G., N.H.P., O.N.O. and J.E.B. wrote the first draft of the report with input from other authors. The corresponding author (M.E.) had the final responsibility for the decision to submit for publication.

**Competing interests** G.D. reports research consulting fees from Resolve to Save Lives, outside of the submitted work. K.K. reports grants in support of investigator and investigator-initiated trials from AstraZeneca, Novartis, Novo Nordisk, Sanofi, Eli Lilly, Merck Sharp & Dohme and Boehringer Ingelheim; consultancy fees from AstraZeneca, Bayer, Novartis, Novo Nordisk, Sanofi-Aventis, Eli Lilly, Merck Sharp & Dohme, Boehringer Ingelheim, Oramed Pharmaceuticals, Pfizer, Roche and Applied Therapeutics; and payments for speaking from AstraZeneca, Bayer, Novartis, Novo Nordisk, Sanofi, Eli Lilly, Merck Sharp & Dohme, Boehringer Ingelheim, Oramed Pharmaceuticals, Pfizer, Roche and Applied Therapeutics, all outside of the submitted work. L.-L.L. reports research grants via her institution from Abbott Diabetes Care, AstraZeneca and Novartis; speaker honoraria from Abbott, AstraZeneca, Boehringer Ingelheim, Novo Nordisk, Roche Diabetes Care and Zuellig Pharma, all outside of the submitted work. C.J.P. reports holding stocks in Pfizer, outside of the submitted work. F.Z. reports consulting fees from Daiichi Sankyo, Servier and Menarini, all outside of the submitted work. M.E. reports payment for advisory board from Lean, outside the submitted work.

**Additional information**
**Correspondence and requests for materials** should be addressed to Majid Ezzati.

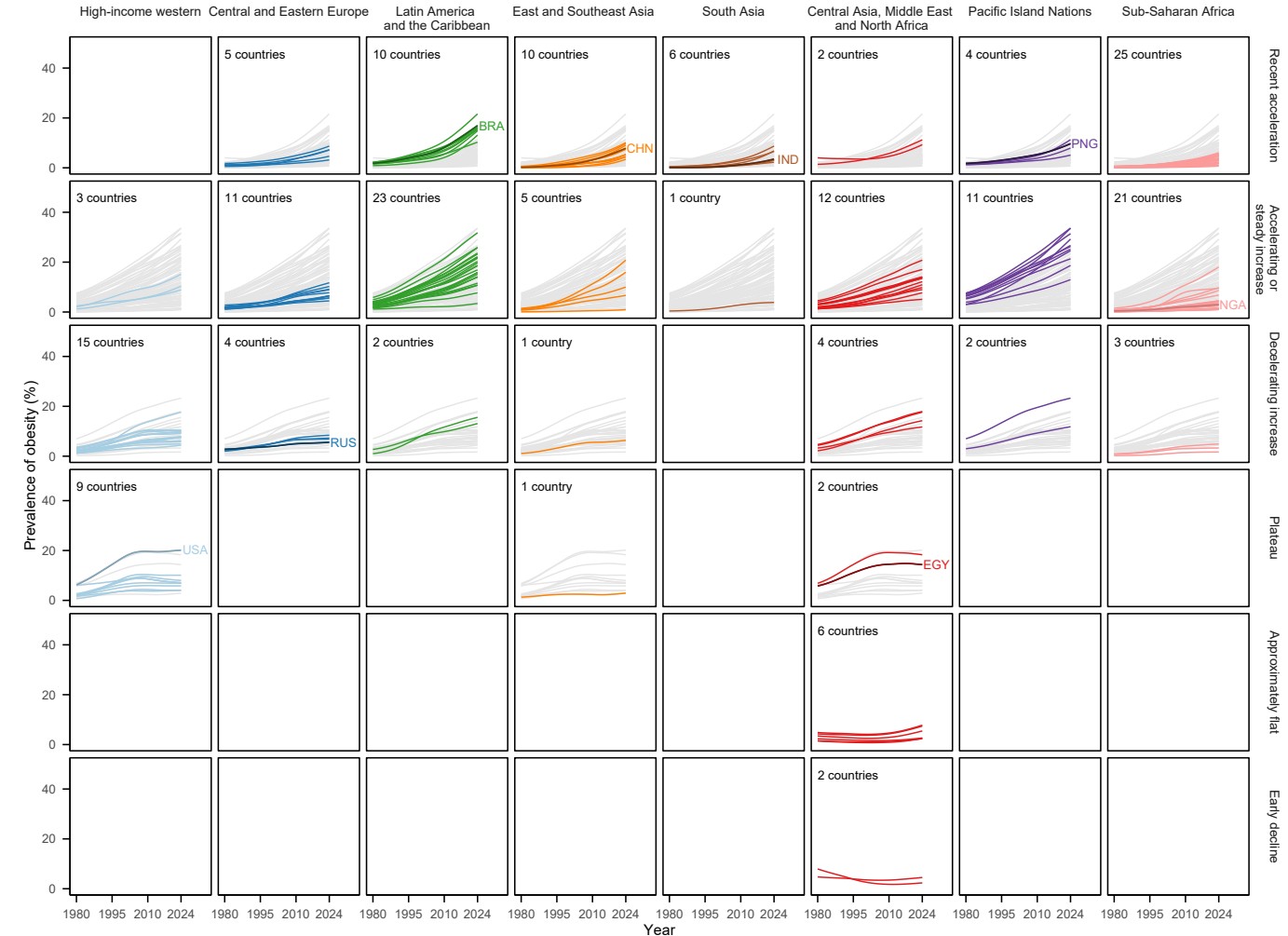

**Extended Data Fig. 1 | Phenotypes of national obesity trajectories by super-region in girls.** Each coloured line shows age-standardized obesity prevalence trends over time for one country belonging to a cluster and super-region combination. They are coloured by super-region. Grey lines show all countries belonging to a cluster across all super-regions. The number at the top of each panel shows the number of countries belonging to each cluster-super-region combination. The trend line for the most populous country in each super-region is identified by their ISO 3166-1 alpha-3 codes (Supplementary Note 1).

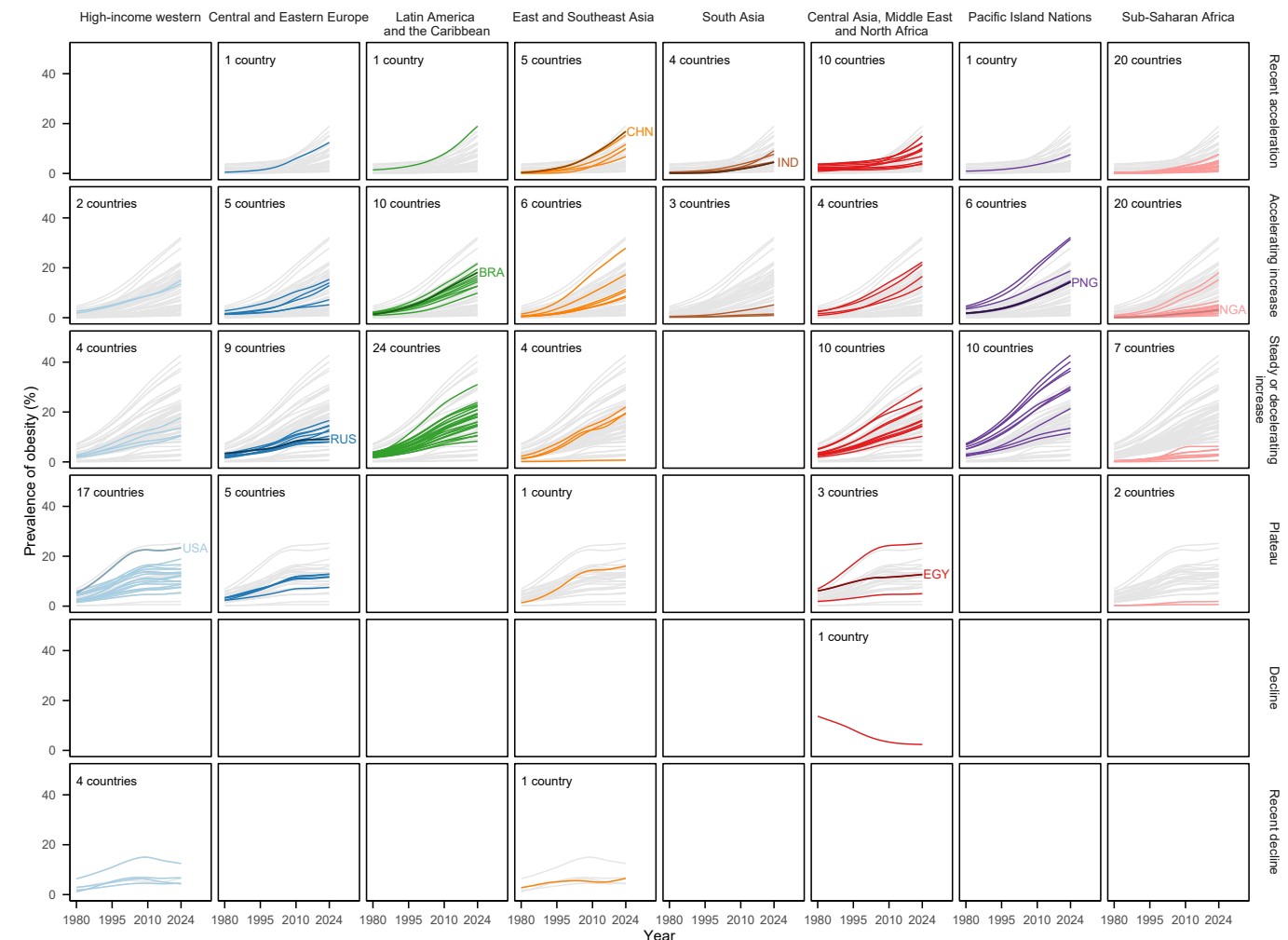

**Extended Data Fig. 2 | Phenotypes of national obesity trajectories by super-region in boys.** Each coloured line shows age-standardized obesity prevalence trends over time for one country belonging to a cluster and super-region combination. They are coloured by super-region. Grey lines show all countries belonging to a cluster across all super-regions. The number at the top of each panel shows the number of countries belonging to each cluster-super-region combination. The trend line for the most populous country in each super-region is identified by their ISO 3166-1 alpha-3 codes (Supplementary Note 1).

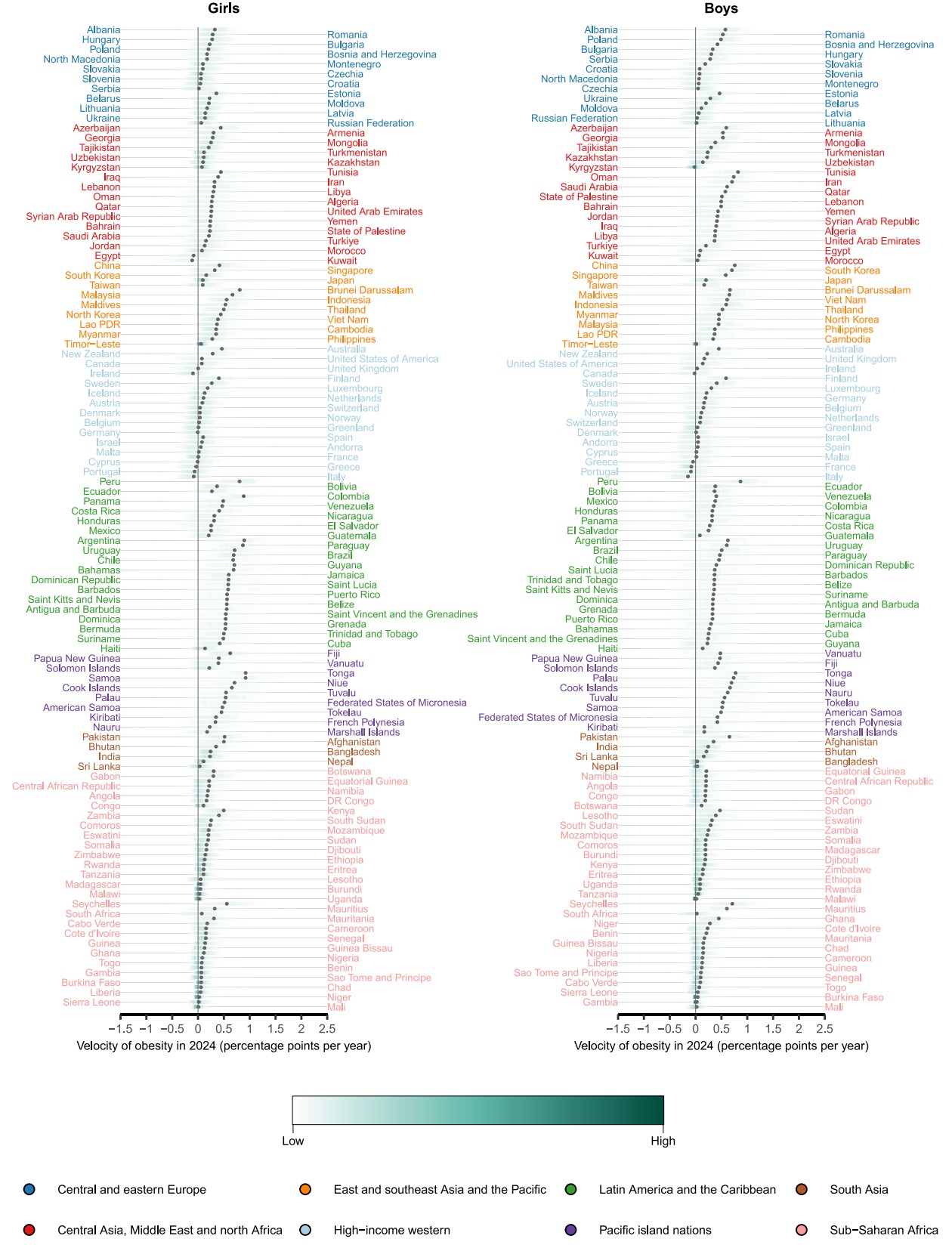

**Extended Data Fig. 3 | Posterior distribution of velocity of obesity in children and adolescents in 2024 by country.** For each country, the shaded area shows the posterior distribution of the velocity of obesity in 2024. Darker shading denotes higher posterior probability and lighter shading denotes lower posterior probability. The point for each country shows the posterior mean of velocity in 2024. Countries are ordered by region and the posterior mean velocity in 2024. Country names are coloured by super-region.

**Girls**

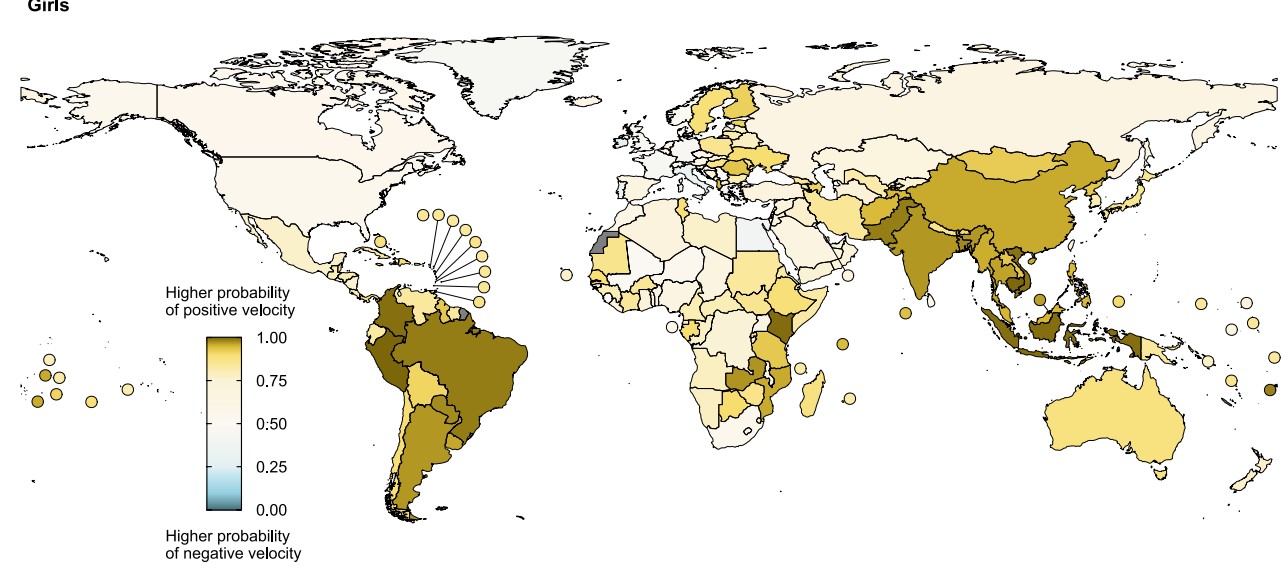

**Boys**

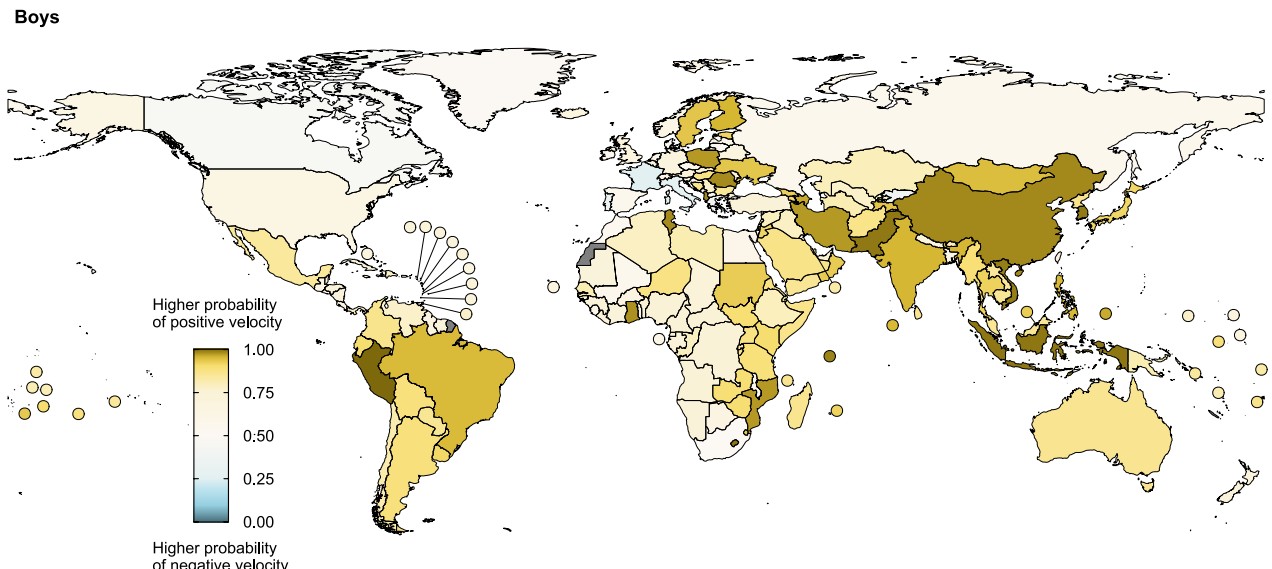

**Extended Data Fig. 4 | Posterior probability that velocity of obesity in children and adolescents was positive in 2024 by country.** If a positive velocity (i.e. an increase in obesity) is statistically indistinguishable from a negative velocity (i.e. a decrease in obesity), the PP is 0.5. PPs closer to 1 indicate more certainty of a positive velocity, those towards 0 indicate more certainty of a negative velocity, and those closer to 0.5 indicate less certainty of a positive or negative velocity.

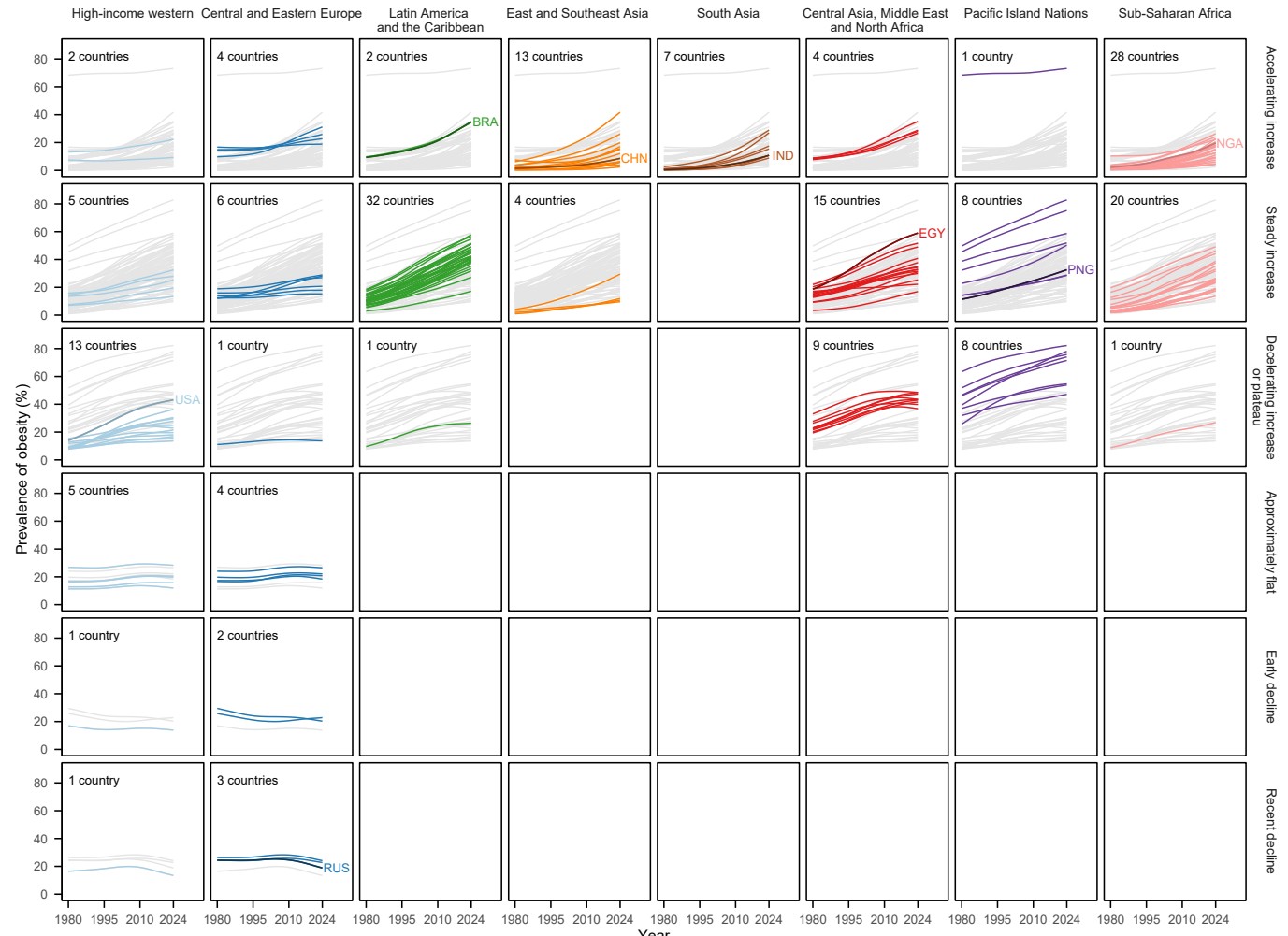

**Extended Data Fig. 5 | Phenotypes of national obesity trajectories by super-region in women.** Each coloured line shows age-standardized obesity prevalence trends over time for one country belonging to a cluster and super-region combination. They are coloured by super-region. Grey lines show all countries belonging to a cluster across all super-regions. The number at the top of each panel shows the number of countries belonging to each cluster-super-region combination. The trend line for the most populous country in each super-region is identified by their ISO 3166-1 alpha-3 codes (Supplementary Note 1).

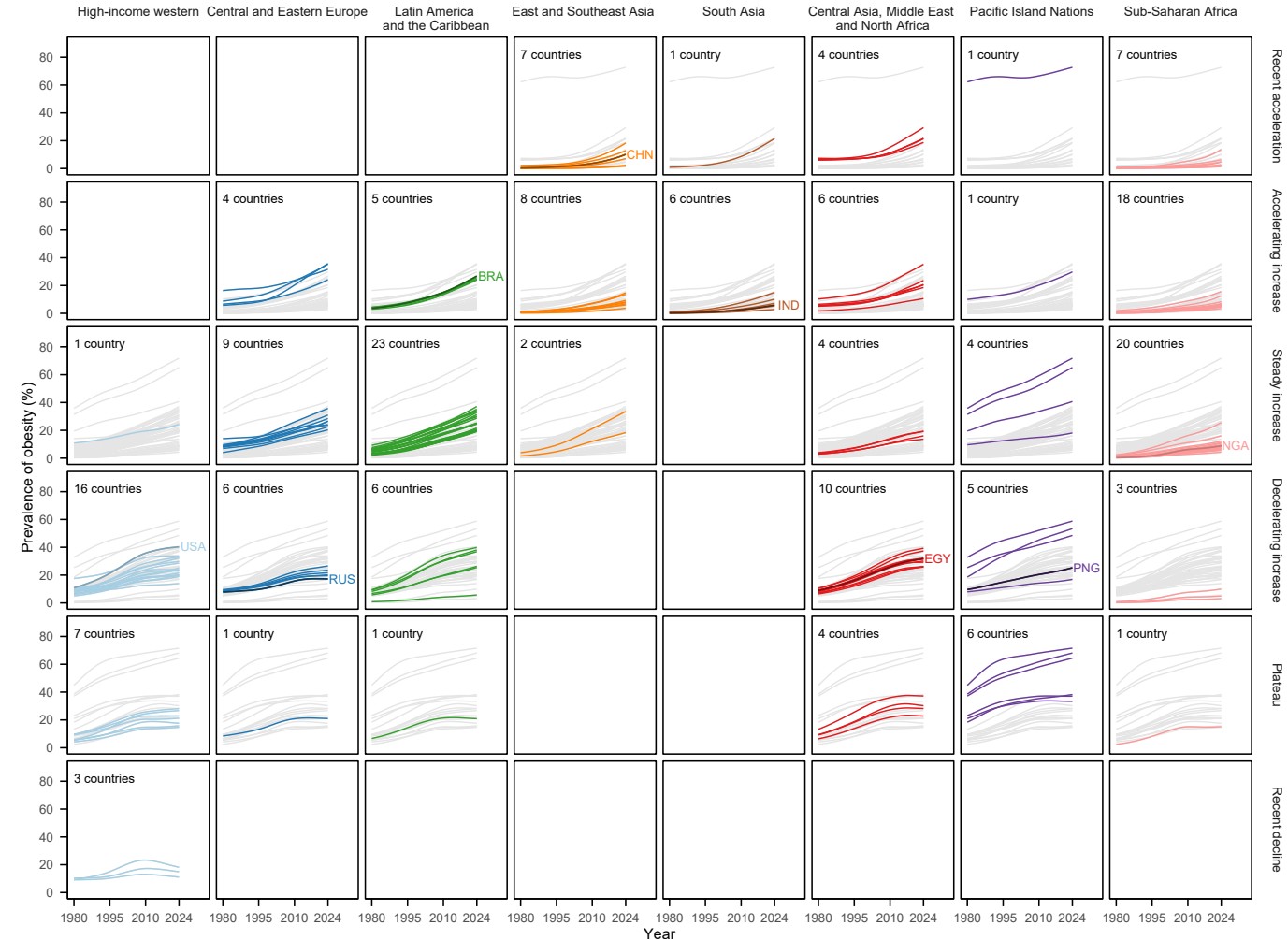

**Extended Data Fig. 6 | Phenotypes of national obesity trajectories by super-region in men.** Each coloured line shows age-standardized obesity prevalence trends over time for one country belonging to a cluster and super-region combination. They are coloured by super-region. Grey lines show all countries belonging to a cluster across all super-regions. The number at the top of each panel shows the number of countries belonging to each cluster-super-region combination. The trend line for the most populous country in each super-region is identified by their ISO 3166-1 alpha-3 codes (Supplementary Note 1).

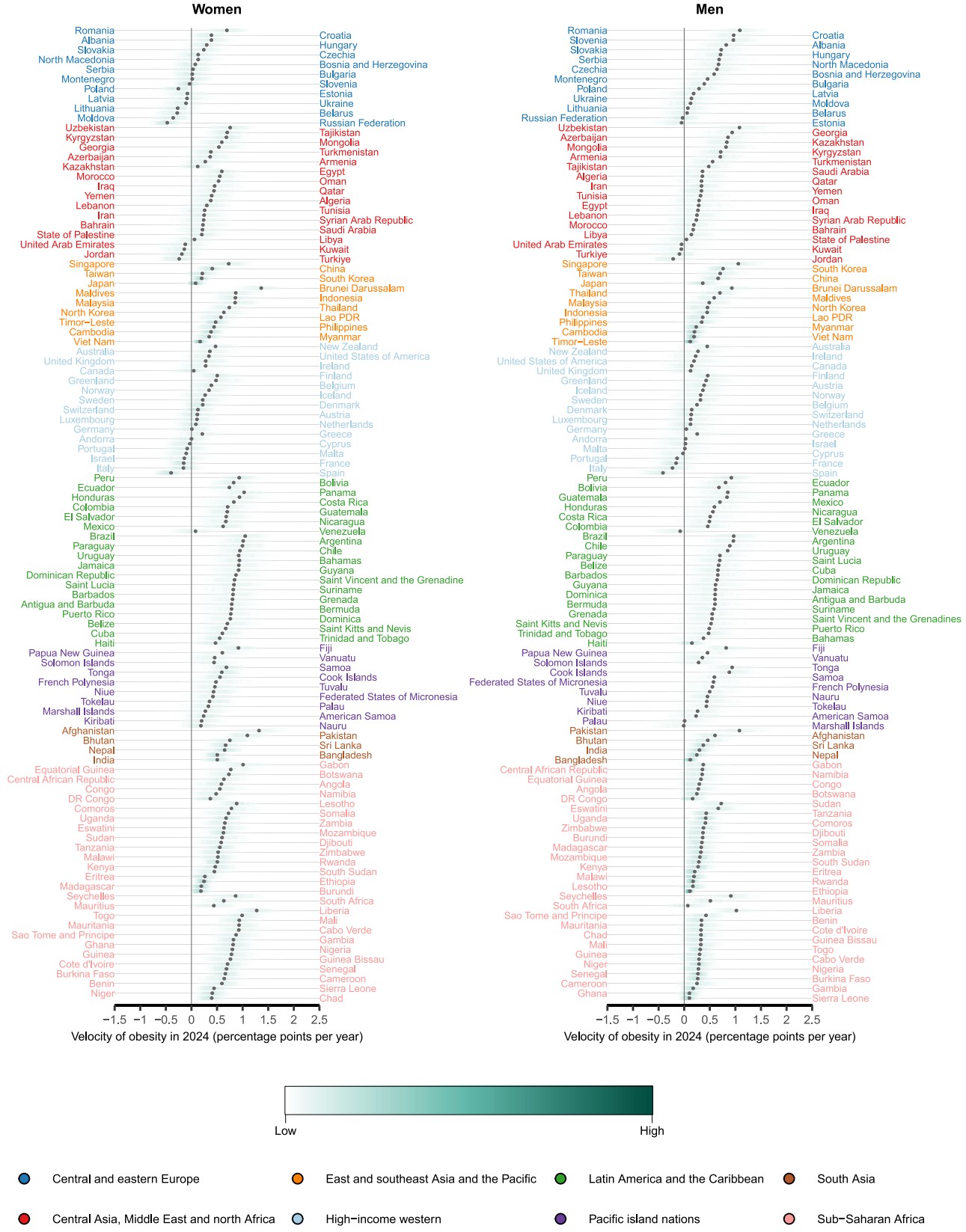

**Extended Data Fig. 7 | Posterior distribution of velocity of obesity in adults in 2024 by country.** For each country, the shaded area shows the posterior distribution of the velocity of obesity in 2024. Darker shading denotes higher posterior probability and lighter shading denotes lower posterior probability. The point for each country shows the posterior mean of velocity in 2024. Countries are ordered by region and the posterior mean velocity in 2024. Country names are coloured by super-region.

**Women**

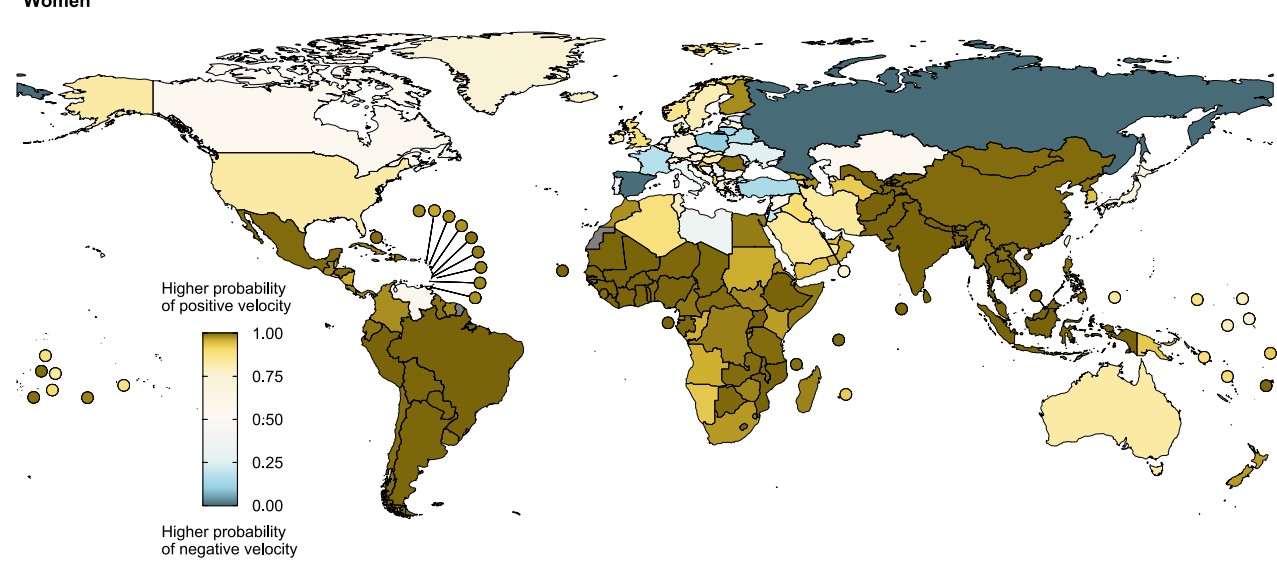

**Men**

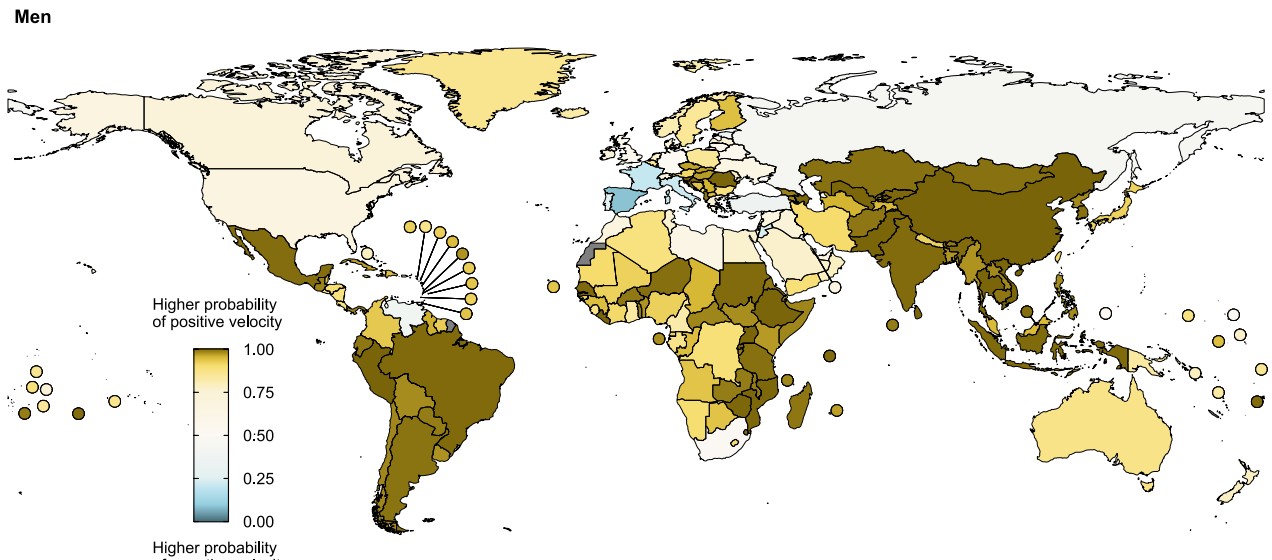

**Extended Data Fig. 8 | Posterior probability that velocity of obesity in adults was positive in 2024 by country.** If a positive velocity (i.e. an increase in obesity) is statistically indistinguishable from a negative velocity (i.e. a decrease in obesity), the PP is 0.5. PPs closer to 1 indicate more certainty of a positive velocity, those towards 0 indicate more certainty of a negative velocity, and those closer to 0.5 indicate less certainty of a positive or negative velocity.

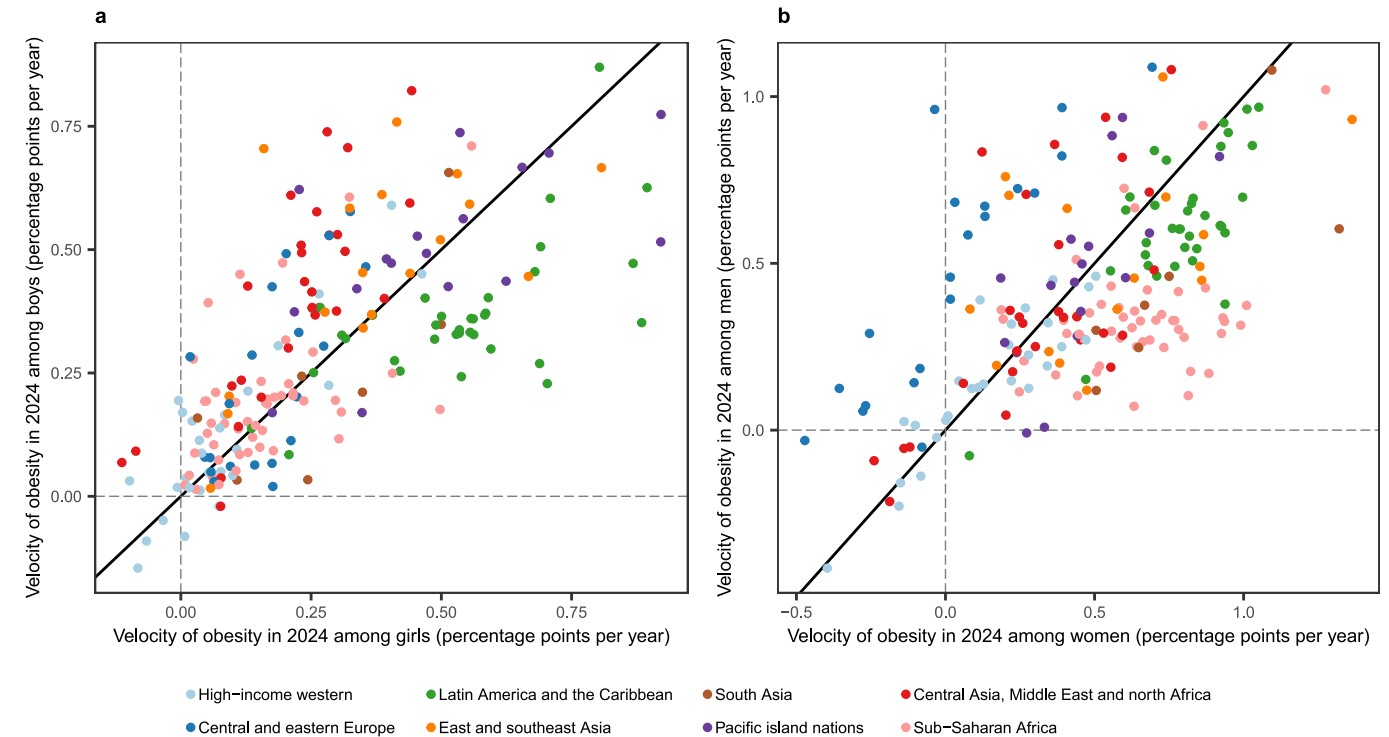

**Extended Data Fig. 9 | Female and male velocity of obesity in 2024.** Relationship between velocity of obesity for girls and boys (**a**) and for women and men (**b**). Each point shows one country, coloured by super-region. The solid grey line denotes equality of female and male velocity.

# Reporting Summary

## Statistics

For all statistical analyses, confirm that the following items are present in the figure legend, table legend, main text, or Methods section.

| n/a | Confirmed | |
|---|---|---|
| ☐ | ☒ | The exact sample size (*n*) for each experimental group/condition, given as a discrete number and unit of measurement |
| ☒ | ☐ | A statement on whether measurements were taken from distinct samples or whether the same sample was measured repeatedly |
| ☒ | ☐ | The statistical test(s) used AND whether they are one- or two-sided<br>*Only common tests should be described solely by name; describe more complex techniques in the Methods section.* |
| ☒ | ☐ | A description of all covariates tested |
| ☒ | ☐ | A description of any assumptions or corrections, such as tests of normality and adjustment for multiple comparisons |
| ☐ | ☒ | A full description of the statistical parameters including central tendency (e.g. means) or other basic estimates (e.g. regression coefficient) AND variation (e.g. standard deviation) or associated estimates of uncertainty (e.g. confidence intervals) |
| ☒ | ☐ | For null hypothesis testing, the test statistic (e.g. *F*, *t*, *r*) with confidence intervals, effect sizes, degrees of freedom and *P* value noted<br>*Give P values as exact values whenever suitable.* |
| ☐ | ☒ | For Bayesian analysis, information on the choice of priors and Markov chain Monte Carlo settings |
| ☒ | ☐ | For hierarchical and complex designs, identification of the appropriate level for tests and full reporting of outcomes |
| ☒ | ☐ | Estimates of effect sizes (e.g. Cohen's *d*, Pearson's *r*), indicating how they were calculated |

*Our web collection on statistics for biologists contains articles on many of the points above.*

## Software and code

Policy information about availability of computer code

| Data collection | No software was used for secondary data collection. |
|---|---|
| Data analysis | All analyses were conducting using the statistical software R (version 4.3.0). We used R package 'survey' (version 4.4.2) and 'rstan' (version 2.26.15). The code for estimation of mean risk factor trends is available at www.ncdrisc.org and Zenodo (https://doi.org/10.5281/zenodo.18368827). |

For manuscripts utilizing custom algorithms or software that are central to the research but not yet described in published literature, software must be made available to editors and reviewers. We strongly encourage code deposition in a community repository (e.g. GitHub). See the Nature Portfolio guidelines for submitting code & software for further information.

## Data

Policy information about availability of data

All manuscripts must include a data availability statement. This statement should provide the following information, where applicable:
- Accession codes, unique identifiers, or web links for publicly available datasets
- A description of any restrictions on data availability
- For clinical datasets or third party data, please ensure that the statement adheres to our policy

Age-standardised and age-specific results of this study are available from www.ncdrisc.org in machine-readable numerical format and as visualization. Input data

## Research involving human participants, their data, or biological material

Policy information about studies with [human participants or human data](). See also policy information about [sex, gender (identity/presentation), and sexual orientation]() and [race, ethnicity and racism]().

| | |
|---|---|
| Reporting on sex and gender | N/A |
| Reporting on race, ethnicity, or other socially relevant groupings | N/A |
| Population characteristics | N/A |
| Recruitment | N/A |
| Ethics oversight | N/A |

Note that full information on the approval of the study protocol must also be provided in the manuscript.

## Field-specific reporting

Please select the one below that is the best fit for your research. If you are not sure, read the appropriate sections before making your selection.

☐ Life sciences  ☒ Behavioural & social sciences  ☐ Ecological, evolutionary & environmental sciences

For a reference copy of the document with all sections, see [nature.com/documents/nr-reporting-summary-flat.pdf]()

## Behavioural & social sciences study design

All studies must disclose on these points even when the disclosure is negative.

| | |
|---|---|
| Study description | We pooled and re-analysed population-based data that had measured height and weight for children and adolescents and adults to estimate national trajectories in obesity from 1980 to 2024 for 200 countries and territories, using a Bayesian hierarchical model. |
| Research sample | We used 4,050 population-based studies that had measured height and weight in 232 million participants in 197 countries. Studies were representative of a national, subnational or community population. We used all available and accessible data which met the criteria described below. |
| Sampling strategy | This is a data pooling study which used all available and accessible data. These are population-based studies, each with sample size set to detect measure of interest in that study. These were pooled in a meta regression which provides more confidence in results by borrowing strength across studies. We included data collected using a probabilistic sampling method with a defined sampling frame. We therefore included studies with simple random and complex survey designs but excluded convenience samples. |
| Data collection | We used 4,050 population-based studies that had measured height and weight in 232 million participants in 197 countries. We used data on measured height and weight to calculate prevalence by sex and age group. We excluded self-reported data. |
| Timing | We pooled data collected from 1980 to 2024. We included national studies for the 3 years prior to start year, assigning them to the start year, and national studies for the year following the end year, assigning them to the end year, so that they can inform the estimates in countries with slightly earlier or more recent national data. We used all available data within these years which met the criteria described below. |
| Data exclusions | We excluded all data sources that were solely based on self-reported weight and height without a measurement component because these data are subject to biases that vary by geography, time, age, sex and socioeconomic characteristics. We also excluded data sources on population subgroups whose anthropometric status may differ systematically from the general population, including:<br>• studies that had included or excluded people based on their health status or cardiovascular risk;<br>• studies whose participants were only ethnic minorities;<br>• specific educational, occupational, or socioeconomic subgroups, with the exception noted below;<br>• those recruited through health facilities, with the exception noted below; and<br>• women aged 15-19 years in surveys which sampled only ever-married women or measured height and weight only among mothers.<br><br>We used school-based data in countries and age-sex groups with school enrolment of 70% or higher. We used data whose sampling frame was health insurance schemes in countries where at least 80% of the population were insured. Finally, we used data collected through general practice and primary care systems in high-income and central European countries with universal insurance, because contact with the primary care systems tends to be as good as or better than response rates for population-based surveys. |
| Non-participation | This was a secondary data analysis thus no participants were included in this study. |

| Randomization | Our study is an analysis of trends, and we did not carry out randomised experiments. |
|---|---|

# Reporting for specific materials, systems and methods

We require information from authors about some types of materials, experimental systems and methods used in many studies. Here, indicate whether each material, system or method listed is relevant to your study. If you are not sure if a list item applies to your research, read the appropriate section before selecting a response.

## Materials & experimental systems

| n/a | Involved in the study |
|---|---|
| ☒ ☐ | Antibodies |
| ☒ ☐ | Eukaryotic cell lines |
| ☒ ☐ | Palaeontology and archaeology |
| ☒ ☐ | Animals and other organisms |
| ☒ ☐ | Clinical data |
| ☒ ☐ | Dual use research of concern |
| ☒ ☐ | Plants |

## Methods

| n/a | Involved in the study |
|---|---|
| ☒ ☐ | ChIP-seq |
| ☒ ☐ | Flow cytometry |
| ☒ ☐ | MRI-based neuroimaging |

## Plants

| Seed stocks | N/A |
|---|---|
| Novel plant genotypes | N/A |
| Authentication | N/A |

