## [Peer Review file · Nature]

Obesity rise plateaus in developed nations and accelerates in developing nations

Corresponding Author: Professor Majid Ezzati

Version 0:

Reviewer comments:

Referee #1

(Remarks to the Author)

While the title and discussion focus on the decline of obesity in many countries, the data differ. For example, no country showed a decline of obesity among girls and only one country showed meaningful change among boys. The title and discussion must remove reversal. You might state slowing down but not reversal. For the other 3 countries the slowing down is very tiny or almost nonexistent.

The slowing down is not clear from the results for children.

Now for adult women it is clear that the obesity increase appears to have plateaued for 10 countries and for only 3 is there a meaningful decline. I see plateauing for men in a segment of countries but zero declines.

So, from the 4 samples, there is a tiny insignificant prevalence for a few countries overall, but a good number of increases seem to have plateaued. I strongly recommend the title and focus be on plateauing, remove decline from the title. Also make it clear in the text that these are statistically manipulated guestimates of the trends using the best methods available. These are not real changes, and repeated national surveys exist for only a number of higher income countries and few of the low- and middle-income countries.

In other words, with these estimates of obesity change, it is crucial to note and focus on plateauing and in the text note but do not focus on declines except if you want to focus on the only country with a meaningful decline. The others are trivial. From a public health and medical perspective.

It should also be noted that with extant national surveys, i.e. real data, if any countries show a decline. I am unsure of that.

(Remarks on code availability)

nothing to add. i have used these data at times but very carefully noting they are estimated trends.

Referee #2

(Remarks to the Author)

This is an extremely valuable paper, filled with the rich data we have come to expect from the NCD-RisC group. This fine-grained analysis of the obesity trajectories over the past 45 years will be especially important as we continue to try to understand the drivers of this pandemic. I particularly commend the authors for their helpful pattern analyses across the countries by age, sex and region and the individual country-level trajectories.

My major comment focusses on the discussion – the interpretation and next steps that might flow from this work. It seems to me that this study has identified several patterns that could profitably be teased out from each other. It is relatively straightforward to draw up quite a long list of factors which could influence obesity, but the findings from this paper can be

used to parse out the likely major ones and how they influence obesity trajectories (drivers, moderators or mediators).

- The heterogeneity of trajectories: these have been well commented upon. There are many factors (eg economic, policy, cultural, food and built environments) that could help explain the heterogeneity, and the authors did identify that some of those factors may be regional in nature. I think this does call for more comparative studies of determinants across countries and regions to identify those factors.
- Instructive outliers: related to the point above, there are outlier countries or regions at the extremes of trajectories (Central Asian countries, Pacific Islands nations, high-income Asian etc) where this work on these wider factors could fruitfully start and these could be noted.
- Commonalities of trajectories: despite the heterogeneity of trajectories, there are common patterns which I believe could be further explored. Therefore, how would this study modify the Obesity Transition concept? The common patterns of women having earlier and greater rises in obesity, children having lower and later rises in obesity, low- and middle-income countries having later rises in obesity should be noted and used to determine whether the proposed stages 1-3 of the Obesity Transition hold up (or not) from this study. The proposed stage 4 of declining prevalence was much more speculative suggesting that the first movers in the decline would be high income countries, children and women. This study can use its data to refine this common pattern of both the rise and early decline in the country-level epidemics. Identifying common patterns allows for instructive outliers to be better detected.
- Inflection points: about the year 2000 seems like it might be an important inflection point marking the start of the deceleration, flattening or decline in prevalence. Is that a correct interpretation of the data and what could that be pointing to? My recollection is that about 2000 is when obesity started to hit the news globally as an issue, but the authors might have other ideas.
- Does obesity beget obesity? – Figures 3 and 6 show what looks like a positive correlation between obesity prevalence and the velocity of change in prevalence. This was not formally tested as a correlation, but it may be telling us something: does obesity beget obesity (ie people with obesity become trapped in vicious cycles which accelerate the increases) or are the drivers and moderators which created the high prevalence in the first place, much harder to turn around at the societal level? These are important research questions to pursue.
- Global drivers: despite all the heterogeneity, there is one central message which needs to be stressed because it is easily forgotten (and is not mentioned in this paper). There is a massive, global, powerful driver of increasing obesity prevalence. This is the first critical message – something very powerful is driving up obesity in almost every country. This ‘something’ has not been named or agreed upon, yet it must be there in plain sight. Is it just multifactorial (as is the case for CVD and cancer) or is it systemic (like poverty)? This is a critical future research direction.

Minor comments:

- L25 – the various influences cited could attenuate OR accentuate the trajectories of countries.
- L36-37 – I think physical activity is a better term than energy expenditure, since most of the total energy expenditure is resting metabolic rate – the biggest determinant of which is body size or more specifically, lean mass size (not economic or cultural factors etc).
- Results – just as the Pacific Islands nations have been extracted as a group because of their very high obesity prevalence, the high-income Asian populations, especially women, have surprising low trajectories, and comments on their patterns would be useful. This population grouping has previously, and helpfully, been analysed separately in other papers from NCD-RisC.
- Results – could the authors please state whether any country has experienced a decline in prevalence amongst boys, girls, women and men.
- L253-254 – BMI is an imperfect measure of body fatness at a single point in time in a single individual, but population changes in BMI over time are a very precise measure of increasing population fatness. Height and weight are easily and accurately measured and there is no other plausible explanation for increasing mean BMI or BMI>30 (which are themselves highly correlated) than increasing fatness of the population. BMI as a measure should be removed from the limitations.
- L306 – energy expenditure of physical activity
- L329 – please rewrite this sentence without using reference 101. That paper reported on the ADJUSTED metabolic rates (ie adjusted for fat-free mass which is proportional to body weight). Thus, while the resting metabolic rate (kcal/kg FFM) may have declined over time, the total energy expenditure (kcal/day) increases with obesity and has increased over time. We measure energy intake in kcal/day so if it is being compared to energy expenditure, this should also be in kcal/day.
- L330 – I doubt that the flattening and decline in obesity in some high income countries is due to a plethora of high impact policies enacted from the early 2000s. All countries, including HICs, have been very sluggish in implementing serious obesity prevention policies.
- L336 – ‘A side-effect of such a response’ does this refer to the SSB taxes or the previous list of societal responses? I have read the methods section. The data collection, inclusion/exclusion, and cleaning were well described. The Bayesian analyses require specialist review and I am not qualified to judge these. However, this group has been publishing for years using this dataset and I am sure that its methods have been reviewed many times.

(Remarks on code availability)

Referee #3

(Remarks to the Author)

This study updates our understanding of the global obesity challenge by broadly identifying the diverging trajectories of change in obesity prevalence rates (i.e., between high- versus low- and middle-income countries) or obesity velocity, with a general trend towards plateauing in HICs and increases in LMICs, albeit with significant variations based on age and geography. The methodology supports these findings, and takes a cautious view to too narrowly attributing drivers, given the global variations and the limited quantitative research done (or data available) in most parts of the world. Most notable was

the finding that 2024 showed the highest velocity to date in more than 86 countries across genders, the overwhelming majority LMICs. This paper represents an important quantification of the burden and variations in its progress, profile, and patterns of change. In terms of suggested improvements, the paper acknowledges the many roots with a focus in the recommendations mainly on food, which although a main pillar might be strengthened by acknowledging studies that have raised questions about other influences such as endocrine-disrupting chemicals, the role played by genetics, and other areas that may prove to have significant influence.

(Remarks on code availability)

Apologies, this is not my area of expertise, as noted in my comments another colleague with expertise was not available.

Referee #4

(Remarks to the Author)

Summary: The Authors contribute a comprehensive analysis of global obesity dynamics from 1980 to 2024 across 200 countries and territories. The study, based on a massive dataset of 3,982 population-based studies with 231 million participants, found that the rise in obesity has decelerated, plateaued, or even reversed in many high-income countries, particularly among children and adolescents. Conversely, obesity prevalence is steadily increasing or accelerating in most low- and middle-income countries for both children and adults. The researchers used Bayesian hierarchical meta-regression modeling and clustering techniques to categorize and estimate the diverse national obesity trajectories, providing crucial data for targeted public health policy interventions.

Review: The Authors contribution provides a rigorous and principled analysis of global obesity dynamics by combining several data sources in a hierarchical Bayesian model developed to allow for borrowing of information within region and super-regions. I only have minor comments, especially relating to the statistical methodology and technical notation. Specifically, I found some of the notation confusing and potentially misleading - likewise, some of the technical definitions in the manuscript section detailing the statistical model used should be revised to be technically faithful to the implied modeling assumptions. Finally, while the statistical models used are quite flexible, it would be important to understand if the inference obtained can be validated in any sense, and whether sensitivities are to be expected in relation to the structure of the assumed hierarchy. Please see my specific comments below:

Major Comments:

[Model hierarchy] = The model hierarchy depends on the definition of regions, and super-regions (Supplementary table 1). These structures are pre-determined and allow for borrowing of information within hierarchies. Is this always appropriate? Some regions like "East Asia" include fairly heterogeneous countries with very different histories, e.g. China and Japan. Do the Authors have a sense of how sensitive are their results to perturbations of the pre-determined hierarchy?

[Hierarchical Inference] = Considering the Authors are fitting a hierarchical model, there seem to be an opportunity there to carry out inference at the different levels of the hierarchy.

[Clustering and statistical significance] = The clustering analysis is inherently exploratory. However, considering the vast amount of information used in inference, it may be useful to provide some measure of cluster significance.

[Model adequacy] = Anytime we reach for model-based inference it would be important to understand whether the models fitted provide an adequate representation of the data. Is there any measure of model adequacy that can be used in this case to provide a measure of trustworthiness for the achieved inference?

Minor Comments:

[Notation] I found the notation often confusing. Especially, the use of parenthesized subscripts, e.g. on line 871 a careless reader may think that $a_{[i]}$ is a parameter indexed by both the country index j and the study index i , whereas my understanding is that you have only as many random intercepts as the number of countries.

[Definitions] The definition of your sampling model on line 868 is technically incorrect, as it is only the conditional variable $y_{[h,i]} \mid p_{[h,i]}$ that is assumed Binomial, the marginal distribution of $y_{[h,i]}$ is much more general.

[Priors] Often the prior distributions used are not easily ascertained from your description. For example, I could not find the prior for the probit variance τ^2 . This parameter is often only weakly identified as it simply modifies the link function, so it would be good to understand its influence and sensitivity to prior assumptions.

(Remarks on code availability)

Version 1:

Reviewer comments:

Referee #1

(Remarks to the Author)

Overall: while it would be nice to strongly note reversals in countries this requires nationally representative surveys and stronger statistical analyses. I feel you should focus on plateauing with potential signs of a slight reversal in a few European countries.

First, thank you for changing the title. You still emphasize in the abstract far too much a decline over a slowdown for children given your data. You show hints of a decline, but it is not meaningful at this point. When one looks at country data from only nationally representative files and not the tons of tiny unrepresentative surveys that provide a good part of the European data one does not see much of a decline, it at all. You must recognize these data are fitted to a ton of tiny as well as larger and national surveys.

I want to see this abstract change.

Second the major contrast is low- and middle-income countries (LMICs) vs higher income ones (HICs). You might want to note this from the start. Far more obesity in those countries. It might be useful to highlight first the differential picture between LMICs and HICs.

Third, show me national evidence that this slowing down relates to more activity. I know of no such data whereas a lot of research highlights dietary effects. You must correct that set of comments. There were some studies suggesting huge declines in work energy mattered in increasing obesity in 1 or 2 LMICs but nothing on decreases in obesity linked with rising activity. This must be corrected throughout. This is all about food or so the data suggests that. While we absolutely need activity for a truly healthy lifestyle, except in limited countries [e.g., Scandinavia], such lifestyles barely exist except among small proportions of any population. The overwhelming data suggests the opposite trend. Even the research on GLP-1 suggests this also.

Fourth in the discussion:

"Our results demonstrate that generalising the trends in obesity as a global epidemic masks highly heterogeneous temporal dynamics across countries and in different age groups. In some high-income western countries, the velocity of obesity in children and adolescents began to slow down as early as around 1990 with the rise coming to a halt by the mid-2000s," Correct this appears to be reaching a plateau by the mid-2000's.

Remove reversal from obesity prevalence to only plateauing fits your data so remove the wording from the paragraph with comparison of other studies!

Limitations: a second limitation is that a sizable proportion of the studies represented in these NCD-Risc data are from small nonrepresentative surveys. You must note that.

Remove reversal so at most you note slowing down and plateauing as the major conclusion. And you might note there may be 1-2 exceptions.

(Remarks on code availability)

Referee #2

(Remarks to the Author)

The authors have comprehensively responded to the reviewer comments. I have no further comments to add.

(Remarks on code availability)

this is not my area of expertise

Referee #4

(Remarks to the Author)

NA

(Remarks on code availability)

We thank the Referees for their careful review, and their comments and suggestions. We have used these to conduct additional analyses and revise the paper, as detailed below, which helped enhance clarity and improve the presentation of results, and illustrated the robustness of the results. All page, line, figure, table and citation numbers refer to the tracked manuscript.

Referee #1

While the title and discussion focus on the decline of obesity in many countries, the data differ. For example, no country showed a decline of obesity among girls and only one country showed meaningful change among boys. The title and discussion must remove reversal. You might state slowing down but not reversal. For the other 3 countries the slowing down is very tiny or almost nonexistent.

We have removed “reversal” from the title as suggested (P. 1, Line 1). We now also emphasise that the estimated reversals are small and state the posterior probabilities that these are different from zero (P. 2, Line 19; P. 5, Lines 97-99; P. 8, Lines 175-177). At the same time, we note, and have stated in the paper, that our results on plateau as well as small declines are consistent with studies focusing on specific countries (P. 11, Lines 273-275; References 41-61).

The slowing down is not clear from the results for children.

Now for adult women it is clear that the obesity increase appears to have plateaued for 10 countries and for only 3 is there a meaningful decline. I see plateauing for men in a segment of countries but zero declines.

We state in the revised Results section that the declines for children were not statistically distinguishable from zero (P. 5, Lines 97-99). There are however multiple countries with near-zero or negative velocities for children and adolescents (Fig. 2; Fig. 3; Extended Data Fig. 9) which suggests that a plateau has been reached.

For adults, despite having a few countries where velocity was negative with a posterior probability >0.80 , we have stated that the negative velocities were small (P. 8, Lines 175-177).

So, from the 4 samples, there is a tiny insignificant prevalence for a few countries overall, but a good number of increases seem to have plateaued. I strongly recommend the title and focus be on plateauing, remove decline from the title. Also make it clear in the text that these are statistically manipulated guestimates of the trends using the best methods available. These are not real changes, and repeated national surveys exist for only a number of higher income countries and few of the low- and middle-income countries.

We have adjusted our text as described in our reply to the first two comments.

Regarding the nature of analysis and the data used, our estimates are fitted estimates, which as the Referee stated is the appropriate way to analyse trends using multiple data points. The fits can be using restrictive (e.g., linear time trend and age association) or more flexible (e.g., our statistical model) approaches, with the former typically having stronger assumptions about how data behave over space, age and time whereas our statistical model allows more flexibility to follow the data.

In terms of data availability, the revised manuscript provides information to demonstrate that our estimates are based on repeated measurements for most countries: specifically, 181 of the 200 countries had at least two national surveys, 166 had at least three, and 128 had at least five (Extended Data Fig. 1-2; P. 32, Lines 1016-1018). All regions had countries with multiple data sources, including many low- and middle-income countries. The availability of multiple studies enabled robust quantification of the dynamics in obesity trajectories, including

for countries with negative velocity in recent years. This can be seen for example in Figure R1 below, which shows the survey data for some countries with negative velocity, and that the estimates are supported by the data.

Figure R1: Measurement data (coloured points) and fitted estimates (black line) for some countries with negative velocity of obesity in 2024. As seen in the figure, the downward trend in estimates is also seen in the data.

Empirically, the amount of data available is directly reflected in the uncertainty in the estimated velocities. The revised manuscript provides extensive information on uncertainty (Extended Data Fig. 7-12), and reports on both the size of negative velocities and their posterior probabilities (P. 5, Lines 97-99; P. 8, Lines 175-177), which are a measure of the statistical confidence in the velocities being negative based on the available data.

In other words, with these estimates of obesity change, it is crucial to note and focus on plateauing and in the text note but do not focus on declines except if you want to focus on the only country with a meaningful decline. The others are trivial. From a public health and medical perspective.

Please see responses to related above comments for modifications to the paper regarding declines and plateaus, including additional details about the uncertainty and size of negative velocities.

It should also be noted that with extant national surveys, i.e. real data, if any countries show a decline. I am unsure of that.

Please see responses to related comments above on how the revisions in the paper clarify that we use extensive data these countries and estimate and document uncertainty around these estimates.

Referee #2

This is an extremely valuable paper, filled with the rich data we have come to expect from the NCD-RisC group. This fine-grained analysis of the obesity trajectories over the past 45 years will be especially important as we continue to try to understand the drivers of this pandemic. I particularly commend the authors for their helpful pattern analyses across the countries by age, sex and region and the individual country-level trajectories.

My major comment focusses on the discussion – the interpretation and next steps that might flow from this work. It seems to me that this study has identified several patterns that could profitably be teased out from each other. It is relatively straightforward to draw up quite a long list of factors which could influence obesity, but the findings from this paper can be used to parse out the likely major ones and how they influence obesity trajectories (drivers, moderators or mediators).

- **The heterogeneity of trajectories: these have been well commented upon. There are many factors (eg economic, policy, cultural, food and built environments) that could help explain the heterogeneity, and the authors did identify that some of those factors may be regional in nature. I think this does call for more comparative studies of determinants across countries and regions to identify those factors.**

- **Instructive outliers: related to the point above, there are outlier countries or regions at the extremes of trajectories (Central Asian countries, Pacific Islands nations, high-income Asian etc) where this work on these wider factors could fruitfully start and these could be noted.**

We have stated the value of and need for such future studies, and the data and methods required in the revised Discussions, including the balance of multi-country analyses and in-depth case studies in countries with particularly promising or alarming trends (PP. 15-16, Lines 385-396).

- **Commonalities of trajectories: despite the heterogeneity of trajectories, there are common patterns which I believe could be further explored. Therefore, how would this study modify the Obesity Transition concept? The common patterns of women having earlier and greater rises in obesity, children having lower and later rises in obesity, low- and middle-income countries having later rises in obesity should be noted and used to determine whether the proposed stages 1-3 of the Obesity Transition hold up (or not) from this study. The proposed stage 4 of declining prevalence was much more speculative suggesting that the first movers in the decline would be high income countries, children and women. This study can use its data to refine this common pattern of both the rise and early decline in the country-level epidemics. Identifying common patterns allows for instructive outliers to be better detected.**

Obesity transition (which included the senior author of the current manuscript as a co-author) is a general framework for qualitative description of the rise, plateau and potential decline in obesity prevalence. Our results do not modify the framework, which is qualitative and relatively simple, but rather demonstrate that the dynamics of obesity have complexities and heterogeneities, in relation to region, sex and age group, that go beyond the original framework. For example, our results show that “earlier plateauing of the rise in obesity prevalence in children and adolescents than in adults in high-income countries, whereas in most other regions deceleration and plateauing of the rise in prevalence occurred in adults before they did in children and adolescents or while obesity continued to increase in children and adolescents” (P. 14, Lines 352-356). Similarly, in relation to sex, our results show that

whether velocity was higher in women or in men varied across regions (P. 14, Lines 357-359) as did the occurrence of a plateau (P. 14, Lines 359-361).

These enrichments are stated in Discussion with specific country/region, sex and age examples (PP. 13-14, Lines 313-361), followed by why they may have occurred (PP. 15-16, Lines 363-404).

Finally, following the suggestions of Referee #1, we have presented a more cautious statement regarding stage 4, emphasising that its confirmation requires additional years of data (P. 13, Lines 322-324).

• Inflection points: about the year 2000 seems like it might be an important inflection point marking the start of the deceleration, flattening or decline in prevalence. Is that a correct interpretation of the data and what could that be pointing to? My recollection is that about 2000 is when obesity started to hit the news globally as an issue, but the authors might have other ideas.

We agree that changes in trends in many high-income countries are visible around the year 2000. We have stated in results that the plateau of obesity in children and adolescents in high-income countries mostly happened between the 1990s and mid-2000s (P. 5, Lines 90-94) and that the deceleration and plateau in adults occurred around or after 2000 (P. 8, Lines 168-173).

We also agree that the global attention for this topic changed around this time. We now note the increasing public discourse and information provision about obesity around this time in the Discussion section, including the first Report of the Surgeon General on obesity and higher media coverage (PP. 16-17, Lines 415-419; References 106-108). We also state in the revised manuscript that this may have allowed people with higher education and income to leverage the information and modify nutrition and physical activity, with an aggregate as well as inequality impact (PP. 16-17, Lines 411-420).

• Does obesity beget obesity? – Figures 3 and 6 show what looks like a positive correlation between obesity prevalence and the velocity of change in prevalence. This was not formally tested as a correlation, but it may be telling us something: does obesity beget obesity (ie people with obesity become trapped in vicious cycles which accelerate the increases) or are the drivers and moderators which created the high prevalence in the first place, much harder to turn around at the societal level? These are important research questions to pursue.

We have added correlation coefficients to these figures which show moderate correlation for children and adolescents, and weak correlation for adults (Fig. 3 and Fig. 6) stated them in Results (P. 7, Lines 144-146; P. 10, Lines 233-235) and Discussion (P. 17, Lines 432-435). Our interpretation is that these moderate correlations do not have an individual-level physiological driver related to weight maintenance because such a scenario would not necessarily lead to a change in prevalence. Rather, they are a result of societal drivers and moderators as stated by the Referee.

• Global drivers: despite all the heterogeneity, there is one central message which needs to be stressed because it is easily forgotten (and is not mentioned in this paper). There is a massive, global, powerful driver of increasing obesity prevalence. This is the first critical message – something very powerful is driving up obesity in almost every country. This ‘something’ has not been named or agreed upon, yet it must be there in plain sight. Is it just multifactorial (as is the case for CVD and cancer) or is it systemic (like poverty)? This is a critical future research direction.

We have stated the continued steady or accelerating rise, which is especially relevant in low- and middle-income countries while the rise has at least peaked in wealthy nations, and its economic and technological drivers in Discussion (PP. 17-18, Lines 432-447).

Minor comments:

- **L25 – the various influences cited could attenuate OR accentuate the trajectories of countries.**

We have modified the statement as suggested (P. 2, Lines 26-29).

- **L36-37 – I think physical activity is a better term than energy expenditure, since most of the total energy expenditure is resting metabolic rate – the biggest determinant of which is body size or more specifically, lean mass size (not economic or cultural factors etc).**

We have modified as suggested (P. 3, Line 39).

- **Results – just as the Pacific Islands nations have been extracted as a group because of their very high obesity prevalence, the high-income Asian populations, especially women, have surprising low trajectories, and comments on their patterns would be useful. This population grouping has previously, and helpfully, been analysed separately in other papers from NCD-RisC.**

We have added to Abstract (P. 2, Line 14) and Results (P. 5, Line 87) the low prevalence and velocity from Japan and Taiwan to highlight this interesting region.

- **Results – could the authors please state whether any country has experienced a decline in prevalence amongst boys, girls, women and men.**

We note in the revised manuscript that Italy and Portugal had negative velocity for all four age group-sex combinations, although some of these were not distinguishable from zero at a high probability (P. 13, Lines 320-322).

- **L253-254 – BMI is an imperfect measure of body fatness at a single point in time in a single individual, but population changes in BMI over time are a very precise measure of increasing population fatness. Height and weight are easily and accurately measured and there is no other plausible explanation for increasing mean BMI or BMI>30 (which are themselves highly correlated) than increasing fatness of the population. BMI as a measure should be removed from the limitations.**

We agree and have therefore removed this limitation as suggested (P. 12, Lines 299-302).

- **L306 – energy expenditure of physical activity**

We have modified as suggested (P. 15, Line 369).

- **L329 – please rewrite this sentence without using reference 101. That paper reported on the ADJUSTED metabolic rates (ie adjusted for fat-free mass which is proportional to body weight). Thus, while the resting metabolic rate (kcal/kg FFM) may have declined over time, the total energy expenditure (kcal/day) increases with obesity and has increased over time. We measure energy intake in kcal/day so if it is being compared to energy expenditure, this should also be in kcal/day.**

We have modified as suggested (P. 16, Lines 410-411).

• **L330 – I doubt that the flattening and decline in obesity in some high income countries is due to a plethora of high impact policies enacted from the early 2000s. All countries, including HICs, have been very sluggish in implementing serious obesity prevention policies.**

We agree and have modified the text as suggested, both the specific sentence (P. 16, Lines 411-415), and additional text that emphasises the so-far-small impact of policies (PP. 16-17, Lines 415-429).

• **L336 – ‘A side-effect of such a response’ does this refer to the SSB taxes or the previous list of societal responses?**

It refers to those policies and programmes that relied on information, now clarified through both rewording and reorganisation (P. 17, Lines 419-429).

I have read the methods section. The data collection, inclusion/exclusion, and cleaning were well described. The Bayesian analyses require specialist review and I am not qualified to judge these. However, this group has been publishing for years using this dataset and I am sure that its methods have been reviewed many times.

We appreciate the support for NCD-RisC’s rigorous data collation and analytical methods.

Referee #3

This study updates our understanding of the global obesity challenge by broadly identifying the diverging trajectories of change in obesity prevalence rates (i.e., between high- versus low- and middle-income countries) or obesity velocity, with a general trend towards plateauing in HICs and increases in LMICs, albeit with significant variations based on age and geography. The methodology supports these findings, and takes a cautious view to too narrowly attributing drivers, given the global variations and the limited quantitative research done (or data available) in most parts of the world. Most notable was the finding that 2024 showed the highest velocity to date in more than 86 countries across genders, the overwhelming majority LMICs. This paper represents an important quantification of the burden and variations in its progress, profile, and patterns of change. In terms of suggested improvements, the paper acknowledges the many roots with a focus in the recommendations mainly on food, which although a main pillar might be strengthened by acknowledging studies that have raised questions about other influences such as endocrine-disrupting chemicals, the role played by genetics, and other areas that may prove to have significant influence.

We have added these additional factors in the revised Discussion (P. 16, Lines 396-399) and also discuss potential interactions with genetics and phenotypic characteristics that arise from foetal and early-life nutrition (P. 16, Lines 399-400).

Referee #4

Summary: The Authors contribute a comprehensive analysis of global obesity dynamics from 1980 to 2024 across 200 countries and territories. The study, based on a massive dataset of 3,982 population-based studies with 231 million participants, found that the rise in obesity has decelerated, plateaued, or even reversed in many high-income countries, particularly among children and adolescents. Conversely, obesity prevalence is steadily increasing or accelerating in most low- and middle-income countries for both children and adults. The researchers used Bayesian hierarchical meta-regression modeling and clustering techniques to categorize and

estimate the diverse national obesity trajectories, providing crucial data for targeted public health policy interventions.

Review: The Authors contribution provides a rigorous and principled analysis of global obesity dynamics by combining several data sources in a hierarchical Bayesian model developed to allow for borrowing of information within region and super-regions. I only have minor comments, especially relating to the statistical methodology and technical notation. Specifically, I found some of the notation confusing and potentially misleading - likewise, some of the technical definitions in the manuscript section detailing the statistical model used should be revised to be technically faithful to the implied modeling assumptions. Finally, while the statistical models used are quite flexible, it would be important to understand if the inference obtained can be validated in any sense, and whether sensitivities are to be expected in relation to the structure of the assumed hierarchy. Please see my specific comments below:

We appreciate the support for the statistical methodology and analysis, and have addressed these points as listed under specific comments below.

Major Comments:

[Model hierarchy] = The model hierarchy depends on the definition of regions, and super-regions (Supplementary table 1). These structures are pre-determined and allow for borrowing of information within hierarchies. Is this always appropriate? Some regions like "East Asia" include fairly heterogeneous countries with very different histories, e.g. China and Japan. Do the Authors have a sense of how sensitive are their results to perturbations of the pre-determined hierarchy?

This valid point illustrates the inherent challenge of "sharing information" in a global analysis. We have considered a fully spatial model, which does not require assignment of countries to regions, but is rather driven by physical proximity, as we and others have applied to national analyses^{1,2}. However, in a global analysis spatial models not only have difficulty handling island nations, but also overlook large heterogeneities among neighbouring countries. For example, in a fully spatial model, Russia would share information with its neighbours which include countries as epidemiologically heterogeneous as Finland, China, North Korea, Mongolia and Azerbaijan. At the other extreme, a structure that relies on epidemiology only might group Malaysia and Mexico together or island nations in the Pacific and the Caribbean together, which is not only non-intuitive but may also require changing the structure over time.

To enable epidemiologically plausible sharing of information and intuitive communication, we have selected our regions to have a balance of geography and epidemiology. We also note that the influence of hierarchy on the fitted estimates is small for countries where multiple data sources are available, which is the overwhelming majority of countries in our data. We show in Figure R2 that China and Japan, as well as their neighbours South Korea and Taiwan, have multiple national surveys that allowed the estimated trends to follow the country-specific data.

Figure R2: Model fits and data points for China, Japan, South Korea and Taiwan.

In the specific case of China, we had in the past grouped it with southeast Asia. However, careful analyses of trends in China alone^{3,4}, as well as exploratory analysis of its data in relation to its neighbours (Figure R3) showed that it was increasingly similar to its industrialised neighbours in East Asia.

Figure R3: Data on obesity from China (coloured points) compared to other countries (grey points) in the East Asia region.

There are nonetheless countries that have features that make them stand out within their region – for example Haiti in the Caribbean, Mauritius in east Africa or Afghanistan in south Asia. The results of the cross-validation analyses, stated under a later comment below, show that the current hierarchy performs well in terms of estimating trends with small errors in both internal and external predictive validity analyses.

An alternative, and somewhat radical, approach would be to remove the hierarchy entirely, so that all countries are “exchangeable” and borrowing of information happens to the same extent for all the countries. The difference to our main model is particularly noticeable for countries with no data because they are now influenced by a global average rather than a regional average. For example, under this approach, the three countries we do not have any data for (Bermuda, Djibouti and North Korea) take the same global average estimates instead of the regional average in the Caribbean, east Africa and southeast Asia (Figure R4).

Figure R4: Model estimates for three countries with no data using global average only (dashed blue lines) and using full hierarchical model accounting for global, super-regional and regional averages (solid black lines). The grey lines show the estimates for other countries in the respective regions of these three countries.

These results collectively demonstrate that the influence of current hierarchy is modest given that we have data for most countries. Together with relatively intuitive grouping of countries, this supports the current hierarchical structure.

We have nonetheless stated in the revised manuscript that there may be alternative hierarchical structures (P. 12, Lines 295-297). If there are specific suggestions for alternative hierarchical structures, we would be happy to compare results with those of the current model.

[Hierarchical Inference] = Considering the Authors are fitting a hierarchical model, there seem to be an opportunity there to carry out inference at the different levels of the hierarchy.

As stated by the Referee, it is possible to report quantities of interest at the region and super-region level directly from a hierarchical model. By the nature of a hierarchical model, these

numbers treat countries as “exchangeable” and are only affected by the distributions of data across countries⁵ rather than the size of their population, with USA and Canada, India and Bhutan, Brazil and Uruguay, or Nigeria and The Gambia contributing about the same to the respective regional estimates. These estimates would be different from regional results that are built up from country estimates by population weighting, especially because differences in population across countries have changed over time, making them hard to interpret for non-statistical readers. If these “average” quantities are of interest, we can add as Supplementary Information attempting to distinguish them from those based on population weighting of country estimates.

[Clustering and statistical significance] = The clustering analysis is inherently exploratory. However, considering the vast amount of information used in inference, it may be useful to provide some measure of cluster significance.

To our knowledge, traditional significance tests are not defined for k-means clustering and the alternatives⁶ involve strong assumptions about the data. We have however tested the stability of the clusters by calculating the Jaccard index⁷⁻⁹, as described in the revised manuscript (PP. 48-49, Lines 1445-1471; Supplementary Table 5). These analyses show that the clusters are robustly separated.

[Model adequacy] = Anytime we reach for model-based inference it would be important to understand whether the models fitted provide an adequate representation of the data. Is there any measure of model adequacy that can be used in this case to provide a measure of trustworthiness for the achieved inference?

To test our model’s performance, we have conducted internal and external (i.e., cross-validation) predictive validity analysis. The validation process is described in detail a newly added section on *Validation of statistical model* (PP. 44-46, Lines 1349-1402) and its results reported in Supplementary Table 4. The results of the validation analysis show that our approach performed very well in terms of how it estimates obesity in different forms of data missingness and scarcity.

Minor Comments:

[Notation] I found the notation often confusing. Especially, the use of parenthesized subscripts, e.g. on line 871 a careless reader may think that $a_{j[i]}$ is a parameter indexed by both the country index j and the study index i , whereas my understanding is that you have only as many random intercepts as the number of countries.

The referee’s understanding is correct regarding the number of random effects. We have used the $j[i]$ notation because we wanted to explicitly highlight that the country index j is uniquely determined by the study index i . We would be happy to follow the Referee’s specific suggestions to further improve clarity.

[Definitions] The definition of your sampling model on line 868 is technically incorrect, as it is only the conditional variable $y_{h,i} \mid p_{h,i}$ that is assumed Binomial, the marginal distribution of $y_{h,i}$ is much more general.

We have now reworded this sentence explicitly stating the conditioning on $p_{h,i}$ and $n_{h,i}$ (P. 34, Lines 1080-1081).

[Priors] Often the prior distributions used are not easily ascertained from your description. For example, I could not find the prior for the probit variance τ^2 . This

parameter is often only weakly identified as it simply modifies the link function, so it would be good to understand its influence and sensitivity to prior assumptions.

We have included a table of the model parameters and their priors (Supplementary Table 3); all priors on standard deviations are non-informative so that the inference is driven by data. Specifically, the probit variance term τ^2 captures residual age-by-study variability not captured by the e 's which are equal across all observations in any given study. For example, a study may have a particularly high obesity prevalence in an age group compared to all other ages, possibly because of a poorer response rate in this age group. As stated in Supplementary Table 3, τ^2 has a non-informative prior which we expect to have little influence on the estimates compared to more informative or restrictive priors¹⁰.

Other revisions

Since submission, we have downloaded or received from NCD-RisC collaborators additional data sources. We have included these data in the NCD-RisC database and re-run our statistical model. The additional data provide additional information for some countries but result in only small changes to the numerical results; the conclusions remain unchanged.

Editors' comments

Regarding Referee 1's comments, while we do not view these as reasons to preclude further consideration of the study, we encourage you to provide a thorough response to the points raised and to make it clearer in the manuscript that the analysis is not based on actual data, together with a fuller discussion of the related implications and caveats.

We have clarified that our analysis fits the extensive data available with a Bayesian statistical model (akin to putting a line through the data) to estimate prevalence by country before using these estimates to calculate velocity (P. 4, Lines 66-74; P. 27, Lines 872-885). All of these steps, including data collation and model fitting as well as calculation of velocity based on the estimated prevalences, are described in the paper. We would be happy to further clarify these steps for the readers based on Editors' and Referees' suggestions.

References

- 1 Dwyer-Lindgren, L. *et al.* US county-level trends in mortality rates for major causes of death, 1980-2014. *JAMA* **316**, 2385-2401 (2016). <https://doi.org/10.1001/jama.2016.13645>
- 2 Rashid, T. *et al.* Life expectancy and risk of death in 6791 communities in England from 2002 to 2019: high-resolution spatiotemporal analysis of civil registration data. *Lancet Public Health* **6**, e805-e816 (2021). [https://doi.org/10.1016/S2468-2667\(21\)00205-X](https://doi.org/10.1016/S2468-2667(21)00205-X)
- 3 Wang, L. *et al.* Body-mass index and obesity in urban and rural China: findings from consecutive nationally representative surveys during 2004-18. *Lancet* **398**, 53-63 (2021). [https://doi.org/10.1016/S0140-6736\(21\)00798-4](https://doi.org/10.1016/S0140-6736(21)00798-4)
- 4 Zhang, M. *et al.* Prevalence, awareness, treatment, and control of hypertension in China, 2004-18: findings from six rounds of a national survey. *BMJ* **380**, e071952 (2023). <https://doi.org/10.1136/bmj-2022-071952>
- 5 Gelman, A. & Hill, J. *Data Analysis Using Regression and Multilevel/Hierarchical Models*. (Cambridge University Press, 2006).
- 6 Chen, Y. T. & Witten, D. M. Selective inference for k-means clustering. *J Mach Learn Res* **24**, 1-41 (2023).
- 7 Hennig, C. Cluster-wise assessment of cluster stability. *Comput Stat Data Anal* **52**, 258-271 (2007). <https://doi.org/https://doi.org/10.1016/j.csda.2006.11.025>
- 8 Tang, M. *et al.* Evaluating single-cell cluster stability using the Jaccard similarity index. *Bioinformatics* **37**, 2212-2214 (2021). <https://doi.org/10.1093/bioinformatics/btaa956>
- 9 Cebola, I. *et al.* Epigenetics override pro-inflammatory PTGS transcriptomic signature towards selective hyperactivation of PGE2 in colorectal cancer. *Clin Epigenetics* **7**, 74 (2015). <https://doi.org/10.1186/s13148-015-0110-4>
- 10 Gelman, A. Prior distributions for variance parameters in hierarchical models (comment on article by Browne and Draper). *Bayesian Anal* **1**, 515-534 (2006).

We thank the Reviewers for reading the revised manuscript, confirming the improvements in content and presentation, and for additional comments and suggestions. We have used these to revise the paper, as detailed below. All page and line numbers refer to the tracked manuscript.

Referee #1

Overall: while it would be nice to strongly note reversals in countries this requires nationally representative surveys and stronger statistical analyses. I feel you should focus on plateauing with potential signs of a slight reversal in a few European countries. First, thank you for changing the title. You still emphasize in the abstract far too much a decline over a slowdown for children given your data. You show hints of a decline, but it is not meaningful at this point. When one looks at country data from only nationally representative files and not the tons of tiny unrepresentative surveys that provide a good part of the European data one does not see much of a decline, it at all. You must recognize these data are fitted to a ton of tiny as well as larger and national surveys.

I want to see this abstract change.

As suggested, we have reworded. We have used “indications of a decline” instead of “hints of a decline” as it seemed more suitable to a journal publication (P. 2, Line 22); we can reword if necessary.

Second the major contrast is low- and middle-income countries (LMICs) vs higher income ones (HICs). You might want to note this from the start. Far more obesity in those countries. It might be useful to highlight first the differential picture between LMICs and HICs.

We tried restructuring the abstract but found that it was more effective if LMICs followed HICs as a contrasting phenomenon. We would be happy to work with the Editors on alternative structures.

Third, show me national evidence that this slowing down relates to more activity. I know of no such data whereas a lot of research highlights dietary effects. You must correct that set of comments. There were some studies suggesting huge declines in work energy mattered in increasing obesity in 1 or 2 LMICs but nothing on decreases in obesity linked with rising activity. This must be corrected throughout. This is all about food or so the data suggests that. While we absolutely need activity for a truly healthy lifestyle, except in limited countries [e.g., Scandinavia], such lifestyles barely exist except among small proportions of any population. The overwhelming data suggests the opposite trend. Even the research on GLP-1 suggests this also.

We agree and have shifted the emphasis towards food as suggested (P. 2, Lines 31-37; P. 15, Lines 378-380; P. 16, Lines 405-412), except for the introduction where physical activity is mentioned as a generic determinant. In the Discussion, as suggested we have highlighted the fact that the role of physical activity is rare using the example of Scandinavian countries (P. 15, Line 379).

Fourth in the discussion:

“ Our results demonstrate that generalising the trends in obesity as a global epidemic masks highly heterogeneous temporal dynamics across countries and in different age groups. In some high-income western countries, the velocity of obesity in children and adolescents began to slow down as early as around 1990 with the rise coming to a halt by the mid-2000s,”

Correct this appears to be reaching a plateau by the mid-2000’s.

To the best of our understanding, this comment confirms that wording we have used; we have nonetheless changed the reversal statement here as we did in the Abstract (P. 12, Lines 303-305).

Remove reversal from obesity prevalence to only plateauing fits your data so remove the wording from the paragraph with comparison of other studies!

We have done as suggested (P. 48, Lines 1452-1456).

Referee #2

The authors have comprehensively responded to the reviewer comments. I have no further comments to add.

Referee #4

NA

Editors' comments

1. Please reduce the length of the title to 75 characters (with spaces) or less, so that it fits on two lines in the final layout.

We have shortened the title. It is slightly longer than 75 characters but based on previous papers, it should fit two lines. In case it is still too long, we have provided two alternative titles. We are happy to take advice from the Editors and the Production team.

2. Please provide the manuscript in .docx format. Currently it is in pdf format.

We have uploaded in the specified file format.

3. Please reduce the Abstract to 230 words or less. Currently there are 311 words.

We have shortened as specified.

4. The number of references should generally not exceed 60. Currently you have 139 references

We have reduced the number of references to 59.

5. Please remove the main figures from the article file and re-supply them individually in an acceptable format such as EPS, AI, PS, PDF, PPT, PSD or XLS (for graphs) with editable vector files.

We have separated the figures as specified.

6. Please reduce subheadings to 40 characters (with spaces) or less.

We have done as specified throughout.

7. Please reduce the total number of Extended Data items to 10, if possible. These ED items can be multi paneled, but each ED figure must fit into a single A4 page. If the existing ED items cannot be consolidated into 10 (or moved to the SI), please let me know.

We have reduced Extended Data items to nine and they each fit into a single A4 page.

8. There are potential third party rights issues in the figures. Please check the sources of all illustrations and clarify whether permissions are needed to adapt or reproduce them. Please make sure to include the relevant details in third party rights table when you resubmit (more information below). If Biorender or a similar software has been used, please also ensure to provide relevant licenses. In particular please check figure/s: main figures 1, 2, 4, 5, ed figure 1, 11, 12

We have generated all the figures via code so there should be no third party rights issues. We would be happy to further clarify as needed.

9. Please make sure to provide a third party rights table (more information below) when you resubmit. If Biorender or similar software has been used, please also ensure to provide relevant licenses.

As above, we have generated all the figures via code so there should be no third party rights issues. We would be happy to further clarify as needed.

10. Flagging that there are display items in the SI - please check if any of these materials should be extended data.

We have used Extended Data for nine figures that provide essential additional information for the paper, and SI for those that are less central and may be of interest to only some readers. We can easily move some of the SI to Extended Data if any of them is deemed to better fit there by the Editors.

11. Please ensure that the text size in all figures is at least 5 pt Arial.

We have done so, consistently with analogous figures in previous papers.

12. Please remove the Extended data figures from the article file and re-supply them individually in EPS, JPEG or TIF format.

We have had some issues with EPS conversion but are working on this and should have ready shortly to replace the PDF files.

13. The code availability statement states that, 'The computer codes for the Bayesian hierarchical model and clustering analysis used in this work will be available at www.ncdrisc.org and Zenodo upon publication of the paper.' Please provide an accessible Zenodo web-link for the same, in the reporting summary.

We have generated the Zenodo link and added to the paper and reporting summary.

14. The data availability statement states that, 'Age-standardised and age-specific results of this study will be available from www.ncdrisc.org in machine-readable numerical format and as visualizations upon publication of the paper.' A request has been made to the authors for the public release of these data, in the reporting summary.

The download and visualisation sites are ready. Releasing them now would violate the embargo as the results will become publicly accessible. To ensure that the embargo on results is maintained, and consistent with previous papers, we will release them at the exact time when the paper is published and embargo is lifted. We have changed to present tense to reflect this simultaneous release.

Referee Report

2025-11-11

Summary: The Authors contribute a comprehensive analysis of global obesity dynamics from 1980 to 2024 across 200 countries and territories. The study, based on a massive dataset of 3,982 population-based studies with 231 million participants, found that the rise in obesity has decelerated, plateaued, or even reversed in many high-income countries, particularly among children and adolescents. Conversely, obesity prevalence is steadily increasing or accelerating in most low- and middle-income countries for both children and adults. The researchers used Bayesian hierarchical meta-regression modeling and clustering techniques to categorize and estimate the diverse national obesity trajectories, providing crucial data for targeted public health policy interventions.

Review: The Authors contribution provides a rigorous and principled analysis of global obesity dynamics by combining several data sources in a hierarchical Bayesian model developed to allow for borrowing of information within region and super-regions. I only have minor comments, especially relating to the statistical methodology and technical notation. Specifically, I found some of the notation confusing and potentially misleading - likewise, some of the technical definitions in the manuscript section detailing the statistical model used should be revised to be technically faithful to the implied modeling assumptions. Finally, while the statistical models used are quite flexible, it would be important to understand if the inference obtained can be validated in any sense, and whether sensitivities are to be expected in relation to the structure of the assumed hierarchy. Please see my specific comments below:

Major Comments:

[Model hierarchy] = The model hierarchy depends on the definition of regions, and super-regions (Supplementary table 1). These structures are pre-determined and allow for borrowing of information within hierarchies. Is this always appropriate? Some regions like “East Asia” include fairly heterogeneous countries with very different histories, e.g. China and Japan. Do the Authors have a sense of how sensitive are their results to perturbations of the pre-determined hierarchy?

[Hierarchical Inference] = Considering the Authors are fitting a hierarchical model, there seem to be an opportunity there to carry out inference at the different levels of the hierarchy.

[Clustering and statistical significance] = The clustering analysis is inherently exploratory. However, considering the vast amount of information used in inference, it may be useful to provide some measure of cluster significance.

[Model adequacy] = Anytime we reach for model-based inference it would be important to understand whether the models fitted provide an adequate representation of the data. Is there any measure of model adequacy that can be used in this case to provide a measure of trustworthiness for the achieved inference?

Minor Comments:

[Notation] I found the notation often confusing. Especially, the use of parenthesized subscripts, e.g. on line 871 a careless reader may think that $a_{j[i]}$ is a parameter indexed by both the country index j and the study index i , whereas my understanding is that you have only as many random intercepts as the number of countries.

[Definitions] The definition of your sampling model on line 868 is technically incorrect, as it is only the conditional variable $y_{h,i} | p_{h,i}$ that is assumed Binomial, the marginal distribution of $y_{h,i}$ is much more general.

[Priors] Often the prior distributions used are not easily ascertained from your description. For example, I could not find the prior for the probit variance τ^2 . This parameter is often only weakly identified as it

simply modifies the link function, so it would be good to understand its influence and sensitivity to prior assumptions.